# When is Agnostic Reinforcement Learning Statistically Tractable?*

**Zeyu Jia**[1]     **Gene Li**[2]     **Alexander Rakhlin**[1]     **Ayush Sekhari**[1]     **Nathan Srebro**[2]

[1]MIT, [2]TTIC

## Abstract

We study the problem of agnostic PAC reinforcement learning (RL): given a policy class $\Pi$, how many rounds of interaction with an unknown MDP (with a potentially large state and action space) are required to learn an $\varepsilon$-suboptimal policy with respect to $\Pi$? Towards that end, we introduce a new complexity measure, called the *spanning capacity*, that depends solely on the set $\Pi$ and is independent of the MDP dynamics. With a generative model, we show that for any policy class $\Pi$, bounded spanning capacity characterizes PAC learnability. However, for online RL, the situation is more subtle. We show there exists a policy class $\Pi$ with a bounded spanning capacity that requires a superpolynomial number of samples to learn. This reveals a surprising separation for agnostic learnability between generative access and online access models (as well as between deterministic/stochastic MDPs under online access). On the positive side, we identify an additional *sunflower* structure, which in conjunction with bounded spanning capacity enables statistically efficient online RL via a new algorithm called POPLER, which takes inspiration from classical importance sampling methods as well as techniques for reachable-state identification and policy evaluation in reward-free exploration.

## 1 Introduction

Reinforcement Learning (RL) has emerged as a powerful paradigm for solving complex decision-making problems, demonstrating impressive empirical successes in a wide array of challenging tasks, from achieving superhuman performance in the game of Go (Silver et al., 2017) to solving intricate robotic manipulation tasks (Lillicrap et al., 2016; Akkaya et al., 2019; Ji et al., 2023). Many practical domains in RL often involve rich observations such as images, text, or audio (Mnih et al., 2015; Li et al., 2016; Ouyang et al., 2022). Since these state spaces can be vast and complex, traditional tabular RL approaches (Kearns and Singh, 2002; Brafman and Tennenholtz, 2002; Azar et al., 2017; Jin et al., 2018) cannot scale. This has led to a need to develop provable and efficient approaches for RL that utilize *function approximation* to generalize observational data to unknown states/actions.

The goal of this paper is to study the sample complexity of policy-based RL, which is arguably the simplest setting for RL with function approximation (Kearns et al., 1999; Kakade, 2003). In policy-based RL, an abstract function class $\Pi$ of *policies* (mappings from states to actions) is given to the learner. For example, $\Pi$ can be the set of all the policies represented by a certain deep neural network architecture. The objective of the learner is to interact with an unknown MDP to find a policy $\widehat{\pi}$ that competes with the best policy in $\Pi$, i.e., for some prespecified $\varepsilon$, the policy $\widehat{\pi}$ satisfies

$$V^{\widehat{\pi}} \geq \max_{\pi \in \Pi} V^{\pi} - \varepsilon, \tag{1}$$

where $V^{\pi}$ denotes the value of policy $\pi$ on the underlying MDP. We henceforth call Eq. (1) the "agnostic PAC reinforcement learning" objective. Our paper addresses the following question:

---

*Authors are listed in alphabetical order of their last names.

*What structural assumptions on $\Pi$ enable*
*statistically efficient agnostic PAC reinforcement learning?*

Characterizing (agnostic) learnability for various problem settings is perhaps the most fundamental question in statistical learning theory. For the simpler setting of supervised learning (which is RL with binary actions, horizon 1, and binary rewards), the story is complete: a hypothesis class $\Pi$ is agnostically learnable if and only if its VC dimension is bounded (Vapnik and Chervonenkis, 1971, 1974; Blumer et al., 1989; Ehrenfeucht et al., 1989), and the ERM algorithm—which returns the hypothesis with the smallest training loss—is statistically optimal (up to log factors). However, RL (with $H > 1$) is significantly more challenging, and we are still far from a rigorous understanding of when agnostic RL is statistically tractable, or what algorithms to use in large-scale RL problems.

While significant effort has been invested over the past decade in both theory and practice to develop algorithms that utilize function approximation, existing theoretical guarantees require additional assumptions on the MDP. The most commonly adopted assumption is *realizability*: the learner can precisely model the value function or the dynamics of the underlying MDP (see, e.g., Russo and Van Roy, 2013; Jiang et al., 2017; Sun et al., 2019; Wang et al., 2020c; Du et al., 2021; Jin et al., 2021a; Foster et al., 2021a). Unfortunately, realizability is a fragile assumption that rarely holds in practice. Moreover, even mild misspecification can cause catastrophic breakdown of theoretical guarantees (Du et al., 2019b; Lattimore et al., 2020). Furthermore, in various applications, the optimal policy $\pi^\star := \arg\max_{\pi \in \Pi} V^\pi$ may have a succinct representation, but the optimal value function $V^\star$ can be highly complex, rendering accurate approximation of dynamics/value functions infeasible without substantial domain knowledge (Dong et al., 2020). Thus, we desire algorithms for agnostic RL that can work with *no modeling assumptions on the underlying MDP*. On the other hand, it is also well known without any assumptions on $\Pi$, when $\Pi$ is large and the MDP has a large state and action space, agnostic RL may be intractable with sample complexity scaling exponentially in the horizon (Agarwal et al., 2019). Thus, some structural assumptions on $\Pi$ are needed, and towards that end, the goal of our paper is to understand what assumptions are sufficient or necessary for statistically efficient agnostic RL, and to develop provable algorithms for learning. Our main contributions are:

- We introduce a new complexity measure called the *spanning capacity*, which solely depends on the policy class $\Pi$ and is independent of the underlying MDP. We illustrate the spanning capacity with examples, and show why it is a natural complexity measure for agnostic PAC RL (Section 3).

- We show that the spanning capacity is both necessary and sufficient for agnostic PAC RL with a generative model, with upper and lower bounds matching up to $\log|\Pi|$ and $\mathrm{poly}(H)$ factors (Section 4). Thus, bounded spanning capacity characterizes agnostic PAC learnability in RL with a generative model.

- Moving to the online setting, we first show that the bounded spanning capacity by itself is *insufficient* for agnostic PAC RL by proving a superpolynomial lower bound on the sample complexity required to learn a specific $\Pi$, thus demonstrating a separation between generative and online interaction models for agnostic PAC RL (Section 5).

- Given the previous lower bound, we propose an additional property of the policy class called the *sunflower* property, that allows for efficient exploration and is satisfied by many policy classes of interest. We provide a new agnostic PAC RL algorithm called POPLER that is statistically efficient whenever the given policy class has both bounded spanning capacity and the sunflower property (Section 6). POPLER leverages importance sampling as well as reachable state identification techniques to estimate the values of policies. Our algorithm and analysis utilize a new tool called the *policy-specific Markov reward process*, which may be of independent interest.

## 2 Setup and Motivation

We begin by introducing our setup for reinforcement learning (RL), the relevant notation, and the goal of agnostic RL.

### 2.1 RL Preliminaries

We consider reinforcement learning in an episodic Markov decision process (MDP) with horizon $H$.

**Markov Decision Processes.** Denote the MDP as $M = \mathrm{MDP}(\mathcal{S}, \mathcal{A}, P, R, H, \mu)$, which consists of a state space $\mathcal{S}$, action space $\mathcal{A}$, horizon $H$, probability transition kernel $P : \mathcal{S} \times \mathcal{A} \to \Delta(\mathcal{S})$, reward function $R : \mathcal{S} \times \mathcal{A} \to \Delta([0, 1])$, and initial distribution $\mu \in \Delta(\mathcal{S})$. For ease of exposition, we assume that $\mathcal{S}$ and $\mathcal{A}$ are finite (but possibly large) with cardinality $S$ and $A$ respectively. We assume a layered state space, i.e., $\mathcal{S} = \mathcal{S}_1 \cup \mathcal{S}_2 \cup \cdots \cup \mathcal{S}_H$ where $\mathcal{S}_i \cap \mathcal{S}_j = \emptyset$ for all $i \neq j$. Thus, given a state $s \in \mathcal{S}$, it can be inferred which layer $\mathcal{S}_h$ in the MDP it belongs to. We denote a trajectory $\tau = (s_1, a_1, r_1, \ldots, s_H, a_H, r_H)$, where at each step $h \in [H]$, an action $a_h \in \mathcal{A}$ is played, a reward $r_h$ is drawn independently from the distribution $R(s_h, a_h)$, and each subsequent state $s_{h+1}$ is drawn from $P(\cdot|s_h, a_h)$. Lastly, we assume that the cumulative reward of any trajectory is bounded by 1.

**Policy-based Reinforcement Learning.** We assume that the learner is given a policy class $\Pi \subseteq \mathcal{A}^{\mathcal{S}}$.[2] For any policy $\pi \in \mathcal{A}^{\mathcal{S}}$, we denote $\pi(s)$ as the action that $\pi$ takes when presented a state $s$. We use $\mathbb{E}^{\pi}[\cdot]$ and $\mathbb{P}^{\pi}[\cdot]$ to denote the expectation and probability under the process of a trajectory drawn from the MDP $M$ by policy $\pi$. Additionally, for any $h, h' \leq H$, we say that a partial trajectory $\tau = (s_h, a_h, s_{h+1}, a_{h+1}, \ldots, s_{h'}, a_{h'})$ is consistent with $\pi$ if for all $h \leq i \leq h'$, we have $\pi(s_i) = a_i$. We use the notation $\pi \rightsquigarrow \tau$ to denote that $\tau$ is consistent with $\pi$.

The state-value function (also called $V$-*function*) and state-action-value function (also called $Q$-*function*) are defined such that for any $\pi$, and $s, a$,

$$V_h^{\pi}(s) = \mathbb{E}^{\pi}\left[\sum_{h'=h}^{H} R(s_{h'}, a_{h'}) \mid s_h = s\right], \quad Q_h^{\pi}(s, a) = \mathbb{E}^{\pi}\left[\sum_{h'=h}^{H} R(s_{h'}, a_{h'}) \mid s_h = s, a_h = a\right].$$

Furthermore, whenever clear from the context, we denote $V^{\pi} := \mathbb{E}_{s_1 \sim \mu} V_1^{\pi}(s_1)$. Finally, for any policy $\pi \in \mathcal{A}^{\mathcal{S}}$, we also define the *occupancy measure* as $d_h^{\pi}(s, a) := \mathbb{P}^{\pi}[s_h = s, a_h = a]$ and $d_h^{\pi}(s) := \mathbb{P}^{\pi}[s_h = s]$.

**Models of Interaction.** We consider two standard models of interaction in the RL literature:

- **Generative Model.** The learner has access to a simulator which it can query for any $(s, a)$, and observe a sample $(s', r)$ drawn as $s' \sim P(\cdot|s, a)$ and $r \sim R(s, a)$.[3]
- **Online Interaction Model.** The learner can submit a (potentially non-Markovian) policy $\widetilde{\pi}$ and receive back a trajectory sampled by running $\widetilde{\pi}$ on the MDP. Since online access can be simulated via generative access, learning under online access is only more challenging than learning under generative access (up to a factor of $H$). Adhering to commonly used terminology, we will refer to RL under the online interaction model as "online RL".

We define $\mathcal{M}^{\mathrm{sto}}$ as the set of all (stochastic and deterministic) MDPs of horizon $H$ over the state space $\mathcal{S}$ and action space $\mathcal{A}$. Additionally, we define $\mathcal{M}^{\mathrm{detP}} \subset \mathcal{M}^{\mathrm{sto}}$ and $\mathcal{M}^{\mathrm{det}} \subset \mathcal{M}^{\mathrm{detP}}$ to denote the set of all MDPs with deterministic transitions but stochastic rewards, and of all MDPs with both deterministic transitions and deterministic rewards, respectively.

## 2.2 Agnostic PAC RL

Our goal is to understand the sample complexity of agnostic PAC RL, i.e., the number of interactions required to find a policy that can compete with the best policy within the given class $\Pi$ for the underlying MDP. An algorithm $\mathbb{A}$ is an $(\varepsilon, \delta)$-PAC RL algorithm for an MDP $M$, if after interacting with $M$ (either in the generative model or online RL), $\mathbb{A}$ returns a policy $\widehat{\pi}$ that satisfies the guarantee[4]

$$V^{\widehat{\pi}} \geq \max_{\pi \in \Pi} V^{\pi} - \varepsilon,$$

---

[2]Throughout the paper, we assume that the policy classes $\Pi$ under consideration consist of deterministic policies. Extending our work to stochastic policy classes is an interesting direction for future research.

[3]Within the generative model, one can further distinguish between a more restrictive "local" access model (also called the "reset" model), where the learner can query $(s, a)$ for any $s \in \mathcal{S}$ that it has seen already, or "global" access, where the learner can query for any $(s, a)$ without restriction. For the generative model, our upper bounds hold in the local access model, while our lower bounds hold for the global access model.

[4]Our results are agnostic in the sense that we do not make the assumption that the optimal policy for the underlying MDP is in $\Pi$, but instead, only wish to complete with the best policy in $\Pi$. We also do not assume that the learner has a value function class or a model class that captures the optimal value functions or dynamics.

with probability at least $1 - \delta$. For a policy class $\Pi$ and a MDP class $\mathcal{M}$, we say that $\mathbb{A}$ has sample complexity $n_{\mathsf{on}}^{\mathbb{A}}(\Pi, \mathcal{M}; \varepsilon, \delta)$ (resp. $n_{\mathsf{gen}}^{\mathbb{A}}(\Pi, \mathcal{M}; \varepsilon, \delta)$) if for every MDP $M \in \mathcal{M}$, $\mathbb{A}$ is an $(\varepsilon, \delta)$-PAC RL algorithm and collects at most $n_{\mathsf{on}}^{\mathbb{A}}(\Pi, \mathcal{M}; \varepsilon, \delta)$ trajectories in the online interaction model (resp. generative model) in order to return $\widehat{\pi}$.

We define the *minimax sample complexity* for agnostically learning $\Pi$ over $\mathcal{M}$ as the minimum sample complexity of any $(\varepsilon, \delta)$-PAC RL algorithm, i.e.

$$n_{\mathsf{on}}(\Pi, \mathcal{M}; \varepsilon, \delta) := \min_{\mathbb{A}} n_{\mathsf{on}}^{\mathbb{A}}(\Pi, \mathcal{M}; \varepsilon, \delta), \quad \text{and} \quad n_{\mathsf{gen}}(\Pi, \mathcal{M}; \varepsilon, \delta) := \min_{\mathbb{A}} n_{\mathsf{gen}}^{\mathbb{A}}(\Pi, \mathcal{M}; \varepsilon, \delta).$$

For brevity, when $\mathcal{M} = \mathcal{M}^{\mathsf{sto}}$, we will drop the dependence on $\mathcal{M}$ in our notation, e.g., we will write $n_{\mathsf{on}}(\Pi; \varepsilon, \delta)$ and $n_{\mathsf{gen}}(\Pi; \varepsilon, \delta)$ to denote $n_{\mathsf{on}}(\Pi, \mathcal{M}; \varepsilon, \delta)$ and $n_{\mathsf{gen}}(\Pi, \mathcal{M}; \varepsilon, \delta)$ respectively.

**Known Results in Agnostic RL.** We first note the following classical result which shows that agnostic PAC RL is statistically intractable in the worst case.

**Proposition 1** (No Free Lunch Theorem for RL; Krishnamurthy et al. (2016))**.** *There exists a policy class $\Pi$ for which the minimax sample complexity under a generative model is at least* $n_{\mathsf{gen}}(\Pi; \varepsilon, \delta) = \Omega(\min\{A^H, |\Pi|, SA\}/\varepsilon^2)$.

Since online RL is only harder than learning with a generative model, the lower bound in Proposition 1 extends to online RL. Proposition 1 is the analogue of the classical *No Free Lunch* results in statistical learning theory (Shalev-Shwartz and Ben-David, 2014); it indicates that without placing further assumptions on the MDP or the policy class $\Pi$ (e.g., by introducing additional structure or constraining the state/action space sizes, policy class size, or the horizon), sample efficient agnostic PAC RL is not possible.

Indeed, an almost matching upper bound of $n_{\mathsf{on}}(\Pi; \varepsilon, \delta) = \widetilde{\mathcal{O}}(\min\{A^H, |\Pi|, HSA\}/\varepsilon^2)$ is quite easy to obtain. The $|\Pi|/\varepsilon^2$ guarantee can simply be obtained by iterating over $\pi \in \Pi$, collecting $\widetilde{\mathcal{O}}(1/\varepsilon^2)$ trajectories per policy, and then picking the policy with highest empirical value. The $HSA/\varepsilon^2$ guarantee can be obtained by running known algorithms for tabular RL (Zhang et al., 2021b). Finally, the $A^H/\varepsilon^2$ guarantee is achieved by the classical importance sampling (IS) algorithm (Kearns et al., 1999; Agarwal et al., 2019). Since importance sampling will be an important technique that we repeatedly use and build upon in this paper, we give a formal description of the algorithm below:

---

ImportanceSampling:

- Collect $n = \mathcal{O}(A^H \log|\Pi|/\varepsilon^2)$ trajectories by executing $(a_1, \dots, a_H) \sim \mathrm{Uniform}(\mathcal{A}^H)$.

- Return $\widehat{\pi} = \arg\max_{\pi \in \Pi} \widehat{v}_{\mathrm{IS}}^{\pi}$, where $\widehat{v}_{\mathrm{IS}}^{\pi} := \frac{A^H}{n} \sum_{i=1}^{n} \mathbb{1}\{\pi \rightsquigarrow \tau^{(i)}\}(\sum_{h=1}^{H} r_h^{(i)})$.

---

For every $\pi \in \Pi$, the quantity $\widehat{v}_{\mathrm{IS}}^{\pi}$ is an unbiased estimate of $V^{\pi}$ with variance $A^H$; the sample complexity result follows by standard concentration guarantees (see, e.g., Agarwal et al., 2019).

**Towards Structural Assumptions for Statistically Efficient Agnostic PAC RL.** Of course, No Free Lunch results do not necessarily spell doom—for example, in supervised learning, various structural assumptions have been studied that enable statistically efficient learning. Furthermore, there has been a substantial effort in developing complexity measures like VC dimension, fat-shattering dimension, covering numbers, etc. that characterize agnostic PAC learnability under different scenarios (Shalev-Shwartz and Ben-David, 2014). In this paper, we consider the agnostic reinforcement learning setting, and explore whether there exists a complexity measure that characterizes learnability for every policy class $\Pi$. Formally, can we establish a complexity measure $\mathfrak{C}$ (a function that maps policy classes to real numbers), such that for any $\Pi$, the minimax sample complexity satisfies

$$n_{\mathsf{on}}(\Pi; \varepsilon, \delta) = \widetilde{\Theta}\big(\mathrm{poly}\big(\mathfrak{C}(\Pi), H, \varepsilon^{-1}, \log \delta^{-1}\big)\big),$$

where $\mathfrak{C}(\Pi)$ denotes the complexity of $\Pi$. We mainly focus on finite (but large) policy classes and assume that the $\log|\Pi|$ factors in our upper bounds are mild. In Appendix G, we discuss how our results can be extended to infinite policy classes.

**Is Proposition 1 Tight for Every $\Pi$?** In light of Proposition 1, one obvious candidate is $\overline{\mathfrak{C}}(\Pi) = \min\{A^H, |\Pi|, SA\}$. While $\overline{\mathfrak{C}}(\Pi)$ is definitely sufficient to upper bound the minimax

sample complexity for any policy class $\Pi$ up to log factors, a priori it is not clear if it is also necessary for every policy class $\Pi$. In fact, our next proposition implies that $\overline{\mathfrak{C}}(\Pi)$ is indeed not the right measure of complexity by giving an example of a policy class for which $\overline{\mathfrak{C}}(\Pi) := \min\{A^H, |\Pi|, SA\}$ is exponentially larger than the minimax sample complexity for agnostic learning for that policy class, even when $\varepsilon$ is constant.

**Proposition 2.** *Let $H \in \mathbb{N}$, $K \in \mathbb{N}$, $\mathcal{S}_h = \{s_{(i,h)} : i \in [K]\}$ for all $h \in [H]$, and $\mathcal{A} = \{0, 1\}$. Consider the singleton policy class: $\Pi_{\mathrm{sing}} := \{\pi_{(i',h')} : i' \in [K], h' \in [H]\}$, where $\pi_{(i',h')}$ takes the action $1$ on state $s_{(i',h')}$, and $0$ everywhere else. Then $\min\{A^H, |\Pi_{\mathrm{sing}}|, SA\} = 2^H$ but $n_{\mathrm{on}}(\Pi_{\mathrm{sing}}; \varepsilon, \delta) \leq \widetilde{\mathcal{O}}(H^3 \cdot \log(1/\delta)/\varepsilon^2)$.*

The above upper bound on minimax sample complexity holds arbitrarily large values of $K$, and can be obtained as a corollary of our more general upper bound in Section 6. The key intuition for why $\Pi_{\mathrm{sing}}$ can be learned in $\mathrm{poly}(H)$ samples is that even though the policy class and the state space are large when $K$ is large, the set of possible trajectories obtained by running any $\pi \in \Pi_{\mathrm{sing}}$ has low complexity. In particular, every trajectory $\tau$ has at most one $a_h = 1$. This observation enables us to employ the straightforward modification of the classical IS algorithm: draw $\mathrm{poly}(H) \cdot \log(1/\delta)/\varepsilon^2$ samples from the uniform distribution over $\Pi_{\mathrm{core}} = \{\pi_h : h \in [H]\}$ where the policy $\pi_h$ takes the action 1 on every state at layer $h$ and 0 everywhere else. The variance of the resulting estimator $\widehat{v}_{\mathrm{IS}}^\pi$ is $1/H$, so the sample complexity of this modified variant of IS has only $\mathrm{poly}(H)$ dependence by standard concentration bounds.

In the sequel, we present a new complexity measure that formalizes this intuition that a policy class $\Pi$ is efficiently learnable if the set of trajectories induced by policies in $\Pi$ is small.

# 3 Spanning Capacity

The spanning capacity precisely captures the intuition that trajectories obtained by running any $\pi \in \Pi$ have "low complexity." We first define a notion of reachability: in deterministic MDP $M \in \mathcal{M}^{\mathrm{det}}$, we say $(s, a)$ is *reachable* by $\pi \in \Pi$ if $(s, a)$ lies on the trajectory obtained by running $\pi$ on $M$. Roughly speaking, the spanning capacity measures "complexity" of $\Pi$ as the maximum number of state-action pairs which are reachable by some $\pi \in \Pi$ in any *deterministic* MDP.

**Definition 1** (spanning capacity)**.** *Fix a deterministic MDP $M \in \mathcal{M}^{\mathrm{det}}$. We define the* cumulative reachability *at layer $h \in [H]$, denoted by $C_h^{\mathrm{reach}}(\Pi; M)$, as*

$$C_h^{\mathrm{reach}}(\Pi; M) := |\{(s, a) : (s, a) \text{ is reachable by } \Pi \text{ at layer } h\}|.$$

*We define the* spanning capacity *of $\Pi$ as*

$$\mathfrak{C}(\Pi) := \max_{h \in [H]} \max_{M \in \mathcal{M}^{\mathrm{det}}} C_h^{\mathrm{reach}}(\Pi; M).$$

To build intuition, we first look at some simple examples with small spanning capacity:

- **Contextual Bandits:** Consider the standard formulation of contextual bandits (i.e., RL with $H = 1$). For any policy class $\Pi_{\mathrm{cb}}$, since $H = 1$, the largest deterministic MDP we can construct has a single state $s_1$ and at most $A$ actions available on $s_1$, so $\mathfrak{C}(\Pi_{\mathrm{cb}}) \leq A$.

- **Tabular MDPs:** Consider tabular RL with the policy class $\Pi_{\mathrm{tab}} = \mathcal{A}^{\mathcal{S}}$ consisting of all deterministic policies on the underlying state space. Depending on the relationship between $S$, $A$ and $H$, we have two possible bounds on $\mathfrak{C}(\Pi_{\mathrm{tab}}) \leq \min\{A^H, SA\}$. If the state space is exponentially large in $H$, then it is possible to construct a full $A$-ary "tree" such that every $(s, a)$ pair at layer $H$ is visited, giving us the $A^H$ bound. However, if the state space is small, then the number of $(s, a)$ pairs available at any layer $H$ is trivially bounded by $SA$.

- **Bounded Cardinality Policy Classes:** For any policy class $\Pi$, we always have that $\mathfrak{C}(\Pi) \leq |\Pi|$, since in any deterministic MDP, in any layer $h \in [H]$, each $\pi \in \Pi$ can visit at most one new $(s, a)$ pair. Thus, for policy classes $|\Pi_{\mathrm{small}}|$ with small cardinality (e.g. $|\Pi_{\mathrm{small}}| = O(\mathrm{poly}(H, A))$), the spanning capacity is also small; Note that in this case, we allow our sample complexity bounds to depend on $|\Pi_{\mathrm{small}}|$.

- **Singletons:** For the singleton class we have $\mathfrak{C}(\Pi_{\mathrm{sing}}) = H + 1$, since once we fix a deterministic MDP, there are at most $H$ states where we can split from the trajectory taken by the policy which

always plays $a = 0$, so the maximum number of $(s, a)$ pairs reachable at layer $h \in [H]$ is $h + 1$. Observe that in light of Proposition 2, the spanning capacity for $\Pi_{\text{sing}}$ is "on the right order" for characterizing the minimax sample complexity for agnostic PAC RL.

Before proceeding, we note that for any policy class $\Pi$, the spanning capacity is always bounded.

**Proposition 3.** *For any policy class $\Pi$, we have $\mathfrak{C}(\Pi) \leq \min\{A^H, |\Pi|, SA\}$.*

Proposition 3 recovers the worst-case upper and lower bound from Section 2.2. However, for many policy classes, spanning capacity is substantially smaller than upper bound of Proposition 3. In addition to the examples we provided above, we list several additional policy classes with small spanning capacity. For these policy classes we set the state/action spaces to be $\mathcal{S}_h = \{s_{(i,h)} : i \in [K]\}$ for all $h \in [H]$ and $\mathcal{A} = \{0, 1\}$, respectively. All proofs are deferred to Appendix B.

- **$\ell$-tons**: A natural generalization of singletons. We define $\Pi_{\ell-\text{ton}} := \{\pi_I : I \subset \mathcal{S}, |I| \leq \ell\}$, where the policy $\pi_I$ is defined s.t. $\pi_I(s) = \mathbb{1}\{s \in I\}$ for any $s \in \mathcal{S}$. Here, $\mathfrak{C}(\Pi_{\ell-\text{ton}}) = \Theta(H^\ell)$.

- **1-Active Policies**: We define $\Pi_{1-\text{act}}$ to be the class of policies which can take both possible actions on a single state $s_{(1,h)}$ in each layer $h$, but on other states $s_{(i,h)}$ for $i \neq 1$ must take action 0. Formally, $\Pi_{1-\text{act}} := \{\pi_b \mid b \in \{0, 1\}^H\}$, where for any $b \in \{0, 1\}^H$ the policy $\pi_b$ is defined such that $\pi_b(s) = b[h]$ if $s = s_{(1,h)}$, and $\pi_b(s) = 0$ otherwise.

- **All-Active Policies**: We define $\Pi_{j-\text{act}} := \{\pi_b \mid b \in \{0, 1\}^H\}$, where for any $b \in \{0, 1\}^H$ the policy $\pi_b$ is defined such that $\pi_b(s) = b[h]$ if $s = s_{(j,h)}$, and $\pi_b(s) = 0$ otherwise. We let $\Pi_{\text{act}} := \bigcup_{j=1}^K \Pi_{j-\text{act}}$. Here, $\mathfrak{C}(\Pi_{\text{act}}) = \Theta(H^2)$.

A natural interpretation of the spanning capacity is that it represents the largest "needle in a haystack" that can be embedded in a deterministic MDP using the policy class $\Pi$. To see this, let $(M^\star, h^\star)$ be the MDP and layer which witnesses $\mathfrak{C}(\Pi)$, and let $\{(s_i, a_i)\}_{i=1}^{\mathfrak{C}(\Pi)}$ be the set of state-action pairs reachable by $\Pi$ in $M^\star$ at layer $h^\star$. Then one can hide a reward of 1 on one of these state-action pairs; since every trajectory visits a single $(s_i, a_i)$ at layer $h^\star$, we need at least $\mathfrak{C}(\Pi)$ samples in order to discover which state-action pair has the hidden reward. Note that in this agnostic learning setup, we only need to care about the states that are reachable using $\Pi$, even though the $h^\star$ layer may have other non-reachable states and actions.

### 3.1 Connection to Coverability

The spanning capacity has another interpretation as the worst-case *coverability*, a structural parameter defined in a recent work by Xie et al. (2022).

**Definition 2** (Coverability, Xie et al. (2022)). *For any MDP $M$ and policy class $\Pi$, the coverability coefficient $C^{\text{cov}}$ is denoted*

$$C^{\text{cov}}(\Pi; M) := \inf_{\mu_1, \ldots \mu_H \in \Delta(\mathcal{S} \times \mathcal{A})} \sup_{\pi \in \Pi, h \in [H]} \left\| \frac{d_h^\pi}{\mu_h} \right\|_\infty = \max_{h \in [H]} \sum_{s,a} \sup_{\pi \in \Pi} d_h^\pi(s, a). \quad (2)$$

The last equality is shown in Lemma 3 of Xie et al. (2022), and it says that the coverability coefficient is equivalent to a notion of cumulative reachability (one can check that their definition coincides with ours for deterministic MDPs).

Coverage conditions date back to the analysis of the classic Fitted Q-Iteration (FQI) algorithm (Munos, 2007; Munos and Szepesvári, 2008), and have extensively been studied in offline RL. Various models like tabular MDPs, linear MDPs, low-rank MDPs, and exogenous MDPs satisfy the above coverage condition (Antos et al., 2008; Chen and Jiang, 2019; Jin et al., 2021b; Rashidinejad et al., 2021; Zhan et al., 2022; Xie et al., 2022), and recently, Xie et al. showed that *coverability* can be used to prove regret guarantees for online RL, albeit under the additional assumption of value function realizability.

It is straightforward from Definition 2 that our notion of spanning capacity is worst-case coverability when we maximize over deterministic MDPs, since for any deterministic MDP, $\sup_{\pi \in \Pi} d_h^\pi(s, a) = \mathbb{1}\{(s, a)$ is reachable by $\Pi$ at layer $h\}$. The next lemma shows that our notion of spanning capacity is *exactly* worst-case coverability even when we maximize over the larger class of stochastic MDPs. As a consequence, there always exists a deterministic MDP that witnesses worst-case coverability.

**Lemma 1.** *For any policy class $\Pi$, we have $\sup_{M \in \mathcal{M}^{\text{sto}}} C^{\text{cov}}(\Pi; M) = \mathfrak{C}(\Pi)$.*

While spanning capacity is equal to the worst-case coverability, we remark that the two definitions have different origins. The notion of coverability bridges offline and online RL, and was introduced in Xie et al. (2022) to characterize when sample efficient learning is possible in value-based RL, where the learner has access to a realizable value function class. On the other hand, spanning capacity is developed for the much weaker agnostic RL setting, where the learner only has access to a policy class (and does not have access to a realizable value function class). Note that a realizable value function class can be used to construct a policy class that contains the optimal policy, but the converse is not true. Furthermore, note that the above equivalence only holds in a worst-case sense (over MDPs). In fact, as we show in Appendix C, coverability alone is not sufficient for sample efficient agnostic PAC RL in the online interaction model.

## 4 Generative Model: Spanning Capacity is Necessary and Sufficient

In this section, we show that for any policy class, the spanning capacity characterizes the minimax sample complexity for agnostic PAC RL under generative model.

**Theorem 1** (Upper Bound for Generative Model). *For any $\Pi$, the minimax sample complexity $(\varepsilon, \delta)$-PAC learning $\Pi$ is at most $n_{\mathsf{gen}}(\Pi; \varepsilon, \delta) \leq \mathcal{O}\Big( \frac{H \cdot \mathfrak{C}(\Pi)}{\varepsilon^2} \cdot \log \frac{|\Pi|}{\delta} \Big)$.*

The proof can be found in Appendix D.1, and is a straightforward modification of the classic *trajectory tree method* from Kearns et al. (1999): using generative access, sample $\mathcal{O}(\log|\Pi|/\varepsilon^2)$ deterministic trajectory trees from the MDP to get unbiased evaluations for every $\pi \in \Pi$; the number of generative queries made is bounded since the size of the maximum deterministic tree is at most $H \cdot \mathfrak{C}(\Pi)$.

**Theorem 2** (Lower Bound for Generative Model). *For any $\Pi$, the minimax sample complexity $(\varepsilon, \delta)$-PAC learning $\Pi$ is at least $n_{\mathsf{gen}}(\Pi; \varepsilon, \delta) \geq \Omega\Big( \frac{\mathfrak{C}(\Pi)}{\varepsilon^2} \cdot \log \frac{1}{\delta} \Big)$.*

The proof can be found in Appendix D.2. Intuitively, given an MDP $M^\star$ which witnesses $\mathfrak{C}(\Pi)$, one can embed a bandit instance on the relevant $(s, a)$ pairs spanned by $\Pi$ in $M^\star$. The lower bound follows by a reduction to the lower bound for $(\varepsilon, \delta)$-PAC learning in multi-armed bandits.

Together, Theorem 1 and Theorem 2 paint a relatively complete picture for the minimax sample complexity of learning any policy class $\Pi$, in the generative model, up to an $H \cdot \log|\Pi|$ factor.

**Deterministic MDPs.** A similar guarantee holds for online RL over deterministic MDPs.

**Corollary 1.** *Over the class of MDPs with deterministic transitions, the minimax sample complexity of $(\varepsilon, \delta)$-PAC learning any $\Pi$ is*

$$\Omega\Big( \frac{\mathfrak{C}(\Pi)}{\varepsilon^2} \cdot \log \frac{1}{\delta} \Big) \leq n_{\mathsf{on}}(\Pi, \mathcal{M}^{\mathrm{detP}}; \varepsilon, \delta) \leq \mathcal{O}\Big( \frac{H \cdot \mathfrak{C}(\Pi)}{\varepsilon^2} \cdot \log \frac{|\Pi|}{\delta} \Big).$$

The upper bound follows because the trajectory tree algorithm for deterministic transitions samples the same tree over and over again (with different stochastic rewards). The lower bound trivially extends because the lower bound of Theorem 2 actually uses an MDP $M \in \mathcal{M}^{\mathrm{detP}}$ (in fact, the transitions of $M$ are also known to the learner).

## 5 Online RL: Spanning Capacity is Not Sufficient

Given that fact that spanning capacity characterizes the minimax sample complexity of agnostic PAC RL in the generative model, one might be tempted to conjecture that spanning capacity is also the right characterization in online RL. The lower bound is clear since online RL is at least as hard as learning with a generative model, so Theorem 2 already shows that spanning capacity is *necessary*. But is it also sufficient?

In this section, we prove a surprising negative result showing that bounded spanning capacity by itself is not sufficient to characterize the minimax sample complexity in online RL. In particular, we provide an example for which we have a *superpolynomial* (in $H$) lower bound on the number of trajectories needed for learning in online RL, that is not captured by any polynomial function of spanning capacity. This implies that, contrary to RL with a generative model, one can not hope for $n_{\mathsf{on}}(\Pi; \varepsilon, \delta) = \widetilde{\Theta}(\mathrm{poly}(\mathfrak{C}(\Pi), H, \varepsilon^{-1}, \log \delta^{-1}))$ in online RL.

**Theorem 3** (Lower Bound for Online RL). *Fix any sufficiently large $H$. Let $\varepsilon \in (1/2^{\mathcal{O}(H)}, \mathcal{O}(1/H))$ and $\ell \in \{2, \dots, H\}$ such that $1/\varepsilon^\ell \leq 2^H$. There exists a policy class $\Pi^{(\ell)}$ of size $\mathcal{O}(1/\varepsilon^\ell)$ with $\mathfrak{C}(\Pi^{(\ell)}) \leq \mathcal{O}(H^{4\ell+2})$ and a family of MDPs $\mathcal{M}$ with state space $\mathcal{S}$ of size $2^{\mathcal{O}(H)}$, binary action space, and horizon $H$ such that: for any $(\varepsilon, 1/8)$-PAC algorithm, there exists an MDP $M \in \mathcal{M}$ for which the algorithm must collect at least $\Omega(\min\{\frac{1}{\varepsilon^\ell}, 2^{H/3}\})$ online trajectories in expectation.*

Informally speaking, the lower bound shows that there exists a policy class $\Pi$ for which $n_{\mathsf{on}}(\Pi; \varepsilon, \delta) = 1/\varepsilon^{\Omega(\log_H \mathfrak{C}(\Pi))}$. In order to interpret this theorem, we can instantiate choices of $\varepsilon = 1/2^{\sqrt{H}}$ and $\ell = \sqrt{H}$ to show an explicit separation.

**Corollary 2.** *For any sufficiently large $H$, there exists a policy class $\Pi$ with $\mathfrak{C}(\Pi) = 2^{\mathcal{O}(\sqrt{H} \log H)}$ such that for any $(1/2^{\sqrt{H}}, 1/8)$-PAC algorithm, there exists an MDP for which the algorithm must collect at least $2^{\Omega(H)}$ online trajectories in expectation.*

In conjunction with the results of Section 4, Theorem 3 shows that (1) online RL is *strictly harder* than RL with generative access, and (2) online RL for stochastic MDPs is *strictly harder* than online RL for MDPs with deterministic transitions. We defer the proof of Theorem 3 to Appendix E. Our lower bound introduces several technical novelties: the family $\mathcal{M}$ utilizes a *contextual* variant of the combination lock, and the policy class $\Pi$ is constructed via a careful probabilistic argument such that it is hard to explore despite having small spanning capacity.

## 6 Statistically Efficient Agnostic Learning in Online RL

The lower bound in Theorem 3 suggests that further structural assumptions on $\Pi$ are needed for statistically efficient agnostic RL under the online interaction model. Essentially, the lower bound example provided in Theorem 3 is hard to agnostically learn because any two distinct policies $\pi, \pi' \in \Pi$ can differ substantially on a large subset of states (of size at least $\varepsilon \cdot 2^{2H}$). Thus, we cannot hope to learn "in parallel" via a low variance IS strategy that utilizes extrapolation to evaluate all policies $\pi \in \Pi$, as we did for singletons.

In the sequel, we consider the following sunflower property to rule out such problematic scenarios, and show how bounded spanning capacity along with the sunflower property enable sample-efficient agnostic RL in the online interaction model. The sunflower property only depends on the state space, action space, and policy class, and is independent of the transition dynamics and rewards of the underlying MDP. We first define a petal, a key ingredient of a sunflower.

**Definition 3** (Petal). *For a policy set $\bar{\Pi}$, and states $\bar{\mathcal{S}} \subseteq \mathcal{S}$, a policy $\pi$ is said to be a $\bar{\mathcal{S}}$-petal on $\bar{\Pi}$ if for all $h \leq h' \leq H$, and partial trajectories $\tau = (s_h, a_h, \cdots, s_{h'}, a_{h'})$ that are consistent with $\pi$: either $\tau$ is also consistent with some $\pi' \in \bar{\Pi}$, or there exists $i \in (h, h']$ s.t. $s_i \in \bar{\mathcal{S}}$.*

Informally, $\pi$ is a $\bar{\mathcal{S}}$-petal on $\bar{\Pi}$ if any trajectory that can be obtained using $\pi$ can either also be obtained using a policy in $\bar{\Pi}$ or must pass through $\bar{\mathcal{S}}$. Thus, any policy that is a $\bar{\mathcal{S}}$-petal on $\bar{\Pi}$ can only differentiate from $\bar{\Pi}$ in a structured way. A policy class is said to be a sunflower if it is a union of petals as defined below:

**Definition 4** (Sunflower). *A policy class $\Pi$ is said to be a $(K, D)$-sunflower if there exists a set $\Pi_{\mathrm{core}}$ of Markovian policies with $|\Pi_{\mathrm{core}}| \leq K$ such that for every policy $\pi \in \Pi$ there exists a set $\mathcal{S}_\pi \subseteq \mathcal{S}$, of size at most $D$, so that $\pi$ is an $S_\pi$-petal on $\Pi_{\mathrm{core}}$.*

Our next theorem provides a sample complexity bound for Agnostic PAC RL for policy classes that have $(K, D)$-sunflower structure. This bound is obtained via a new exploration algorithm called POPLER that takes input the set $\Pi_{\mathrm{core}}$ and corresponding petals $\{\mathcal{S}_\pi\}_{\pi \in \Pi}$ and leverages importance sampling as well as reachable state identification techniques to simultaneously estimate the value of every policy in $\Pi$.

**Theorem 4.** *Let $\varepsilon, \delta > 0$. Suppose the policy class $\Pi$ satisfies Definition 1 with spanning capacity $\mathfrak{C}(\Pi)$, and is a $(K, D)$-sunflower. Then, for any MDP $M$, with probability at least $1 - \delta$, POPLER (Algorithm 1) succeeds in returning a policy $\widehat{\pi}$ that satisfies $V^{\widehat{\pi}} \geq \max_{\pi \in \Pi} V^\pi - \varepsilon$, after collecting*

$$\widetilde{\mathcal{O}}\left(\left(\tfrac{1}{\varepsilon^2} + \tfrac{HD^6 \mathfrak{C}(\Pi)}{\varepsilon^4}\right) \cdot K^2 \log \tfrac{|\Pi|}{\delta}\right) \quad \text{online trajectories in } M.$$

The proof of Theorem 4, and the corresponding hyperparameters in POPLER needed to obtain the above bound, can be found in Appendix F. Before diving into the algorithm and proof details, let us highlight several key aspects of the above sample complexity bound:

- Note that a class $\Pi$ may be a $(K, D)$-sunflower for many different choices of $K$ and $D$. Barring computational issues, one can enumerate over all choices of $\Pi_{\text{core}}$ and $\{\mathcal{S}_\pi\}_{\pi \in \Pi}$, and check if $\Pi$ is a $(K, D)$-sunflower for that choice of $K = |\Pi_{\text{core}}|$ and $D = \max_{\pi \in \Pi} |\mathcal{S}_\pi|$. Since our bound in Theorem 4 scales with $K$ and $D$, we are free to choose $K$ and $D$ to minimize the corresponding sample complexity bound.

- In order to get a polynomial sample complexity in Theorem 4, both $\mathfrak{C}(\Pi)$ and $(K, D)$ are required to be $\text{poly}(H, \log|\Pi|)$. All of the policy classes considered in Section 3 have the sunflower property, with both $K, D = \text{poly}(H)$, and thus our sample complexity bound extends for all these classes. See Appendix B for details.

- Notice that for Theorem 4 to hold, we need both bounded spanning capacity as well as the sunflower structure on the policy class with bounded $(K, D)$. Thus, one may wonder if we can obtain a similar polynomial sample complexity guarantee in online RL under weaker assumptions. In Theorem 3, we already showed that bounded $\mathfrak{C}(\Pi)$ alone is not sufficient to obtain polynomial sample complexity in online RL. Likewise, as we show in Appendix F, sunflower property with bounded $(K, D)$ alone is also not sufficient for polynomial sample complexity, and hence both assumptions cannot be individually removed. However, it is an interesting question if there is some other structural assumption that combines both spanning capacity and the sunflower property, and is both sufficient and necessary for agnostic PAC learning in online RL. See Section 7 for further discussions on this.

**Why Does the Sunflower Property Enable Sample-Efficient Learning?**  Intuitively, the sunflower property captures the intuition of simultaneous estimation of all policies $\pi \in \Pi$ via importance sampling (IS), and allows control of both bias and variance. Let $\pi$ be a $\mathcal{S}_\pi$-petal on $\Pi_{\text{core}}$. Any trajectory $\tau \rightsquigarrow \pi$ that avoids $\mathcal{S}_\pi$ must be consistent with some policy in $\Pi_{\text{core}}$, and will thus be covered by the data collected using $\pi' \sim \text{Uniform}(\Pi_{\text{core}})$. Thus, using IS with variance scaling with $K$, one can create a biased estimator for $V^\pi$, where the bias is *only due* to trajectories that pass through $\mathcal{S}_\pi$. There are two cases. If every state in $\mathcal{S}_\pi$ has small reachability under $\pi$, i.e. $d^\pi(s) \ll \varepsilon$ for every $s \in \mathcal{S}_\pi$, then the IS estimate will have a low bias (linear in $|\mathcal{S}_\pi|$), so we can compute $V^\pi$ up to error at most $\varepsilon|\mathcal{S}_\pi|$. On the other hand, if $d^\pi(s)$ is large for some $s \in \mathcal{S}_\pi$, it is possible to explicitly control the bias that arises from trajectories passing through them since there are at most $D$ of them.

## 6.1 Algorithm and Proof Ideas

POPLER (Algorithm 1) takes as input a policy class $\Pi$, as well as sets $\Pi_{\text{core}}$ and $\{\mathcal{S}_\pi\}_{\pi \in \Pi}$, which can be computed beforehand by enumeration. POPLER has two phases: a *state identification phase*, where it finds "petal" states $s \in \bigcup_{\pi \in \Pi} \mathcal{S}_\pi$ that are reachable with sufficiently large probability; and an *evaluation phase* where it computes estimates $\widehat{V}^\pi$ for every $\pi \in \Pi$. It uses three subroutines DataCollector, EstReachability, and Evaluate, whose pseudocodes are stated in Appendix F.2.

The structure of the algorithm is reminiscent of reward-free exploration algorithms in tabular RL, e.g. Jin et al. (2020), where we first identify (petal) states that are reachable with probability at least $\Omega(\varepsilon/D)$ and build a policy cover for these states, and then use dynamic programming to estimate the values. However, contrary to the classical tabular RL setting, because the state space can be large, our setting is much more challenging and necessitates technical innovations. In particular, we can no longer enumerate over all petal states and check if they are sufficiently reachable by some policy $\pi \in \Pi$ (since the total number of petal states $\sum_{\pi \in \Pi} |\mathcal{S}_\pi|$ could scale linearly in $|\Pi|$, a factor that we do not want to appear in our sample complexity). Instead, the key observation that we rely on is that if spanning capacity is bounded, then by the equivalence of spanning capacity and worst-case coverability (Lemma 1) and due to (2), the number of highly reachable (and thus relevant) petal states is also bounded. Thus, we only need to build a policy cover to reach these relevant petal states. Our algorithm does this in a sample-efficient *sequential* manner. For both state identification as well as evaluation, we interleave importance sampling estimates with the construction of a *policy-specific* Markov Reward Processes (MRPs), which are defined for every $\pi \in \Pi$. The challenge is doing all of this "in parallel" for every $\pi \in \Pi$ through extensive sample reuse to avoid a blowup of $|\Pi|$ or $S$ in the sample complexity. For the exact construction of these policy-specific MRPs and how they are used in the algorithm, please see Appendix F.2.

**Algorithm 1** **P**olicy **OP**timization by **L**earning $\varepsilon$-**R**eachable States (POPLER)

---

**Require:** Policy class $\Pi$, Sets $\Pi_{\mathrm{core}}$ and $\{\mathcal{S}_\pi\}_{\pi \in \Pi}$, Parameters $K, D, n_1, n_2, \varepsilon, \delta$.

1: Define an additional start state $s_\top$ (at $h = 0$) and end state $s_\bot$ (at $h = H + 1$).
2: Initialize $\mathcal{S}^{\mathrm{rch}} = \{s_\top\}$, $\mathcal{T} \leftarrow \{(s_\top, \mathrm{Null})\}$, and for every $\pi \in \Pi$, define $\mathcal{S}_\pi^+ := \mathcal{S}_\pi \cup \{s_\top, s_\bot\}$.
3: $\mathcal{D}_\top \leftarrow \mathsf{DataCollector}(s_\top, \mathrm{Null}, \Pi_{\mathrm{core}}, n_1)$
4: /* Identification of Petal States that are Reachable with $\Omega(\varepsilon/D)$ Probability */
5: **while** Terminate = False **do**
6:     Set Terminate = True.
7:     **for** $\pi \in \Pi$ **do**
8:         Compute the set of already explored reachable states $\mathcal{S}_\pi^{\mathrm{rch}} = \mathcal{S}_\pi^+ \cap \mathcal{S}^{\mathrm{rch}}$, and the remaining states $\mathcal{S}_\pi^{\mathrm{rem}} = \mathcal{S}_\pi \setminus (\mathcal{S}_\pi^{\mathrm{rch}} \cup \{s_\bot\})$.
9:         Estimate the policy-specific MRP $\widehat{\mathfrak{M}}_{\mathcal{S}^{\mathrm{rch}}}^\pi$ according to (14) and (15).
10:         **for** $\bar{s} \in \mathcal{S}_\pi^{\mathrm{rem}}$ **do**
11:             Estimate probability of reaching $\bar{s}$ under $\pi$ as $\widehat{d}^\pi(\bar{s}) \leftarrow$ $\mathsf{EstReachability}(\mathcal{S}_\pi^+, \widehat{\mathfrak{M}}_{\mathcal{S}^{\mathrm{rch}}}^\pi, \bar{s})$.
12:             **if** $\widehat{d}^\pi(\bar{s}) \geq \varepsilon/6D$ **then**
13:                 Update $\mathcal{S}^{\mathrm{rch}} \leftarrow \mathcal{S}^{\mathrm{rch}} \cup \{\bar{s}\}$, $\mathcal{T} \leftarrow \mathcal{T} \cup \{(\bar{s}, \pi)\}$ and set Terminate = False.
14:                 Collect dataset $\mathcal{D}_{\bar{s}} \leftarrow \mathsf{DataCollector}(\bar{s}, \pi, \Pi_{\mathrm{core}}, n_2)$.
15:             **end if**
16:         **end for**
17:     **end for**
18: **end while**
19: /* Policy Evaluation and Optimization */
20: **for** $\pi \in \Pi$ **do**
21:     $\widehat{V}^\pi \leftarrow \mathsf{Evaluate}(\Pi_{\mathrm{core}}, \mathcal{S}^{\mathrm{rch}}, \{\mathcal{D}_s\}_{s \in \mathcal{S}^{\mathrm{rch}}}, \pi)$.
22: **end for**
23: **Return** $\widehat{\pi} \in \arg\max_\pi \widehat{V}^\pi$.

---

# 7 Conclusion and Discussion

In this paper, we investigated when agnostic RL is statistically tractable in large state and action spaces, and introduced spanning capacity as a natural measure of complexity that only depends on the policy class, and is independent of the MDP rewards and transitions. We first showed that bounded spanning capacity is both necessary and sufficient for agnostic PAC RL under the generative access model. However, we also provided a negative result showing that bounded spanning capacity does not suffice for online RL, thus showing a surprising separation between agnostic RL with a generative model and online interaction. We then provided an additional structural assumption, called the sunflower property, that allows for statistically efficient learning in online RL. Our sample complexity bound for online RL is obtained using a novel exploration algorithm called POPLER that relies on certain policy-specific Markov Reward Processes to guide exploration, and takes inspiration from the classical importance sampling method and reward-free exploration algorithms for tabular MDPs.

Our results pave the way for several future lines of inquiry. The most important direction is exploring complexity measures to tightly characterize the minimax sample complexity for online RL. On the upper bound side, Theorem 4 shows that bounded spanning capacity along with an additional sunflower property is sufficient for online RL. On the lower bound side, while bounded spanning capacity is necessary (by Theorem 2), we do not know if the sunflower property is also necessary. Another direction is understanding if one can improve the statistical complexity of policy-based learning using stronger feedback models; in Appendix I we explore whether receiving expert feedback in the form of the optimal value function $\{Q^\star(s, a)\}_{a \in \mathcal{A}}$ on the visited states can help. Surprisingly, the answer depends on realizability of the optimal policy $\pi^\star \in \Pi$: when $\pi^\star \in \Pi$, $Q^\star$ feedback can be utilized to achieve an $O(\mathrm{poly}(\log|\Pi|, H, 1/\varepsilon))$ sample complexity bound, with no dependence on $\mathfrak{C}(\Pi)$; if $\pi^\star \notin \Pi$, then one can not hope to learn with less than $\Omega(\mathfrak{C}(\Pi))$ samples in the worst case. Understanding the role of $\pi^\star$-realizability and exploring the benefits of other feedback models in agnostic RL are interesting future research directions. Other research directions include sharpening the sample complexity bound in Theorem 4, extending POPLER for regret minimization, and developing computationally efficient algorithms.

## Acknowledgments and Disclosure of Funding

We thank Dylan Foster, Pritish Kamath, Jason D. Lee, Wen Sun, and Cong Ma for helpful discussions. GL and NS are partially supported by the NSF and the Simons Foundation. Part of this work was completed while GL was visiting Princeton University. AR acknowledges support from the ONR through award N00014-20-1-2336, ARO through award W911NF-21-1-0328, and from the DOE through award DE-SC0022199.

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

# Contents of Appendix

# A  Detailed Comparison to Related Works

Reinforcement Learning (RL) has seen substantial progress over the past few years, with several different directions of work being pursued for efficiently solving RL problems that occur in practice. The classical approach to solving an RL problem is to model it as a tabular MDP. A long line of work (Sutton and Barto, 2018; Agarwal et al., 2019; Kearns and Singh, 2002; Brafman and Tennenholtz, 2002; Auer et al., 2008; Azar et al., 2017; Gheshlaghi Azar et al., 2013; Jin et al., 2018) has studied provably sample-efficient learning algorithms that can find the optimal policy in tabular RL. Unfortunately, the sample complexity of such algorithms unavoidably scales with the size of the state/action spaces, so they fail to be efficient in practical RL problems with large state/action spaces. In order to develop algorithms for the more practical large state/action RL settings, various assumptions have been considered in the prior works. In the following, we provide a detailed comparison of our setup and assumptions with the existing literature.

**RL with Function Approximation.**  A popular paradigm for developing algorithms for MDPs with large state/action spaces is to use function approximation to either model the MDP dynamics or optimal value functions. Over the last decade, there has been a long line of work (Jiang et al., 2017; Dann et al., 2018; Sun et al., 2019; Du et al., 2019a; Wang et al., 2020c; Du et al., 2021; Foster et al., 2021a; Jin et al., 2021a; Zhong et al., 2022; Foster et al., 2023) in understanding structural conditions on the function class and the underlying MDP that enable statistically efficient RL. However, all of these works rely crucially on the realizability assumption, namely that the true model / value function belong to the chosen class. Unfortunately, such an assumption is too strong to hold in practice. Furthermore, the prior works using function approximation make additional assumptions like Bellman Completeness that are difficult to verify for the underlying task.

In our work, we study the problem of agnostic RL to sidestep these challenges. In particular, instead of modeling the value/dynamics, the learner now models "good policies" for the underlying task, and the learning objective is to find a policy that can perform as well as the best in the chosen policy class. We note that while a realizable value class/dynamics class $\mathcal{F}$ can be converted into a realizable policy class $\Pi_{\mathcal{F}}$ by choosing the greedy policies for each value function/dynamics, the converse is not true. Thus, our agnostic RL objective relies on a strictly weaker modeling assumption.

**Connections to Decision-Estimation Coefficient (DEC).**  The seminal work of Foster et al. (2021a) provides a unified complexity measure called Decision-Estimation Coefficient (DEC) that characterizes the complexity of model-based RL. Given the generality of the E2D algorithm of Foster et al. (2021a), one may be wondering if our results can be recovered using their framework via a model-based approach. In particular, can we recover the sample complexity bound in Theorems 1 or 4 by considering the model class $\mathcal{M}^{\text{det}}$ or $\mathcal{M}^{\text{sto}}$ along with the decision set $\Pi$. To the best of our knowledge, the framework of Foster et al. (2021a) do not directly recover our results, however, the complexity measures are closely related. Note that one can upper-bound the DEC by the coverability coefficient Xie et al. (2022); furthermore we show that $\mathfrak{C}(\Pi)$ is worst-case coverability (Lemma 1) so it follows that DEC is upper-bounded by $\mathfrak{C}(\Pi)$. However, the algorithm in Foster et al. (2021a) achieves regret bounds which scale with $\text{DEC} \cdot \log|\mathcal{M}^{\text{sto}}|$, which can be vacuous in the large state-space setting since $\log|\mathcal{M}^{\text{sto}}| \propto |\mathcal{S}|$; In contrast, our upper bounds have no explicit dependence on $|\mathcal{S}|$.

**RL with Rich Observations.**  Various settings have been studied where the dynamics are determined by a simple latent state space, but instead of observing the latent states directly, the learner receives rich observations corresponding to the underlying latent states. These include the Block MDP (Krishnamurthy et al., 2016; Du et al., 2019a; Misra et al., 2020; Mhammedi et al., 2023), Low-Rank MDPs (Uehara et al., 2021; Huang et al., 2023), Exogenous MDPs (Efroni et al., 2021; Xie et al., 2022; Efroni et al., 2022), etc. However, all of these prior works assume that the learner is given a realizable decoder class (consisting of functions that map observations to latent states) that contains the true decoder for the underlying MDP. Additionally, they require strong assumptions on the underlying latent state space dynamics, e.g. it is tabular or low-rank, in order to make learning tractable. Thus, their guarantees are not agnostic. In fact, given a realizable decoder class and additional structure on the latent state dynamics, one can construct a policy class that contains the optimal policy for the MDP, but the converse is not true. Thus, our agnostic RL setting is strictly more general.

**Relation to Exponential Lower Bounds for RL with Function Approximation.** Recently, many statistical lower bounds have been developed in RL with function approximation under only realizability. A line of work including (Wang et al., 2020b; Zanette, 2021; Weisz et al., 2021; Foster et al., 2021b) showed that the sample complexity scales exponentially in the horizon $H$ for learning the optimal policy for RL problems where only the optimal value function $Q^\star$ is linear w.r.t. the given features. Similarly, Du et al. (2019b) showed that one may need exponentially in $H$ even if the optimal policy is linear w.r.t. the true features. These lower bounds can be extended to our agnostic RL setting, giving similar exponential in $H$ lower bounds for agnostic RL, thus supplementing the well-known lower bounds (Krishnamurthy et al., 2016) which show that agnostic RL is intractable without additional structural assumptions on the policy class. However, to recall, the focus of this paper is to propose assumptions, like Definition 1 or 4, that circumvent these lower bounds and allow for sample efficient agnostic RL.

**Importance Sampling for RL.** Various importance sampling based estimators (Xie et al., 2019; Jiang and Li, 2016; Gottesman et al., 2019; Yin and Wang, 2020; Thomas and Brunskill, 2016; Nachum et al., 2019) have been developed in RL literature to provide reliable off-policy evaluation in offline RL. However, these methods also require realizable value function approximation and rely on additional assumptions on the off-policy/offline data, in particular, that the offline data covers the state/action space that is explored by the comparator policy. We note that this line of work does not directly overlap with our current approach but provides a valuable tool for dealing with off-policy data.

**Agnostic RL in Low-Rank MDPs.** Sekhari et al. (2021) explored agnostic PAC RL in low-rank MDPs, and showed that one can perform agnostic learning w.r.t. any policy class for MDPs that have a small rank. While their guarantees are similar to ours, i.e., they compete with the best policy in the given class and do not assume access to a realizable dynamics / value-function class, the key objectives of the two works are complementary. Sekhari et al. (2021) explore assumptions on the underlying MDP dynamics which suffice for agnostic learning for any given policy class, whereas we ask what assumptions on the given policy class suffice for agnostic learning for any underlying dynamics. Exploring the benefits of structure in both the policy class and the underlying MDP in agnostic RL is an interesting direction for future research.

**Policy Gradient Methods.** A significant body of work in RL, in both theory (Agarwal et al., 2021; Abbasi-Yadkori et al., 2019; Bhandari and Russo, 2019; Liu et al., 2020; Agarwal et al., 2020; Zhan et al., 2021; Xiao, 2022) and practice (Kakade, 2001; Kakade and Langford, 2002; Levine and Koltun, 2013; Schulman et al., 2015, 2017), studies policy-gradient based methods that directly search for the best policy in a given policy class. These approaches often leverage mirror-descent style analysis, and can deliver guarantees that are similar to ours, i.e. the returned policy can compete with any policy in the given class, which is an agnostic guarantee in some sense. However, these works primarily study smooth and parametric policy classes, e.g. tabular and linear policy classes, which limits their applicability for a broader range of problem instances. Furthermore, they require strong additional assumptions to work: for instance, that the learner is given a good reset distribution that can cover the occupancy measure of the policy that we wish to compare to, and that the policy class satisfies a certain "policy completeness assumption"; both of which are difficult to verify in practice. In contrast, our work makes no such assumptions but instead studies what kind of policy classes are learnable for any MDP.

**CPI, PSDP, and Other Reductions to Supervised Learning.** Various RL methods have been developed that return a policy that performs as well as the best policy in the given policy class, by reducing the RL problem from supervised learning. The key difference from policy-gradient based methods (discussed previously) is that these approaches do not require a smoothly parameterized policy class, but instead rely on access to a supervised learning oracle w.r.t. the given policy class. Popular approaches include Conservative Policy Iteration (CPI) (Kakade and Langford, 2002; Kakade, 2003; Brukhim et al., 2022b; Agarwal et al., 2023), PSDP (Bagnell et al., 2003), Behavior Cloning (Ross and Bagnell, 2010; Torabi et al., 2018), etc. We note that these algorithms rely on additional assumptions, including "policy completeness assumption" and a good sampling / reset distribution that covers the policies that we wish to compare to; in comparison, we do not make any such assumptions in our work.

Efficient RL via reductions to online regression oracles w.r.t. the given policy class has also been studied, see, e.g., DAgger (Ross et al., 2011), AggreVaTe (Ross and Bagnell, 2014), etc. However, these algorithms rely on much stronger feedback. In particular the learner, on the states which it visits, can query an expert policy (that we wish to complete with) for its actions or the value function. On the other hand, in this paper, we restrict ourselves to the standard RL setting where the learner only gets instantaneous reward signal. In Appendix I we investigate whether such stronger feedback can be used for agnostic RL.

**Reward-Free RL.** From a technical viewpoint, our algorithm (Algorithm 1) share similarities to algorithms developed in the reward-free RL literature (Jin et al., 2020). In reward-free RL, the goal of the learner is to output a dataset or a set of policies, after interacting with the underlying MDP, that can be later used for planning (with no further interaction with the MDP) for downstream reward functions. The key ideas in our Algorithm 1, in particular, that the learner first finds states $\mathcal{I}$ that are $\Omega(\varepsilon)$-reachable and corresponding policies that can reach them, and then outputs datasets $\{\mathcal{D}_s\}_{s \in \mathcal{I}}$ that can be later used for evaluating any policy $\pi \in \Pi$, share similarities to algorithmic ideas used in reward-free RL. However, our algorithm strictly generalizes prior works in reward-free RL, and in particular can work with large state-action spaces where the notion of reachability as well as the offline RL objective, is defined w.r.t. the given policy class. In comparison, prior reward-free RL works compete with the best policy for the underlying MDP, and make structure assumptions on the dynamics, e.g. tabular structure (Jin et al., 2020; Ménard et al., 2021; Li et al., 2023) or linear dynamics (Wang et al., 2020a; Zanette et al., 2020; Zhang et al., 2021a; Wagenmaker et al., 2022), to make the problem tractable.

**Instance Optimal Measures.** Several recent works including Wagenmaker and Jamieson (2022); Tirinzoni et al. (2023); Bottou et al. (2013); Al-Marjani et al. (2023) have explored instance-dependent complexity measures for PAC RL. At a high level, these instance-dependent bounds are obtained via similar algorithmic ideas to ours that combine reward-free exploration with policy elimination. However, there are major differences. Firstly, these prior works in instance-dependent PAC RL operate under additional modeling assumptions on the MDP dynamics, e.g., that it is a tabular or linear MDP. Secondly, they require additional reachability assumptions on the state space, which is restrictive for MDPs with large states/actions; in fact, their sample complexity bounds typically have a dependence on the number of states/actions in the lower order terms. Finally, they implicitly assume that the optimal policy $\pi^\star \in \Pi$, and thus the provided algorithms do not transfer cleanly to the agnostic PAC RL setting considered in our paper.

**Other Complexity Measures for RL.** A recent work by Mou et al. (2020) proposed a new notion of eluder dimension for the policy class, and provide upper bounds for policy-based RL when the class $\Pi$ has bounded eluder dimension. However, they require various additional assumptions: that the policy class contains the optimal policy, the learner has access to a generative model, and that the optimal value function has a gap. On the other hand, we do not make any such assumptions and characterize learnability in terms of spanning capacity or the size of the minimal sunflower in $\Pi$. We discuss connections to the eluder dimension, as well as other classical complexity measures in learning theory in Appendix H.

## B    Examples of Policy Classes

In this section, we will prove that examples in Section 3 have both bounded spanning capacity and the sunflower property with small $K$ and $D$. To facilitate our discussion, we define the following notation: for any policy class $\Pi$ we let

$$\mathfrak{C}_h(\Pi) := \max_{M \in \mathcal{M}^{\mathrm{det}}} C_h^{\mathrm{reach}}(\Pi; M),$$

where $C_h^{\mathrm{reach}}(\Pi; M)$ is defined in Definition 1. That is, $\mathfrak{C}_h(\Pi)$ is the per-layer spanning capacity of $\Pi$. Then as defined in Definition 1, we have

$$\mathfrak{C}(\Pi) = \max_{h \in [H]} \mathfrak{C}_h(\Pi).$$

**Tabular MDP.** Since there are at most $|\mathcal{S}_h|$ states in layer $h$, it is obvious that $\mathfrak{C}_h(\Pi) \leq |\mathcal{S}_h|A$, so therefore $\mathfrak{C}(\Pi) \leq SA$. Additionally, if we choose $\Pi_{\mathrm{core}} = \{\pi_a : \pi_a(s) = a, a \in \mathcal{A}\}$ to be the set

of policies which play the constant $a$ for each $a \in \mathcal{A}$ and $\mathcal{S}_\pi = \mathcal{S}$ for every $\pi \in \Pi$, then any partial trajectory which satisfies the condition in Definition 4 is of the form $(s_h, a_h)$, which is consistent with $\pi_{a_h} \in \Pi_{\text{core}}$. Hence $\Pi$ is a $(A, S)$-sunflower.

**Contextual Bandit.** Since there is only one layer, any deterministic MDP has a single state with at most $A$ actions possible, so $\mathfrak{C}(\Pi) \leq A$. Additionally, if we choose $\Pi_{\text{core}} = \{\pi_a : \pi_a(s) \equiv a, a \in \mathcal{A}\}$, and $\mathcal{S}_\pi = \emptyset$ for every $\pi \in \Pi$, then any partial trajectory which satisfies the condition in Definition 4 is in the form $(s, a)$, which is consistent with $\pi_a \in \Pi_{\text{core}}$. Hence $\Pi$ is a $(A, 0)$-sunflower.

**$H$-Layer Contextual Bandit.** By induction, it is easy to see that any deterministic MDP spans at most $A^{h-1}$ states in layer $h$, each of which has at most $A$ actions. Hence $\mathfrak{C}(\Pi) \leq A^H$. Additionally, if we choose
$$\Pi_{\text{core}} = \{\pi_{a_1, \cdots, a_H} : \pi_{a_1, \cdots, a_H}(s_h) \equiv a_h, a_1, \cdots, a_H \in \mathcal{A}\}$$
and $\mathcal{S}_\pi = \emptyset$ for every $\pi \in \Pi$, then any partial trajectory which satisfies the condition in Definition 4 is in the form $(s_1, a_1, \cdots, s_H, a_H)$, which is consistent with $\pi_{a_1, a_2, \cdots, a_H} \in \Pi_{\text{core}}$. Hence $\Pi$ is a $(A^H, 0)$-sunflower.

**$\ell$-tons.** In the following, we will denote $\Pi_\ell := \Pi_{\ell-\text{ton}}$. We will first prove that $\mathfrak{C}(\Pi_\ell) \leq 2H^\ell$. To show this, we will prove that $\mathfrak{C}_h(\Pi_\ell) \leq 2h^\ell$ by induction on $H$. When $H = 1$, the class is a subclass of the above contextual bandit class, hence we have $\mathfrak{C}_1(\Pi_\ell) \leq 2$. Next, suppose $\mathfrak{C}_{h-1}(\Pi_\ell) \leq 2(h-1)^\ell$. Fix any deterministic MDP and call the first state $s_1$. Policies taking $a = 1$ at $s_1$ can only take $a = 1$ on at most $\ell - 1$ states in the following layers. Such policies reach at most $\mathfrak{C}_{h-1}(\Pi_{\ell-1})$ states in layer $h$. Policies taking $a = 0$ at $s_1$ can only take $a = 1$ on at most $\ell$ states in the following layers. Such policies reach at most $\mathfrak{C}_{h-1}(\Pi_\ell)$ states in layer $h$. Hence we get

$$\mathfrak{C}_h(\Pi_\ell) \leq \mathfrak{C}_{h-1}(\Pi_{\ell-1}) + \mathfrak{C}_{h-1}(\Pi_\ell) \leq 2(h-1)^{\ell-1} + 2(h-1)^\ell \leq 2h^\ell.$$

This finishes the proof of the induction hypothesis. Based on the induction argument, we get

$$\mathfrak{C}(\Pi_\ell) = \max_{h \in [H]} \mathfrak{C}_h(\Pi_\ell) \leq 2H^\ell.$$

Additionally, choose

$$\Pi_{\text{core}} = \{\pi_0\} \cup \{\pi_h : 1 \leq h \leq H\},$$

where $\pi_0(s) \equiv 0$, and $\pi_h$ chooses the action 1 on all the states at layer $h$, i.e., $\pi_h(s) := \mathbb{1}\{s \in \mathcal{S}_h\}$. For every $\pi \in \Pi_\ell$, we choose $\mathcal{S}_\pi$ to be the states for which $\pi(s) = 1$ (there are at most $\ell$ such states). Fix any partial trajectory $\tau = (s_h, a_h \cdots, s_{h'}, a_{h'})$ which satisfies $\pi \rightsquigarrow \tau$. Suppose that for all $i \in (h, h']$, $s_i \notin \mathcal{S}_\pi$. Then we must have $a_i = 0$ for all $i \in (h, h']$. Hence $\pi_h \rightsquigarrow \tau$ (if $a_h = 1$) or $\pi_0 \rightsquigarrow \tau$ (if $a_h = 0$), and $\tau$ is consistent with some policy in $\Pi_{\text{core}}$. Therefore, $\Pi_\ell$ is an $(H + 1, \ell)$-sunflower.

**1-Active Policies.** We will first prove that $\mathfrak{C}(\Pi_{1-\text{act}}) \leq 2H$. For any deterministic MDP, we use $\bar{\mathcal{S}}_h$ to denote the set of states reachable by $\Pi_{1-\text{act}}$ at layer $h$. We will show that $\bar{\mathcal{S}}_h \leq 2h$ by induction on $h$. For $h = 1$, this holds since any deterministic MDP has only one state in the first layer. Suppose it holds at layer $h$. Then, we have

$$|\bar{\mathcal{S}}_{h+1}| \leq |\{(s, \pi(s)) : s \in \bar{\mathcal{S}}_h, \pi \in \Pi\}|.$$

Note that policies in $\Pi_{1-\text{act}}$ must take $a = 0$ on every $s \notin \{s_{(1,1)}, s_{(1,2)}, \cdots, s_{(1,H)}\}$. Hence $|\{(s, \pi(s)) \mid s \in \bar{\mathcal{S}}_h, \pi \in \Pi\}| \leq |\bar{\mathcal{S}}_h| + 1 \leq h + 1$. Thus, the induction argument is complete. As a consequence we have $\mathfrak{C}_h(\Pi) \leq 2h$ for all $h$, so

$$\mathfrak{C}(\Pi_{1-\text{act}}) = \max_{h \in [H]} \mathfrak{C}_h(\Pi_{1-\text{act}}) \leq 2H.$$

Additionally, if we choose $\mathcal{S}_\pi = \{s_{(1,1)}, s_{(1,2)}, \cdots, s_{(1,H)}\}$ for all $\pi \in \Pi$ as well as

$$\Pi_{\text{core}} = \{\pi_0\} \cup \{\pi_h : 1 \leq h \leq H\},$$

where $\pi_0(s) \equiv 0$ and $\pi_h(s) := \mathbb{1}\{s \in \mathcal{S}_h\}$. Now fix any partial trajectory $\tau = (s_h, a_h \cdots, s_{h'}, a_{h'})$ which satisfies $\pi \rightsquigarrow \tau$. If we have $i \in (h, h']$, $s_i \notin \mathcal{S}_\pi$, then we must have $a_i = 0$. Thus, $\pi_h \rightsquigarrow \tau$ (if $a_h = 1$) or $\pi_0 \rightsquigarrow \tau$ (if $a_h = 0$), so $\tau$ is consistent with some policy in $\Pi_{\text{core}}$. Therefore, $\Pi_{1-\text{act}}$ is a $(H + 1, H)$-sunflower.

**All-Active Policies.** For any deterministic MDP, there is a single state $s_{(j,1)}$ in the first layer. Any policy which takes $a = 1$ at state $s_{(j,1)}$ must belong to $\Pi_{j-\text{act}}$. Hence such policies can reach at most $\mathfrak{C}_{h-1}(\Pi_{j-\text{act}})$ states in layer $h$. For polices which take action 0 at state $h$, all these policies will transit to a fixed state in layer 2. Hence such policies can reach at most $\mathfrak{C}_{h-1}(\Pi_{\text{act}})$ states at layer $h$. Therefore, we get

$$\mathfrak{C}_h(\Pi_{\text{act}}) \leq \mathfrak{C}_{h-1}(\Pi_{\text{act}}) + \max_j \mathfrak{C}_{h-1}(\Pi_{j-\text{act}}) \leq \mathfrak{C}_{h-1}(\Pi_{\text{act}}) + 2(h-1).$$

By telescoping, we get

$$\mathfrak{C}_h(\Pi_{\text{act}}) \leq h(h-1),$$

which indicates that

$$\mathfrak{C}(\Pi_{\text{act}}) = \max_{h \in [H]} \mathfrak{C}_h(\Pi_{\text{act}}) \leq H(H-1).$$

Additionally, if we choose $\mathcal{S}_\pi = \{s_{(j,1)}, \cdots, s_{(j,H)}\}$ for all $\pi \in \Pi_{j-\text{act}}$, as well as

$$\Pi_{\text{core}} = \{\pi_0\} \cup \{\pi_h : 1 \leq h \leq H\},$$

where $\pi_0(s) := 0$ and $\pi_h(s) := \mathbb{1}\{s \in \mathcal{S}_h\}$. Now fix any partial trajectory $\tau = (s_h, a_h \cdots, s_{h'}, a_{h'})$ which satisfies $\pi \rightsquigarrow \tau$. If we have $i \in (h, h']$, $s_i \notin \mathcal{S}_\pi$, then we must have $a_i = 0$. Thus, $\pi_h \rightsquigarrow \tau$ (if $a_h = 1$) or $\pi_0 \rightsquigarrow \tau$ (if $a_h = 0$), so $\tau$ is consistent with some policy in $\Pi_{\text{core}}$. Therefore, $\Pi_{\text{act}}$ is a $(H+1, H)$-sunflower.

**Policy Classes for Continuous State Spaces.** In some cases, it is possible to construct policy classes over continuous state spaces that have bounded spanning capacity. For example, consider $\Pi_{\text{sing}}$, which is defined over a discrete (but large) state space. We can extend this to continuous state space by defining new state spaces $\mathcal{S}_h = \{s_{(x,h)} : x \in \mathbb{R}\}$ for all $h \in [H]$, action space $\mathcal{A} = \{0, 1\}$, and policy class

$$\widetilde{\Pi_{\text{sing}}} := \{\pi_{(i,h')} : \pi_{(i,h')}(s_{(x,h)}) = \mathbb{1}\{x \in [i, i+1) \text{ and } h = h'\}, i \in \mathbb{N}, h' \in [H]\}.$$

Essentially, we have expanded each state to be an interval on the real line. Using the same reasoning, we have the bound $\mathfrak{C}(\widetilde{\Pi_{\text{sing}}}) = H + 1$. One can also generalize this construction to the policy class $\widetilde{\Pi_{\ell-\text{ton}}}$ and preserve the same value of $\mathfrak{C}$.[5]

However, in general, this expansion to continuous state spaces may blow up the spanning capacity. Consider a similar modification to $\Pi_{1-\text{act}}$ (again, with the same new state space and action space $\mathcal{A} = \{0, 1\}$):

$$\widetilde{\Pi_{1-\text{act}}} := \{\pi : \pi(s_{(x,h)}) = 0 \text{ if } x \notin [0, 1)]\}.$$

While $\mathfrak{C}(\Pi_{1-\text{act}}) = \Theta(H)$, it is easy to see that $\mathfrak{C}(\widetilde{\Pi_{1-\text{act}}}) = 2^H$ since one can construct a $H$-layer deterministic tree using states in $[0, 1)$ as every $(s, a)$ pair at layer $H$ will be reachable by $\widetilde{\Pi_{1-\text{act}}}$.

## C   Proofs for Section 3

### C.1   Proof of Lemma 1

Fix any $M \in \mathcal{M}^{\text{sto}}$, as well as $h \in [H]$. We claim that

$$\Gamma_h := \sum_{s_h \in \mathcal{S}_h, a_h \in \mathcal{A}_h} \sup_{\pi \in \Pi} d_h^\pi(s_h, a_h; M) \leq \max_{M' \in \mathcal{M}^{\text{det}}} C_h^{\text{reach}}(\Pi; M'). \tag{3}$$

Here, $d_h^\pi(s_h, a_h; M)$ is the state-action visitation distribution of the policy $\pi$ on MDP $M$.

We first set up additional notation. Let us define a *prefix* as any tuple of pairs of the form

$$(s_1, a_1, s_2, a_2, \ldots, s_k, a_k) \quad \text{or} \quad (s_1, a_1, s_2, a_2, \ldots, s_k, a_k, s_{k+1}).$$

---

[5]To compute the $(K, D)$ values of $\widetilde{\Pi_{\ell-\text{ton}}}$, the previous arguments do not go through, since the sets $\mathcal{S}_\pi$ are infinite. With a suitable extension of Definition 4 to allow for non-Markovian $\Pi_{\text{core}}$, it is possible to show that $\widetilde{\Pi_{\ell-\text{ton}}}$ is an $(\mathcal{O}(H^\ell), 0)$-sunflower; Furthermore, the proof of Theorem 4 can be easily adapted to work under this extension.

We will denote prefix sequences as $(s_{1:k}, a_{1:k})$ or $(s_{1:k+1}, a_{1:k})$ respectively. For any prefix $(s_{1:k}, a_{1:k})$ (similarly prefixes of the type $(s_{1:k+1}, a_{1:k})$) we let $d_h^\pi(s_h, a_h \mid (s_{1:k}, a_{1:k}); M)$ denote the conditional probability of reaching $(s_h, a_h)$ under policy $\pi$ given one observed the prefix $(s_{1:k}, a_{1:k})$ in MDP $M$, with $d_h^\pi(s_h, a_h \mid (s_{1:k}, a_{1:k}); M) = 0$ if $\pi \not\rightsquigarrow (s_{1:k}, a_{1:k})$ or $\pi \not\rightsquigarrow (s_h, a_h)$.

In the following proof, we assume that the start state $s_1$ is fixed, but this is without any loss of generality, and the proof can easily be adapted to hold for stochastic start states.

Our strategy will be to explicitly compute the quantity $\Gamma_h$ in terms of the dynamics of $M$ and show that we can upper bound it by a "derandomized" MDP $M'$ which maximizes reachability at layer $h$. Let us unroll one step of the dynamics:

$$
\begin{aligned}
\Gamma_h &:= \sum_{s_h \in \mathcal{S}_h, a_h \in \mathcal{A}} \sup_{\pi \in \Pi} d_h^\pi(s_h, a_h; M) \\
&\overset{(i)}{=} \sum_{s_h \in \mathcal{S}_h, a_h \in \mathcal{A}} \sup_{\pi \in \Pi} d_h^\pi(s_h, a_h \mid s_1; M), \\
&\overset{(ii)}{=} \sum_{s_h \in \mathcal{S}_h, a_h \in \mathcal{A}} \sup_{\pi \in \Pi} \left\{ \sum_{a_1 \in \mathcal{A}} d_h^\pi(s_h, a_h \mid s_1, a_1; M) \right\} \\
&\overset{(iii)}{\leq} \sum_{a_1 \in \mathcal{A}} \sum_{s_h \in \mathcal{S}_h, a_h \in \mathcal{A}} \sup_{\pi \in \Pi} d_h^\pi(s_h, a_h \mid s_1, a_1; M).
\end{aligned}
$$

The equality $(i)$ follows from the fact that $M$ always starts at $s_1$. The equality $(ii)$ follows from the fact that $\pi$ is deterministic, so there exists exactly one $a' = \pi(s_1)$ for which $d_h^\pi(s_h, a_h \mid s_1, a'; M) = d_h^\pi(s_h, a_h \mid s_1; M)$, with all other $a'' \neq a'$ satisfying $d_h^\pi(s_h, a_h \mid s_1, a''; M) = 0$. The inequality $(iii)$ follows by swapping the supremum and the sum.

Continuing in this way, we can show that

$$
\begin{aligned}
\Gamma_h &= \sum_{a_1 \in \mathcal{A}} \sum_{s_h \in \mathcal{S}_h, a_h \in \mathcal{A}} \sup_{\pi \in \Pi} \left\{ \sum_{s_2 \in \mathcal{S}_2} P(s_2|s_1, a_1) \sum_{a_2 \in \mathcal{A}} d_h^\pi(s_h, a_h \mid (s_{1:2}, a_{1:2}); M) \right\} \\
&\leq \sum_{a_1 \in \mathcal{A}} \sum_{s_2 \in \mathcal{S}_2} P(s_2|s_1, a_1) \sum_{a_2 \in \mathcal{A}} \sum_{s_h \in \mathcal{S}_h, a_h \in \mathcal{A}} \sup_{\pi \in \Pi} d_h^\pi(s_h, a_h \mid (s_{1:2}, a_{1:2}); M) \\
&\quad\vdots \\
&\leq \sum_{a_1 \in \mathcal{A}} \sum_{s_2 \in \mathcal{S}_2} P(s_2|s_1, a_1) \sum_{a_2 \in \mathcal{A}} \cdots \sum_{s_{h-1} \in \mathcal{S}_{h-1}} P(s_{h-1}|s_{h-2}, a_{h-2}) \\
&\qquad\qquad \times \sum_{a_{h-1} \in \mathcal{A}} \sum_{s_h \in \mathcal{S}_h, a_h \in \mathcal{A}} \sup_{\pi \in \Pi} d_h^\pi(s_h, a_h \mid (s_{1:h-1}, a_{1:h-1}); M).
\end{aligned}
$$

Now we examine the conditional visitation $d_h^\pi(s_h, a_h \mid (s_{1:h-1}, a_{1:h-1}); M)$. Observe that it can be rewritten as

$$
d_h^\pi(s_h, a_h \mid (s_{1:h-1}, a_{1:h-1}); M) = P(s_h|s_{h-1}, a_{h-1}) \cdot \mathbb{1}\{\pi \rightsquigarrow (s_{1:h}, a_{1:h})\}.
$$

Plugging this back into the previous display, and again swapping the supremum and the sum, we get that

$$
\begin{aligned}
\Gamma_h &\leq \sum_{a_1 \in \mathcal{A}} \cdots \sum_{s_h \in \mathcal{S}_h} P(s_h|s_{h-1}, a_{h-1}) \sum_{a_h \in \mathcal{A}} \sup_{\pi \in \Pi} \mathbb{1}\{\pi \rightsquigarrow (s_{1:h}, a_{1:h})\} \\
&= \sum_{a_1 \in \mathcal{A}} \cdots \sum_{s_h \in \mathcal{S}_h} P(s_h|s_{h-1}, a_{h-1}) \sum_{a_h \in \mathcal{A}} \mathbb{1}\{\exists \pi \in \Pi : \pi \rightsquigarrow (s_{1:h}, a_{1:h})\}
\end{aligned}
$$

Our last step is to derandomize the stochastic transitions in the above stochastic MDP, simply by taking the sup over the transition probabilities:

$$
\Gamma_h \leq \sum_{a_1 \in \mathcal{A}} \sup_{s_2 \in \mathcal{S}_2} \sum_{a_2 \in \mathcal{A}} \cdots \sup_{s_h \in \mathcal{S}_h} \sum_{a_h \in \mathcal{A}} \mathbb{1}\{\exists \pi \in \Pi : \pi \rightsquigarrow (s_{1:h}, a_{1:h})\} = \max_{M' \in \mathcal{M}^{\text{det}}} C_h^{\text{reach}}(\Pi; M').
$$

The right hand side of the inequality is exactly the definition of $\max_{M' \in \mathcal{M}^{\mathrm{det}}} C_h^{\mathrm{reach}}(\Pi; M')$, thus proving Eq. (3). In particular, the above process defines the deterministic MDP which maximizes the reachability at level $h$. Taking the maximum over $h$ as well as supremum over $M$, we see that $\sup_{M \in \mathcal{M}^{\mathrm{sto}}} C^{\mathrm{cov}}(\Pi; M) \leq \mathfrak{C}(\Pi)$. Furthermore, from the definitions we have

$$\mathfrak{C}(\Pi) = \sup_{M \in \mathcal{M}^{\mathrm{det}}} C^{\mathrm{cov}}(\Pi; M) \leq \sup_{M \in \mathcal{M}^{\mathrm{sto}}} C^{\mathrm{cov}}(\Pi; M).$$

This concludes the proof of Lemma 1. $\qquad\square$

### C.2 Coverability is Not Sufficient for Online RL

In this section, we observe that bounded coverability by itself is not sufficient to ensure sample efficient agnostic PAC RL in the online interactive model. First note that Theorem 3 already shows this indirectly. In particular, in Theorem 3, we show that there exists a policy class with bounded spanning capacity that is hard to learn in online RL. However, recall Lemma 1 which implies that any policy class with bounded spanning capacity must also have bounded coverability; and thus the lower bound in Theorem 3 can be trivially extended to argue that bounded coverability by itself does not suffice for statistically efficient agnostic online RL.

However, we can also show the insufficiency of coverability through a much simpler route by directly invoking the lower bound construction in Sekhari et al. (2021). In particular, Sekhari et al. (2021) provides a construction for a low rank MDP with rich observations which satisfies $C^{\mathrm{cov}}(\Pi; M) = \mathcal{O}(1)$ for every $M \in \mathcal{M}$, but still needs $2^{\Omega(H)}$ many samples for any $(\Theta(1), \Theta(1))$-PAC learner (see Theorem 2 in their paper for more details; we simply set $d = \Theta(H)$ to get our lower bound).

We do not know if coverability is also insufficient for the generative model setting; we conjecture that one may be able to show, using a similar construction, that coverability is insufficient when $\mathfrak{C}(\Pi)$ is large, showing that one cannot adapt to benign problem instances.

## D   Proofs for Section 4

### D.1   Proof of Theorem 1

---
**Algorithm 2** TrajectoryTree (Kearns et al., 1999)

---
**Require:** Policy class $\Pi$, generative access to the underlying MDP $M$, number of samples $n$
1: Initialize dataset of trajectory trees $\mathcal{D} = \emptyset$.
2: **for** $i = 1, \ldots, n$ **do**
3:      Initialize trajectory tree $\widehat{T}_i = \emptyset$.
4:      Sample initial state $s_1^{(i)} \sim \mu$.
5:      **while** True **do**               // Sample transitions and rewards for a trajectory tree
6:          Find any unsampled $(s, a)$ s.t. $(s, a)$ is reachable in $\widehat{T}_i$ by some $\pi \in \Pi$.
7:          **if** no such $(s, a)$ exists **then break**
8:          **end if**
9:          Sample $s' \sim P(\cdot|s, a)$ and $r \sim R(s, a)$
10:          Add transition $(s, a, r, s')$ to $\widehat{T}_i$.
11:      **end while**
12:      $\mathcal{D} \leftarrow \mathcal{D} \cup \widehat{T}_i$.
13: **end for**
14: **for** $\pi \in \Pi$ **do**                                               // Policy evaluation
15:      Set $\widehat{V}^\pi \leftarrow \frac{1}{n} \sum_{i=1}^n \widehat{v}_i^\pi$, where $\widehat{v}_i^\pi$ is the cumulative reward of $\pi$ on $\widehat{T}_i$.
16: **end for**
17: **Return** $\widehat{\pi} \leftarrow \arg\max_{\pi \in \Pi} \widehat{V}^\pi$.

---

We show that, with minor changes, the TrajectoryTree algorithm of Kearns et al. (1999) attains the guarantee in Theorem 1. The pseudocode can be found in Algorithm 2. The key modification is line 6: we simply observe that only $(s, a)$ pairs which are reachable by some $\pi \in \Pi$ in the current

tree $\widehat{T}_i$ need to be sampled (in contrast, in the original algorithm of Kearns et al. (1999), they sample all $A^H$ transitions).

Fix any $\pi \in \Pi$. For every trajectory tree $i \in [n]$, the algorithm has collected enough transitions so that $\widehat{v}_i^\pi$ is well-defined, by line 6 of the algorithm. By the sampling process, it is clear that the values $\{\widehat{v}_i^\pi\}_{i \in [n]}$ are i.i.d. generated. We claim that they are unbiased estimates of $V^\pi$. Observe that one way of defining $V^\pi$ is the expected value of the following process:

(1) For every $(s, a) \in \mathcal{S} \times \mathcal{A}$, independently sample a next state $s' \sim P(\cdot | s, a)$ and a reward $r \sim R(s, a)$ to define a deterministic MDP $\widehat{M}$

(2) Return the value $\widehat{v}^\pi$ to be the value of $\pi$ run on $\widehat{M}$.

Define the law of this process as $\overline{\mathcal{Q}}$. The sampling process of TrajectoryTree (call the law of this process $\mathcal{Q}$) can be viewed as sampling the subset of $\widehat{M}^{\mathrm{det}}$ which is reachable by some $\pi \in \Pi$. Thus, we have

$$V^\pi = \mathbb{E}_{\widehat{M} \sim \overline{\mathcal{Q}}}[\widehat{v}^\pi] = \mathbb{E}_{\widehat{T} \sim \mathcal{Q}}\left[\mathbb{E}\left[\widehat{v}^\pi \mid \widehat{T}\right]\right] = \mathbb{E}_{\widehat{T} \sim \mathcal{Q}}[\widehat{v}^\pi],$$

where the second equality is due to the law of total probability, and the third equality is due to the fact that $\widehat{v}^\pi$ is measurable with respect to the trajectory tree $\widehat{T}$. Thus, $\{\widehat{v}_i^\pi\}_{i \in [n]}$ are unbiased estimates of $V^\pi$.

Therefore, by Hoeffding's inequality (Lemma 17) we see that $|V^\pi - \widehat{V}^\pi| \leq \sqrt{\frac{\log(2/\delta)}{2n}}$. Applying union bound we see that when the number of trajectory trees exceeds $n \gtrsim \frac{\log(|\Pi|/\delta)}{\varepsilon^2}$, with probability at least $1 - \delta$, for all $\pi \in \Pi$, the estimates satisfy $|V^\pi - \widehat{V}^\pi| \leq \varepsilon/2$. Thus the TrajectoryTree algorithm returns an $\varepsilon$-optimal policy. Since each trajectory tree uses at most $H \cdot \mathfrak{C}(\Pi)$ queries to the generative model, we have the claimed sample complexity bound. $\qquad\square$

## D.2 Proof of Theorem 2

Fix any worst-case deterministic MDP $M^\star$ which witnesses $\mathfrak{C}(\Pi)$ at layer $h^\star$. Since $\mathfrak{C}(\Pi)$ is a property depending on the dynamics of $M^\star$, we can assume that $M^\star$ has zero rewards. We can also assume that the algorithm knows $M^\star$ and $h^\star$ (this only makes the lower bound stronger). We construct a family of instances $\mathcal{M}^\star$ where all the MDPs in $\mathcal{M}^\star$ have the same dynamics as $M^\star$ but different nonzero rewards at the reachable $(s, a)$ pairs at layer $h^\star$.

Observe that we can embed a multi-armed bandit instance with $\mathfrak{C}(\Pi)$ arms using the class $\mathcal{M}^\star$. The value of any policy $\pi \in \Pi$ is exactly the reward that it receives at the *unique* $(s, a)$ pair in layer $h^\star$ that it reaches. Any $(\varepsilon, \delta)$-PAC algorithm that works over the family of instances $\mathcal{M}^\star$ must return a policy $\hat{\pi}$ that reaches an $(s, a)$ pair in layer $h^\star$ with near-optimal reward. Furthermore, in the generative model setting, the algorithm can only receive information about a single $(s, a)$ pair. Thus, such a PAC algorithm must also be able to PAC learn the best arm for multi-armed bandits with $\mathfrak{C}(\Pi)$ arms. Therefore, we can directly apply existing PAC lower bounds which show that the sample complexity of $(\varepsilon, \delta)$-PAC learning the best arm for $K$-armed multi-armed bandits is at least $\Omega(\frac{K}{\varepsilon^2} \cdot \log \frac{1}{\delta})$ (see, e.g., Mannor and Tsitsiklis, 2004). $\qquad\square$

## D.3 Proof of Corollary 1

The upper bound is obtained by a simple modification of the argument in the proof of Theorem 1. In terms of data collection, the trajectory tree collected every time is the same fixed deterministic MDP (with different rewards); furthermore, one can always execute line 6 and line 9 for a deterministic MDP since the algorithm can execute a sequence of actions to get to any new $(s, a)$ pair required by line 9. Thus in every episode of online interaction we are guaranteed to add the new $(s, a)$ pair to the trajectory tree.

The lower bound trivially extends because the proof of Appendix D.2 uses a family of MDPs with deterministic transitions (that are even known to the algorithm beforehand).

# E Proofs for Section 5

In this section, we prove Theorem 3, which shows a superpolynomial lower bound on the sample complexity required to learn bounded spanning capacity classes, ruling out $\text{poly}(\mathfrak{C}(\Pi), H, \log|\Pi|)$ sample complexity for online RL. We restate the theorem below with the precise constants:

**Theorem 5** (Lower bound for online RL). *Let $h_0 \in \mathbb{N}$ and $c \in (0, 1)$ be universal constants. Fix any $H \geq h_0$. Let $\varepsilon \in (1/2^{cH}, 1/(100H))$ and $\ell \in \{2, \ldots, H\}$ such that $1/\varepsilon^\ell \leq 2^H$. There exists a policy class $\Pi^{(\ell)}$ of size $1/(6\varepsilon^\ell)$ with $\mathfrak{C}(\Pi^{(\ell)}) \leq O(H^{4\ell+2})$ and a family of MDPs $\mathcal{M}$ with state space $\mathcal{S}$ of size $H \cdot 2^{2H+1}$, binary action space, horizon $H$ such that: for any $(\varepsilon/16, 1/8)$-PAC algorithm, there exists an $M \in \mathcal{M}$ in which the algorithm has to collect at least*

$$\min\left\{\frac{1}{120\varepsilon^\ell}, 2^{H/3-3}\right\} \quad \text{online trajectories in expectation.}$$

## E.1 Construction of State Space, Action Space, and Policy Class

**State and Action Spaces.** We define the state space $\mathcal{S}$. In every layer $h \in [H]$, there will be $2^{2H+1}$ states. The states will be paired up, and each state will be denoted by either $j[h]$ or $j'[h]$, so $\mathcal{S}_h = \{j[h] : j \in [2^{2H}]\} \cup \{j'[h] : j \in [2^{2H}]\}$. For any state $s \in \mathcal{S}$, we define the *index* of $s$, denoted $\text{idx}(s)$ as the unique $j \in [2^{2H}]$ such that $s \in \{j[h]\}_{h \in [H]} \cup \{j'[h]\}_{h \in [H]}$. In total there are $H \cdot 2^{2H+1}$ states. The action space is $\mathcal{A} = \{0, 1\}$.

**Policy Class.** For the given $\varepsilon$ and $\ell \in \{2, \ldots, H\}$, we show via a probabilistic argument the existence of a large policy class $\Pi^{(\ell)}$ which has bounded spanning capacity but is hard to explore. We state several properties in Lemma 2 which will be exploited in the lower bound.

We introduce some additional notation. For any $j \in [2^{2H}]$ we denote

$$\Pi_j^{(\ell)} := \{\pi \in \Pi^{(\ell)} : \exists h \in [H], \pi(j[h]) = 1\},$$

that is, $\Pi_j^{(\ell)}$ are the policies which take an action $a = 1$ on at least one state with index $j$.

We also define the set of *relevant state indices* for a given policy $\pi \in \Pi^{(\ell)}$ as

$$\mathcal{J}_{\text{rel}}^\pi := \{j \in [2^{2H}] : \pi \in \Pi_j^{(\ell)}\}.$$

For any policy $\pi$ we denote $\pi(j_{1:H}) := (\pi(j[1]), \ldots, \pi(j[H])) \in \{0, 1\}^H$ to be the vector that represents the actions that $\pi$ takes on the states in index $j$. The vector $\pi(j'_{1:H})$ is defined similarly.

**Lemma 2.** *Let $H$, $\varepsilon$, and $\ell$ satisfy the assumptions of Theorem 5. There exists a policy class $\Pi^{(\ell)}$ of size $N = 1/(6\varepsilon^\ell)$ which satisfies the following properties.*

*(1) For every $j \in [2^{2H}]$ we have $|\Pi_j^{(\ell)}| \in [\varepsilon N/2, 2\varepsilon N]$.*

*(2) For every $\pi \in \Pi$ we have $|\mathcal{J}_{\text{rel}}^\pi| \geq \varepsilon/2 \cdot 2^{2H}$.*

*(3) For every $\pi \in \Pi_j^{(\ell)}$, the vector $\pi(j_{1:H})$ is unique and always equal to $\pi(j'_{1:H})$.*

*(4) Bounded spanning capacity: $\mathfrak{C}(\Pi^{(\ell)}) \leq c \cdot H^{4\ell+2}$ for some universal constant $c > 0$.*

## E.2 Construction of MDP Family

The family $\mathcal{M} = \{M_{\pi^\star, \phi}\}_{\pi^\star \in \Pi^{(\ell)}, \phi \in \Phi}$ will be a family of MDPs which are indexed by a policy $\pi^\star$ as well as a *decoder* function $\phi : \mathcal{S} \mapsto \{\text{GOOD}, \text{BAD}\}$, which assigns each state to be "good" or "bad" in a sense that will be described later on. An example construction of an MDP $M_{\pi^\star, \phi}$ is illustrated in Figure 1. For brevity, the bracket notation used to denote the layer that each state lies in has been omitted in the figure.

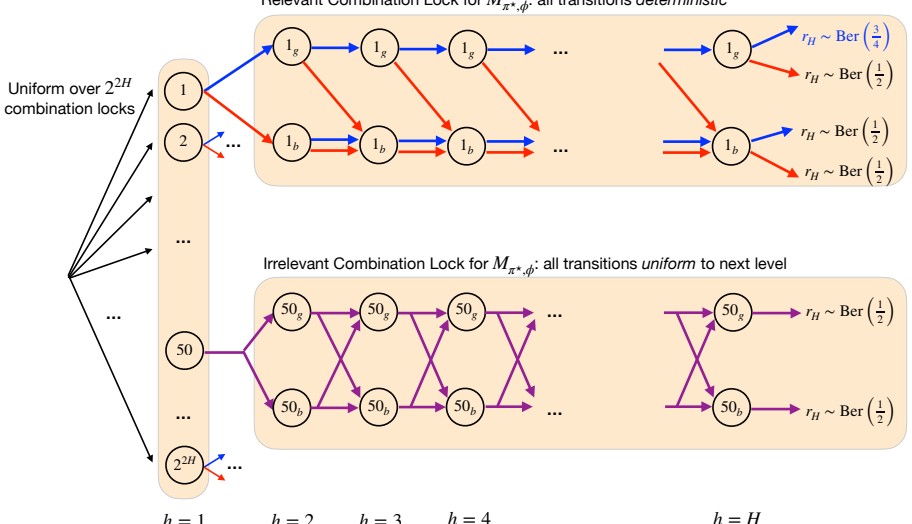

Figure 1: Illustration of the lower bound from Theorem 3. Blue arrows represent taking the action $\pi^\star(s)$, while red arrows represent taking the action $1 - \pi^\star(s)$. Purple arrows denote uniform transition to the states in the next layer, regardless of action. The MDP $M_{\pi^\star,\phi}$ is a uniform distribution of $2^{2H}$ combination locks of two types. In the *relevant* combination locks (such as Lock 1 in the figure), following $\pi^\star$ keeps one in the "good" chain and gives reward of $\mathrm{Ber}(3/4)$ in the last layer, while deviating from $\pi^\star$ leads one to the "bad" chain and gives reward of $\mathrm{Ber}(1/2)$. In *irrelevant* combination locks (such as Lock 50 in the figure), the next state is uniform regardless of action, and all rewards at the last layer are $\mathrm{Ber}(1/2)$.

**Decoder Function Class.** The decoder function class $\Phi$ will be the set of all possible mappings which for every $j \in [2^{2H}]$ and $h \geq 2$ assign exactly one of $j[h]$ or $j'[h]$ to the label GOOD (where the other is assigned to the label BAD). There are $(2^{H-1})^{2^{2H}}$ such functions. The label of a state will be used to describe the transition dynamics. Intuitively, a learner who does not know the decoder function $\phi$ will not be able to tell if a certain state has the label GOOD or BAD when visiting that state for the first time.

**Transition Dynamics.** The MDP $M_{\pi^\star,\phi}$ will be a uniform distribution over $2^{2H}$ combination locks $\{\mathsf{CL}_j\}_{j \in [2^{2H}]}$ with disjoint states. More formally, $s_1 \sim \mathrm{Uniform}(\{j[1]\}_{j \in [2^{2H}]})$. From each start state $j[1]$, only the $2H - 2$ states corresponding to index $j$ at layers $h \geq 2$ will be reachable in the combination lock $\mathsf{CL}_j$.

In the following, we will describe each combination lock $\mathsf{CL}_j$, which forms the basic building block of the MDP construction.

- **Good/Bad Set.** At every layer $h \in [H]$, for each $j[h]$ and $j'[h]$, the decoder function $\phi$ assigns one of them to be GOOD and one of them to be BAD. We will henceforth denote $j_g[h]$ to be the good state and $j_b[h]$ to be the bad state. Observe that by construction in Eq. (7), for every $\pi \in \Pi^{(\ell)}$ and $h \in [H]$ we have $\pi(j_g[h]) = \pi(j_b[h])$.

- **Dynamics of $\mathsf{CL}_j$, if $j \in \mathcal{J}^{\pi^\star}_{\mathrm{rel}}$.** Here, the transition dynamics of the combination locks are deterministic. For every $h \in [H]$,

  - On good states $j_g[h]$ we transit to the next good state iff the action is $\pi^\star$:

  $$P(s' \mid j_g[h], a) = \begin{cases} \mathbb{1}\{s' = j_g[h+1]\}, & \text{if } a = \pi^\star(j_g[h]) \\ \mathbb{1}\{s' = j_b[h+1]\}, & \text{if } a \neq \pi^\star(j_g[h]). \end{cases}$$

  - On bad states $j_b[h]$ we always transit to the next bad state:

  $$P(s' \mid j_b[h], a) = \mathbb{1}\{s' = j_b[h+1]\}, \quad \text{for all } a \in \mathcal{A}.$$

- **Dynamics of $\mathsf{CL}_j$, if $j \notin \mathcal{J}_{\mathrm{rel}}^{\pi^\star}$.** If $j$ is not a relevant index for $\pi^\star$, then the transitions are uniformly random regardless of the current state/action. For every $h \in [H]$,

$$P(\cdot \mid j_g[h], a) = P(\cdot \mid j_b[h], a) = \mathrm{Uniform}(\{j_g[h+1], j_b[h+1]\}), \quad \text{for all } a \in \mathcal{A}.$$

- **Reward Structure.** The reward function is nonzero only at layer $H$, and is defined as

$$R(s, a) = \mathrm{Ber}\left(\frac{1}{2} + \frac{1}{4} \cdot \mathbb{1}\{\pi^\star \in \Pi_j^{(\ell)}\} \cdot \mathbb{1}\{s = j_g[H], a = \pi^\star(j_g[H])\}\right)$$

That is, we get $3/4$ whenever we reach the $H$-th good state for an index $j$ which is relevant for $\pi^\star$, and $1/2$ reward otherwise.

**Reference MDPs.** We define several reference MDPs.

- In the reference MDP $M_0$, the initial start state is again taken to be the uniform distribution, i.e., $s_1 \sim \mathrm{Uniform}(\{j[1]\}_{j \in [2^{2H}]})$, and all the combination locks behave the same and have uniform transitions to the next state along the chain: for every $h \in [H]$ and $j \in [2^{2H}]$,

$$P(\cdot \mid j[h], a) = P(\cdot \mid j'[h], a) = \mathrm{Uniform}(\{j[h+1], j'[h+1]\}), \quad \text{for all } a \in \mathcal{A}.$$

The rewards for $M_0$ are $\mathrm{Ber}(1/2)$ for every $(s, a) \in \mathcal{S}_H \times \mathcal{A}$.

- For any decoder $\phi \in \Phi$, the reference MDP $M_{0,\pi^\star,\phi}$ has the same transitions as $M_{\pi^\star,\phi}$ but the rewards are $\mathrm{Ber}(1/2)$ for every $(s, a) \in \mathcal{S}_H \times \mathcal{A}$.

### E.3 Proof of Theorem 5

We are now ready to prove the lower bound using the construction of the MDP family $\mathcal{M}$.

**Value Calculation.** Consider any $M_{\pi^\star,\phi} \in \mathcal{M}$. For any policy $\pi \in \mathcal{A}^{\mathcal{S}}$ we use $V_{\pi^\star,\phi}(\pi)$ to denote the value of running $\pi$ in MDP $M_{\pi^\star,\phi}$. By construction we can see that

$$V_{\pi^\star,\phi}(\pi) = \frac{1}{2} + \frac{1}{4} \cdot \mathbb{P}_{\pi^\star,\phi}\left[\mathrm{idx}(s_1) \in \mathcal{J}_{\mathrm{rel}}^{\pi^\star} \text{ and } \pi(\mathrm{idx}(s_1)_{1:H}) = \pi^\star(\mathrm{idx}(s_1)_{1:H})\right], \qquad (4)$$

where in the above, we defined for any $s_1$, $\pi(\mathrm{idx}(s_1)_{1:H}) = \pi(j_{1:H}) = (\pi(j[1]), \ldots, \pi(j[H]))$, where $j$ denotes $\mathrm{idx}(s_1)$. Informally speaking, the second term counts the additional reward that $\pi$ gets for solving a combination lock rooted at a relevant state index $\mathrm{idx}(s_1) \in \mathcal{J}_{\mathrm{rel}}^{\pi^\star}$. By Property (2) and (3) of Lemma 2, we additionally have $V_{\pi^\star,\phi}(\pi^\star) \geq 1/2 + \varepsilon/8$, as well as $V_{\pi^\star,\phi}(\pi) = 1/2$ for all other $\pi \neq \pi^\star \in \Pi^{(\ell)}$.

By Eq. (4), if $\pi$ is an $\varepsilon/16$-optimal policy on $M_{\pi^\star,\phi}$ it must satisfy

$$\mathbb{P}_{\pi^\star,\phi}\left[\mathrm{idx}(s_1) \in \mathcal{J}_{\mathrm{rel}}^{\pi^\star} \text{ and } \pi(\mathrm{idx}(s_1)_{1:H}) = \pi^\star(\mathrm{idx}(s_1)_{1:H})\right] \geq \frac{\varepsilon}{4}.$$

**Averaged Measures.** We define the following measures which will be used in the analysis.

- Define $\mathbb{P}_{\pi^\star}[\cdot] = \frac{1}{|\Phi|} \sum_{\phi \in \Phi} \mathbb{P}_{\pi^\star,\phi}[\cdot]$ to be the averaged measure where we first pick $\phi$ uniformly among all decoders and then consider the distribution induced by $M_{\pi^\star,\phi}$.

- Define the averaged measure $\mathbb{P}_{0,\pi^\star}[\cdot] = \frac{1}{|\Phi|} \sum_{\phi \in \Phi} \mathbb{P}_{0,\pi^\star,\phi}[\cdot]$ where we pick $\phi$ uniformly and then consider the distribution induced by $M_{0,\pi^\star,\phi}$.

For both averaged measures the expectations $\mathbb{E}_{\pi^\star}$ and $\mathbb{E}_{0,\pi^\star}$ are defined analogously.

**Algorithm and Stopping Time.** Recall that an algorithm $\mathbb{A}$ is comprised of two phases. In the first phase, it collects some number of trajectories by interacting with the MDP in episodes. We use $\eta$ to denote the (random) number of episodes after which $\mathbb{A}$ terminates. We also use $\mathbb{A}_t$ to denote the intermediate policy that the algorithm runs in round $t$ for $t \in [\eta]$. In the second phase, $\mathbb{A}$ outputs[6] a

---

[6] We present the lower bound for the class of deterministic algorithms that output a deterministic policy. However, all the arguments could be extended to stochastic algorithms.

policy $\widehat{\pi}$. We use the notation $\mathbb{A}_f : \{\tau^{(t)}\}_{t\in[\eta]} \mapsto \mathcal{A}^{\mathcal{S}}$ to denote the second phase of $\mathbb{A}$ which outputs $\widehat{\pi}$ as a measurable function of collected data.

For any policy $\pi^\star$, decoder $\phi$, and dataset $\mathcal{D}$ we define the event

$$\mathcal{E}(\pi^\star, \phi, \mathbb{A}_f(\mathcal{D})) := \left\{ \mathbb{P}_{\pi^\star, \phi}\left[\mathrm{idx}(s_1) \in \mathcal{J}_{\mathrm{rel}}^{\pi^\star} \text{ and } \mathbb{A}_f(\mathcal{D})(\mathrm{idx}(s_1)_{1:H}) = \pi^\star(\mathrm{idx}(s_1)_{1:H})\right] \geq \frac{\varepsilon}{4} \right\}.$$

The event $\mathcal{E}(\pi^\star, \phi, \mathbb{A}_f(\mathcal{D}))$ is measurable with respect to the random variable $\mathcal{D}$, which denotes the collected data.

Under this notation, the PAC learning guarantee on $\mathbb{A}$ implies that for every $\pi^\star \in \Pi^{(\ell)}$, $\phi \in \Phi$ we have

$$\mathbb{P}_{\pi^\star, \phi}[\mathcal{E}(\pi^\star, \phi, \mathbb{A}_f(\mathcal{D}))] \geq 7/8.$$

Moreover via an averaging argument we also have

$$\mathbb{P}_{\pi^\star}[\mathcal{E}(\pi^\star, \phi, \mathbb{A}_f(\mathcal{D}))] \geq 7/8. \tag{5}$$

**Lower Bound Argument.** We apply a truncation to the stopping time $\eta$. Define $T_{\max} := 2^{H/3}$. Observe that if $\mathbb{P}_{\pi^\star}[\eta > T_{\max}] > 1/8$ for some $\pi^\star \in \Pi^{(\ell)}$ then the lower bound immediately follows, since

$$\max_{\phi \in \Phi} \mathbb{E}_{\pi^\star, \phi}[\eta] \; > \; \mathbb{E}_{\pi^\star}[\eta] \; \geq \; \mathbb{P}_{\pi^\star}[\eta > T_{\max}] \cdot T_{\max} \; \geq \; T_{\max}/8,$$

so there must exist an MDP $M_{\pi^\star, \phi}$ for which $\mathbb{A}$ collects at least $T_{\max}/8 = 2^{H/3-3}$ samples in expectation.

Otherwise we have $\mathbb{P}_{\pi^\star}[\eta > T_{\max}] \leq 1/8$ for all $\pi^\star \in \Pi^{(\ell)}$. This further implies that for all $\pi^\star \in \Pi^{(\ell)}$,

$$\mathbb{P}_{\pi^\star}[\eta < T_{\max} \text{ and } \mathcal{E}(\pi^\star, \phi, \mathbb{A}_f(\mathcal{D}))]$$
$$= \mathbb{P}_{\pi^\star}[\mathcal{E}(\pi^\star, \phi, \mathbb{A}_f(\mathcal{D}))] - \mathbb{P}_{\pi^\star}[\eta > T_{\max} \text{ and } \mathcal{E}(\pi^\star, \phi, \mathbb{A}_f(\mathcal{D}))] \geq 3/4. \tag{6}$$

However, in the following, we will argue that if Eq. (6) holds then $\mathbb{A}$ must query a significant number of samples in $M_0$.

**Lemma 3** (Stopping Time Lemma). *Let $\delta \in (0, 1/8]$. Let $\mathbb{A}$ be an $(\varepsilon/16, \delta)$-PAC algorithm. Let $T_{\max} \in \mathbb{N}$. Suppose that $\mathbb{P}_{\pi^\star}[\eta < T_{\max} \text{ and } \mathcal{E}(\pi^\star, \phi, \mathbb{A}_f(\mathcal{D}))] \geq 1 - 2\delta$ for all $\pi^\star \in \Pi^{(\ell)}$. The expected stopping time for $\mathbb{A}$ on $M_0$ is at least*

$$\mathbb{E}_0[\eta] \geq \left( \frac{|\Pi^{(\ell)}|}{2} - \frac{4}{\varepsilon} \right) \cdot \frac{1}{7} \log\left( \frac{1}{2\delta} \right) - |\Pi^{(\ell)}| \cdot \frac{T_{\max}^2}{2^{H+3}} \left( T_{\max} + \frac{1}{7} \log\left( \frac{1}{2\delta} \right) \right).$$

Using Lemma 3 with $\delta = 1/8$ and plugging in the value of $|\Pi^{(\ell)}|$ and $T_{\max}$, we see that

$$\mathbb{E}_0[\eta] \geq \left( \frac{|\Pi^{(\ell)}|}{2} - \frac{4}{\varepsilon} \right) \cdot \frac{1}{7} \log\left( \frac{1}{2\delta} \right) - |\Pi^{(\ell)}| \cdot \frac{T_{\max}^2}{2^{H+3}} \left( T_{\max} + \frac{1}{7} \log\left( \frac{1}{2\delta} \right) \right) \geq \frac{|\Pi^{(\ell)}|}{20}.$$

For the second inequality, we used the fact that $\ell \geq 2$, $H \geq 10^5$, and $\varepsilon < 1/10^7$.

We have shown that either there exists some MDP $M_{\pi^\star, \phi}$ for which $\mathbb{A}$ collects at least $T_{\max}/8 = 2^{H/3-3}$ samples in expectation, or $\mathbb{A}$ must query at least $|\Pi^{(\ell)}|/20 = 1/(120\varepsilon^\ell)$ trajectories in expectation in $M_0$. Putting it all together, the lower bound on the sample complexity is at least

$$\min\left\{ \frac{1}{120\varepsilon^\ell}, 2^{H/3-3} \right\}.$$

This concludes the proof of Theorem 5. $\qquad\square$

### E.4 Proof of Lemma 2

To prove Lemma 2, we first use a probabilistic argument to construct a certain binary matrix $B$ which satisfies several properties, and then construct $\Pi^{(\ell)}$ using $B$ and verify it satisfies Properties (1)-(4).

**Binary Matrix Construction.** First, we define a block-free property of binary matrices.

**Definition 5** (Block-free Matrices). *Fix parameters $k, \ell, N, d \in \mathbb{N}$ where $k \leq N$ and $l \leq d$. We say a binary matrix $B \in \{0,1\}^{N \times d}$ is $(k, \ell)$-block-free if the following holds: for every $I \subseteq [N]$ with $|I| = k$, and $J \subseteq [d]$ with $|J| = \ell$ there exists some $(i,j) \in I \times J$ with $B_{ij} = 0$.*

In words, matrices which are $(k, \ell)$-block-free do not contain a $k \times \ell$ block of all 1s.

**Lemma 4.** *Fix any $\varepsilon \in (0, 1/10)$ and $\ell \in \mathbb{N}$. For any*

$$d \in \left[ \frac{16\ell \cdot \log(1/\varepsilon)}{\varepsilon}, \frac{1}{20} \cdot \exp\left( \frac{1}{48\varepsilon^{\ell-1}} \right) \right],$$

*there exists a binary matrix $B \in \{0,1\}^{N \times d}$ with $N = 1/(6 \cdot \varepsilon^\ell)$ such that:*

1. *(Row sum): for every row $i \in [N]$, we have $\sum_j B_{ij} \geq \varepsilon d/2$.*

2. *(Column sum): for every column $j \in [d]$, we have $\sum_i B_{ij} \in [\varepsilon N/2, 2\varepsilon N]$.*

3. *The matrix $B$ is $(\ell \log d, \ell)$-block-free.*

**Proof of Lemma 4.** The existence of $B$ is proven using the probabilistic method. Let $\widetilde{B} \in \{0,1\}^{N \times d}$ be a random matrix where each entry is i.i.d. chosen to be 1 with probability $\varepsilon$.

By Chernoff bounds (Lemma 18), for every row $i \in [N]$, we have $\mathbb{P}[\sum_j B_{ij} \leq \frac{\varepsilon d}{2}] \leq \exp(-\varepsilon d/8)$; likewise for every column $j \in [d]$, we have $\mathbb{P}[\sum_j B_{ij} \notin [\frac{\varepsilon N}{2}, 2\varepsilon N]] \leq 2\exp(-\varepsilon N/8)$. By union bound, the matrix $\widetilde{B}$ satisfies the first two properties with probability at least $0.8$ as long as

$$d \geq (8 \log 10N)/\varepsilon, \quad \text{and} \quad N \geq (8 \log 20d)/\varepsilon.$$

One can check that under the choice of $N = 1/(6 \cdot \varepsilon^\ell)$ and the assumption on $d$, both constraints are met.

Now we examine the probability of $\widetilde{B}$ satisfies the block-free property with parameters $(k := \ell \log d, \ell)$. Let $X$ be the random variable which denotes the number of submatrices which violate the block-free property in $\widetilde{B}$, i.e.,

$$X = |\{I \times J : I \subset [N], |I| = k, J \subset [d], |J| = \ell, \widetilde{B}_{ij} = 1 \ \forall \ (i,j) \in I \times J\}|.$$

By linearity of expectation, we have

$$\mathbb{E}[X] \leq N^k d^\ell \varepsilon^{k\ell}.$$

We now plug in the choice $k = \ell \log d$ and observe that as long as $N \leq 1/(2e \cdot \varepsilon^\ell)$ we have $\mathbb{E}[X] \leq 1/2$. By Markov's inequality, $\mathbb{P}[X = 0] \geq 1/2$.

Therefore with positive probability, $\widetilde{B}$ satisfies all 3 properties (otherwise we would have a contradiction via inclusion-exlusion principle). Thus, there exists a matrix $B$ which satisfies all of the above three properties, proving the result of Lemma 4. $\square$

**Policy Class Construction.** For the given $\varepsilon$ and $\ell \in \{2, \dots, H\}$ we will use Lemma 4 to construct a policy class $\Pi^{(\ell)}$ which has bounded spanning capacity but is hard to explore. We instantiate Lemma 4 with the given $\ell$ and $d = 2^{2H}$, and use the resulting matrix $B$ to construct $\Pi^{(\ell)} = \{\pi_i\}_{i \in [N]}$ with $|\Pi^{(\ell)}| = N = 1/(6\varepsilon^\ell)$.

Recall that we assume that

$$H \geq h_0, \quad \text{and} \quad \varepsilon \in \left[ \frac{1}{2^{cH}}, \frac{1}{100H} \right].$$

We claim that under these assumptions, the requirement of Lemma 4 is met:

$$d = 2^{2H} \in \left[ \frac{16\ell \cdot \log(1/\varepsilon)}{\varepsilon}, \frac{1}{20} \cdot \exp\left( \frac{1}{48\varepsilon^{\ell-1}} \right) \right].$$

For the lower bound, we can check that:

$$\frac{16\ell \cdot \log(1/\varepsilon)}{\varepsilon} \leq 16H \cdot cH \cdot 2^{cH} \leq 2^{2H},$$

where we use the bound $\ell \leq H$ and $\varepsilon \geq 2^{-cH}$. The last inequality holds for sufficiently small universal constant $c \in (0,1)$ and sufficiently large $H \geq h_0$.

For the upper bound, we can also check that

$$\frac{1}{20} \cdot \exp\left(\frac{1}{48\varepsilon^{\ell-1}}\right) \geq \frac{1}{20} \cdot \exp\left(\frac{100H}{48}\right) \geq 2^{2H},$$

where we use the bound $\ell \geq 2$ and $\varepsilon \leq 1/(100H)$. The last inequality holds for sufficiently large $H$. We define the policies as follows: for every $\pi_i \in \Pi^{(\ell)}$ we set

$$\text{for every } j \in [2^{2H}]: \quad \pi_i(j[h]) = \pi_i(j'[h]) = \begin{cases} \text{bit}_h(\sum_{a \leq i} B_{aj}) & \text{if } B_{ij} = 1, \\ 0 & \text{if } B_{ij} = 0. \end{cases} \tag{7}$$

The function $\text{bit}_h : [2^H - 1] \mapsto \{0, 1\}$ selects the $h$-th bit in the binary representation of the input.

**Verifying Properties** $(1) - (4)$ **of Lemma 2.** Properties $(1) - (3)$ are straightforward from the construction of $B$ and $\Pi^{(\ell)}$, since $\pi_i \in \Pi_j^{(\ell)}$ if and only if $B_{ij} = 1$. The only detail which requires some care is that we require that $2\varepsilon N < 2^H$ in order for Property $(3)$ to hold, since otherwise we cannot assign the behaviors of the policies according to Eq. $(7)$. However, by assumption, this always holds, since $2\varepsilon N = 1/(3\varepsilon^{\ell-1}) \leq 2^H$.

We now prove Property $(4)$ that $\Pi^{(\ell)}$ has bounded spanning capacity. To prove this we will use the block-free property of the underlying binary matrix $B$.

Fix any deterministic MDP $M^\star$ which witnesses $\mathfrak{C}(\Pi^{(\ell)})$ at layer $h^\star$. To bound $\mathfrak{C}(\Pi^{(\ell)})$, we need to count the contribution to $C_{h^\star}^{\text{reach}}(\Pi; M^\star)$ from trajectories $\tau$ which are produced by some $\pi \in \Pi^{(\ell)}$ on $M$. We first define a *layer decomposition* for a trajectory $\tau = (s_1, a_1, s_2, a_2, \ldots, s_H, a_H)$ as the unique tuple of indices $(h_1, h_2, \ldots h_m)$, where each $h_k \in [H]$, that satisfies the following properties:

- The layers satisfy $h_1 < h_2 < \cdots < h_m$.
- The layer $h_1$ represents the first layer where $a_{h_1} = 1$.
- The layer $h_2$ represents the first layer where $a_{h_2} = 1$ on some state $s_{h_2}$ such that

$$\text{idx}(s_{h_2}) \notin \{\text{idx}(s_{h_1})\}.$$

- The layer $h_3$ represents the first layer where $a_{h_3} = 1$ on some state $s_{h_3}$ such that

$$\text{idx}(s_{h_3}) \notin \{\text{idx}(s_{h_1}), \text{idx}(s_{h_2})\}.$$

- More generally the layer $h_k$, $k \in [m]$ represents the first layer where $a_{h_k} = 1$ on some state $s_{h_k}$ such that

$$\text{idx}(s_{h_k}) \notin \{\text{idx}(s_{h_1}), \ldots, \text{idx}(s_{h_{k-1}})\}.$$

In other words, the layer $h_k$ represents the $k$-th layer for where action is $a = 1$ on a new state index which $\tau$ has never played $a = 1$ on before.

We will count the contribution to $C_{h^\star}^{\text{reach}}(\Pi; M^\star)$ by doing casework on the length of the layer decomposition for any $\tau$. That is, for every length $m \in \{0, \ldots, H\}$, we will bound $C_{h^\star}(m)$, which is defined to be the total number of $(s, a)$ at layer $h^\star$ which, for some $\pi \in \Pi^{(\ell)}$, a trajectory $\pi \rightsquigarrow \tau$ that has a $m$-length layer decomposition visits. Then we apply the bound

$$C_{h^\star}^{\text{reach}}(\Pi; M^\star) \leq \sum_{m=0}^{H} C_{h^\star}(m). \tag{8}$$

Note that this will overcount, since the same $(s, a)$ pair can belong to multiple different trajectories with different length layer decompositions.

**Lemma 5.** *The following bounds hold:*

- *For any $m \le \ell$, $C_{h^\star}(m) \le H^m \cdot \prod_{k=1}^{m}(2kH) = \mathcal{O}(H^{4m})$.*

- *We have $\sum_{m \ge \ell+1} C_{h^\star}(m) \le \mathcal{O}(\ell \cdot H^{4\ell+1})$.*

Therefore, applying Lemma 5 to Eq. (8), we have the bound that

$$\mathfrak{C}(\Pi^{(\ell)}) \le \left( \sum_{m \le \ell} O(H^{4m}) \right) + O(\ell \cdot H^{4\ell+1}) \le O(H^{4\ell+2}).$$

This concludes the proof of Lemma 2. $\qquad\qquad\qquad\qquad\qquad\qquad\qquad\qquad\qquad\quad\square$

**Proof of Lemma 5.** All of our upper bounds will be monotone in the value of $h^\star$, so we will prove the bounds for $C_H(m)$. In the following, fix any deterministic MDP $M^\star$.

First we start with the case where $m = 0$. The trajectory $\tau$ must play $a = 0$ at all times; since there is only one such $\tau$, we have $C_H(0) = 1$.

Now we will bound $C_H(m)$, for any $m \in \{1, \ldots, \ell\}$. Observe that there are $\binom{H}{m} \le H^m$ ways to pick the tuple $(h_1, \ldots, h_m)$. Now we will fix $(h_1, \ldots, h_m)$ and count the contributions to $C_H(m)$ for trajectories $\tau$ which have this fixed layer decomposition, and then sum up over all possible choices of $(h_1, \ldots, h_m)$.

In the MDP $M^\star$, there is a unique state $s_{h_1}$ which $\tau$ must visit. In the layers between $h_1$ and $h_2$, all trajectories are only allowed take 1 on states with index $\mathrm{idx}(s_{h_1})$, but they are not required to. Thus we can compute that the contribution to $C_{h_2}(m)$ from trajectories with the fixed layer decomposition to be at most $2H$. The reasoning is as follows. At $h_1$, there is exactly one $(s, a)$ pair which is reachable by trajectories with this fixed layer decomposition, since any $\tau$ must take $a = 1$ at $s_{h_1}$. Subsequently we can add at most two reachable pairs in every layer $h \in \{h_1 + 1, \ldots, h_2 - 1\}$ due to encountering a state $j[h]$ or $j'[h]$ where $j = \mathrm{idx}(s_{h_1})$, and at layer $h_2$ we must play $a = 1$, for a total of $1 + 2(h_2 - h_1 - 1) \le 2H$. Using similar reasoning the contribution to $C_{h_3}(m)$ from trajectories with this fixed layer decomposition is at most $(2H) \cdot (4H)$, and so on. Continuing in this way, we have the final bound of $\prod_{k=1}^{m}(2kH)$. Since this holds for a fixed choice of $(h_1, \ldots, h_m)$ in total we have $C_H(m) \le H^m \cdot \prod_{k=1}^{m}(2kH) = \mathcal{O}(H^{4m})$.

When $m \ge \ell + 1$, observe that the block-free property on $B$ implies that for any $J \subseteq [2^H]$ with $|J| = \ell$ we have $|\cap_{j \in J} \Pi_j| \le \ell \log 2^{2H}$. So for any trajectory $\tau$ with layer decomposition such that $m \ge \ell$ we can redo the previous analysis and argue that there is at most $\ell \log 2^{2H}$ multiplicative factor contribution to the value $C_H(m)$ due to *all* trajectories which have layer decompositions longer than $\ell$. Thus we arrive at the bound $\sum_{m \ge \ell+1} C_H(m) \le \mathcal{O}(H^{4\ell}) \cdot \ell \log 2^{2H} \le \mathcal{O}(\ell \cdot H^{4\ell+1})$. $\quad\square$

### E.5 Proof of Lemma 3

The proof of this stopping time lemma follows standard machinery for PAC lower bounds (Garivier et al., 2019; Domingues et al., 2021; Sekhari et al., 2021). In the following we use $\mathrm{KL}(P\|Q)$ to denote the Kullback-Leibler divergence between two distributions $P$ and $Q$ and $\mathrm{kl}(p\|q)$ to denote the Kullback-Leibler divergence between two Bernoulli distributions with parameters $p, q \in [0, 1]$.

For any $\pi^\star \in \Pi^{(\ell)}$ we denote the random variable

$$N^{\pi^\star} = \sum_{t=1}^{\eta \wedge T_{\max}} \mathbb{1}\left\{ \mathbb{A}_t(\mathrm{idx}(s_1)_{1:H}) = \pi^\star(\mathrm{idx}(s_1)_{1:H}) \text{ and } \mathrm{idx}(s_1) \in \mathcal{J}_{\mathrm{rel}}^{\pi^\star} \right\},$$

the number of episodes for which the algorithm's policy at round $t \in [\eta \wedge T_{\max}]$ matches that of $\pi^\star$ on a certain relevant state of $\pi^\star$.

In the sequel we will prove upper and lower bounds on the intermediate quantity $\sum_{\pi^\star \in \Pi} \mathbb{E}_0[N^{\pi^\star}]$ and relate these quantities to $\mathbb{E}_0[\eta]$.

**Step 1: Upper Bound.** First we prove an upper bound. We can compute that

$$\sum_{\pi^\star \in \Pi} \mathbb{E}_0\left[N^{\pi^\star}\right]$$

$$= \sum_{t=1}^{T_{\max}} \sum_{\pi^\star \in \Pi} \mathbb{E}_0\left[\mathbb{1}\{\eta > t-1\} \mathbb{1}\left\{\mathbb{A}_t(\mathrm{idx}(s_1)_{1:H}) = \pi^\star(\mathrm{idx}(s_1)_{1:H}) \text{ and } \mathrm{idx}(s_1) \in \mathcal{J}_{\mathrm{rel}}^{\pi^\star}\right\}\right]$$

$$= \sum_{t=1}^{T_{\max}} \mathbb{E}_0\left[\mathbb{1}\{\eta > t-1\} \sum_{\pi^\star \in \Pi} \mathbb{1}\left\{\mathbb{A}_t(\mathrm{idx}(s_1)_{1:H}) = \pi^\star(\mathrm{idx}(s_1)_{1:H}) \text{ and } \mathrm{idx}(s_1) \in \mathcal{J}_{\mathrm{rel}}^{\pi^\star}\right\}\right]$$

$$\overset{(i)}{\le} \sum_{t=1}^{T_{\max}} \mathbb{E}_0[\mathbb{1}\{\eta > t-1\}] \le \mathbb{E}_0[\eta \wedge T_{\max}] \le \mathbb{E}_0[\eta]. \tag{9}$$

Here, the first inequality follows because for every index $j$ and every $\pi^\star \in \Pi_j^{(\ell)}$, each $\pi^\star$ admits a unique sequence of actions (by Property (3) of Lemma 2), so any policy $\mathbb{A}_t$ can completely match with at most one of the $\pi^\star$.

**Step 2: Lower Bound.** Now we turn to the lower bound. We use a change of measure argument.

$$\mathbb{E}_0\left[N^{\pi^\star}\right] \overset{(i)}{\ge} \mathbb{E}_{0,\pi^\star}\left[N^{\pi^\star}\right] - T_{\max}\Delta(T_{\max})$$

$$= \frac{1}{|\Phi|} \sum_{\phi \in \Phi} \mathbb{E}_{0,\pi^\star,\phi}\left[N^{\pi^\star}\right] - T_{\max}\Delta(T_{\max})$$

$$\overset{(ii)}{\ge} \frac{1}{7} \cdot \frac{1}{|\Phi|} \sum_{\phi \in \Phi} \mathrm{KL}\left(\mathbb{P}_{0,\pi^\star,\phi}^{\mathcal{F}_{\eta \wedge T_{\max}}} \,\|\, \mathbb{P}_{\pi^\star,\phi}^{\mathcal{F}_{\eta \wedge T_{\max}}}\right) - T_{\max}\Delta(T_{\max})$$

$$\overset{(iii)}{\ge} \frac{1}{7} \cdot \mathrm{KL}\left(\mathbb{P}_{0,\pi^\star}^{\mathcal{F}_{\eta \wedge T_{\max}}} \,\|\, \mathbb{P}_{\pi^\star}^{\mathcal{F}_{\eta \wedge T_{\max}}}\right) - T_{\max}\Delta(T_{\max})$$

The inequality $(i)$ follows from a change of measure argument using Lemma 6, with $\Delta(T_{\max}) := T_{\max}^2/2^{H+3}$. Here, $\mathcal{F}_{\eta \wedge T_{\max}}$ denotes the natural filtration generated by the first $\eta \wedge T_{\max}$ episodes. The inequality $(ii)$ follows from Lemma 7, using the fact that $M_{0,\pi^\star,\phi}$ and $M_{\pi^\star,\phi}$ have identical transitions and only differ in rewards at layer $H$ for the trajectories which reach the end of a relevant combination lock. The number of times this occurs is exactly $N^{\pi^\star}$. The factor $1/7$ is a lower bound on $\mathrm{kl}(1/2\|3/4)$. The inequality $(iii)$ follows by the convexity of KL divergence.

Now we apply Lemma 8 to lower bound the expectation for any $\mathcal{F}_{\eta \wedge T_{\max}}$-measurable random variable $Z \in [0,1]$ as

$$\mathbb{E}_0\left[N^{\pi^\star}\right] \ge \frac{1}{7} \cdot \mathrm{kl}(\mathbb{E}_{0,\pi^\star}[Z]\|\mathbb{E}_{\pi^\star}[Z]) - T_{\max}\Delta(T_{\max})$$

$$\ge \frac{1}{7} \cdot (1 - \mathbb{E}_{0,\pi^\star}[Z]) \log\left(\frac{1}{1 - \mathbb{E}_{\pi^\star}[Z]}\right) - \frac{\log(2)}{7} - T_{\max}\Delta(T_{\max}),$$

where the second inequality follows from the bound $\mathrm{kl}(p\|q) \ge (1-p)\log(1/(1-q)) - \log(2)$ (see, e.g., Domingues et al., 2021, Lemma 15).

Now we pick $Z = Z_{\pi^\star} := \mathbb{1}\{\eta < T_{\max} \text{ and } \mathcal{E}(\pi^\star, \phi, \mathbb{A}_f(\mathcal{D}))\}$ and note that $\mathbb{E}_{\pi^\star}[Z_{\pi^\star}] \ge 1 - 2\delta$ by assumption. This implies that

$$\mathbb{E}_0\left[N^{\pi^\star}\right] \ge (1 - \mathbb{E}_{0,\pi^\star}[Z_{\pi^\star}]) \cdot \frac{1}{7} \log\left(\frac{1}{2\delta}\right) - \frac{\log(2)}{7} - T_{\max}\Delta(T_{\max}).$$

Another application of Lemma 6 gives

$$\mathbb{E}_0\left[N^{\pi^\star}\right] \ge (1 - \mathbb{E}_0[Z_{\pi^\star}]) \cdot \frac{1}{7} \log\left(\frac{1}{2\delta}\right) - \frac{\log(2)}{7} - \Delta(T_{\max})\left(T_{\max} + \frac{1}{7}\log\left(\frac{1}{2\delta}\right)\right).$$

Summing the above over $\pi^\star \in \Pi^{(\ell)}$, we get

$$\sum_{\pi^\star} \mathbb{E}_0\left[N^{\pi^\star}\right] \geq \left(|\Pi^{(\ell)}| - \sum_{\pi^\star} \mathbb{E}_0[Z_{\pi^\star}]\right) \cdot \frac{1}{7}\log\left(\frac{1}{2\delta}\right) - |\Pi^{(\ell)}| \cdot \frac{\log(2)}{7} - |\Pi^{(\ell)}| \cdot \Delta(T_{\max})\left(T_{\max} + \frac{1}{7}\log\left(\frac{1}{2\delta}\right)\right).$$
(10)

It remains to prove an upper bound on $\sum_{\pi^\star} \mathbb{E}_0[Z_{\pi^\star}]$. We calculate that

$$\sum_{\pi^\star} \mathbb{E}_0[Z_{\pi^\star}] = \sum_{\pi^\star} \mathbb{E}_0[\mathbb{1}\{\eta < T_{\max} \text{ and } \mathcal{E}(\pi^\star, \phi, \mathbb{A}_f(\mathcal{D}))\}]$$

$$\leq \sum_{\pi^\star} \mathbb{E}_0\left[\mathbb{1}\left\{\mathbb{P}_{\pi^\star}\left[\mathrm{idx}(s_1) \in \mathcal{J}_{\mathrm{rel}}^{\pi^\star} \text{ and } \mathbb{A}_f(\mathcal{D})(\mathrm{idx}(s_1)_{1:H}) = \pi^\star(\mathrm{idx}(s_1)_{1:H})\right] \geq \frac{\varepsilon}{4}\right\}\right]$$

$$\leq \frac{4}{\varepsilon} \cdot \mathbb{E}_0\left[\sum_{\pi^\star} \mathbb{P}_{\pi^\star}\left[\mathrm{idx}(s_1) \in \mathcal{J}_{\mathrm{rel}}^{\pi^\star} \text{ and } \mathbb{A}_f(\mathcal{D})(\mathrm{idx}(s_1)_{1:H}) = \pi^\star(\mathrm{idx}(s_1)_{1:H})\right]\right]$$
(11)

The last inequality is an application of Markov's inequality.

Now we carefully investigate the sum. For any $\phi \in \Phi$, the sum can be rewritten as

$$\sum_{\pi^\star} \mathbb{P}_{\pi^\star,\phi}\left[\mathrm{idx}(s_1) \in \mathcal{J}_{\mathrm{rel}}^{\pi^\star} \text{ and } \mathbb{A}_f(\mathcal{D})(\mathrm{idx}(s_1)_{1:H}) = \pi^\star(\mathrm{idx}(s_1)_{1:H})\right]$$

$$= \sum_{\pi^\star} \sum_{s_1 \in \mathcal{S}_1} \mathbb{P}_{\pi^\star,\phi}[s_1]\, \mathbb{P}_{\pi^\star,\phi}\left[\mathrm{idx}(s_1) \in \mathcal{J}_{\mathrm{rel}}^{\pi^\star} \text{ and } \mathbb{A}_f(\mathcal{D})(\mathrm{idx}(s_1)_{1:H}) = \pi^\star(\mathrm{idx}(s_1)_{1:H}) \mid s_1\right]$$

$$\overset{(i)}{=} \frac{1}{|\mathcal{S}_1|} \sum_{s_1 \in \mathcal{S}_1} \sum_{\pi^\star} \mathbb{P}_{\pi^\star,\phi}\left[\mathrm{idx}(s_1) \in \mathcal{J}_{\mathrm{rel}}^{\pi^\star} \text{ and } \mathbb{A}_f(\mathcal{D})(\mathrm{idx}(s_1)_{1:H}) = \pi^\star(\mathrm{idx}(s_1)_{1:H}) \mid s_1\right]$$

$$\overset{(ii)}{=} \frac{1}{|\mathcal{S}_1|} \sum_{s_1 \in \mathcal{S}_1} \sum_{\pi^\star} \mathbb{1}\left\{\mathrm{idx}(s_1) \in \mathcal{J}_{\mathrm{rel}}^{\pi^\star} \text{ and } \mathbb{A}_f(\mathcal{D})(\mathrm{idx}(s_1)_{1:H}) = \pi^\star(\mathrm{idx}(s_1)_{1:H})\right\}.$$
(12)

The equality $(i)$ follows because regardless of which MDP $M_{\pi^\star}$ we are in, the first state is distributed uniformly over $\mathcal{S}_1$. The equality $(ii)$ follows because once we condition on the first state $s_1$, the probability is either 0 or 1.

Fix any start state $s_1$. We can write

$$\sum_{\pi^\star} \mathbb{1}\left\{\mathrm{idx}(s_1) \in \mathcal{J}_{\mathrm{rel}}^{\pi^\star} \text{ and } \mathbb{A}_f(\mathcal{D})(\mathrm{idx}(s_1)_{1:H})\pi^\star(\mathrm{idx}(s_1)_{1:H})\right\}$$

$$= \sum_{\pi^\star \in \Pi_{\mathrm{idx}(s_1)}^{(\ell)}} \mathbb{1}\{\mathbb{A}_f(\mathcal{D})(\mathrm{idx}(s_1)_{1:H}) = \pi^\star(\mathrm{idx}(s_1)_{1:H})\} = 1,$$

where the second equality uses the fact that on any index $j$, each $\pi^\star \in \Pi_j^{(\ell)}$ behaves differently (Property (3) of Lemma 2), so $\mathbb{A}_f(\mathcal{D})$ can match at most one of these behaviors. Plugging this back into Eq. (12), averaging over $\phi \in \Phi$, and combining with Eq. (11), we arrive at the bound

$$\sum_{\pi^\star} \mathbb{E}_0[Z_{\pi^\star}] \leq \frac{4}{\varepsilon}.$$

We now use this in conjunction with Eq. (10) to arrive at the final lower bound

$$\sum_{\pi^\star} \mathbb{E}_0\left[N^{\pi^\star}\right] \geq \left(|\Pi^{(\ell)}| - \frac{4}{\varepsilon}\right) \cdot \frac{1}{7}\log\left(\frac{1}{2\delta}\right) - |\Pi^{(\ell)}| \cdot \frac{\log(2)}{7} - |\Pi^{(\ell)}| \cdot \Delta(T_{\max})\left(T_{\max} + \frac{1}{7}\log\left(\frac{1}{2\delta}\right)\right).$$
(13)

**Step 3: Putting it All Together.** Combining Eqs. (9) and (13), plugging in our choice of $\Delta(T_{\max})$, and simplifying we get

$$\mathbb{E}_0[\eta] \geq \left(|\Pi^{(\ell)}| - \frac{4}{\varepsilon}\right) \cdot \frac{1}{7}\log\left(\frac{1}{2\delta}\right) - |\Pi^{(\ell)}| \cdot \frac{\log(2)}{7} - |\Pi^{(\ell)}| \cdot \Delta(T_{\max})\left(T_{\max} + \frac{1}{7}\log\left(\frac{1}{2\delta}\right)\right).$$

$$\geq \left(\frac{|\Pi^{(\ell)}|}{2} - \frac{4}{\varepsilon}\right) \cdot \frac{1}{7}\log\left(\frac{1}{2\delta}\right) - |\Pi^{(\ell)}| \cdot \frac{T_{\max}^2}{2^{H+3}}\left(T_{\max} + \frac{1}{7}\log\left(\frac{1}{2\delta}\right)\right).$$

The last inequality follows since $\delta \leq 1/8$ implies $\log(1/(2\delta)) \geq 2\log(2)$.

This concludes the proof of Lemma 3. $\qquad\square$

### E.6  Change of Measure Lemma

**Lemma 6.** *Let $Z \in [0,1]$ be a $\mathcal{F}_{T_{\max}}$-measurable random variable. Then, for every $\pi^\star \in \Pi^{(\ell)}$,*

$$|\mathbb{E}_0[Z] - \mathbb{E}_{0,\pi^\star}[Z]| \leq \Delta(T_{\max}) := \frac{T_{\max}^2}{2^{H+3}}$$

**Proof.** First, we note that

$$|\mathbb{E}_0[Z] - \mathbb{E}_{0,\pi^\star}[Z]| \leq \mathrm{TV}\left(\mathbb{P}_0^{\mathcal{F}_{T_{\max}}}, \mathbb{P}_{0,\pi^\star}^{\mathcal{F}_{T_{\max}}}\right) \leq \sum_{t=1}^{T_{\max}} \mathbb{E}_0[\mathrm{TV}(\mathbb{P}_0[\cdot|\mathcal{F}_{t-1}], \mathbb{P}_{0,\pi^\star}[\cdot|\mathcal{F}_{t-1}])].$$

Here $\mathbb{P}_0[\cdot|\mathcal{F}_t]$ denotes the conditional distribution of the $t$-th trajectory given the first $t-1$ trajectories. Similarly $\mathbb{P}_{0,\pi^\star}[\cdot|\mathcal{F}_t]$ is the averaged over decoders condition distribution of the $t$-th trajectory given the first $t-1$ trajectories. The second inequality follows by chain rule of TV distance (see, e.g., Polyanskiy and Wu, 2022, pg. 152).

Now we examine each term $\mathrm{TV}(\mathbb{P}_0[\cdot|\mathcal{F}_{t-1}], \mathbb{P}_{0,\pi^\star}[\cdot|\mathcal{F}_{t-1}])$. Fix a history $\mathcal{F}_{t-1}$ and sequence $s_{1:H}$ where all $s_i$ have the same index. We want to bound the quantity

$$\left|\mathbb{P}_{0,\pi^\star}\left[S_{1:H}^{(t)} = s_{1:H} \mid \mathcal{F}_{t-1}\right] - \mathbb{P}_0\left[S_{1:H}^{(t)} = s_{1:H} \mid \mathcal{F}_{t-1}\right]\right|,$$

where it is understood that the random variable $S_{1:H}^{(t)}$ is drawn according to the MDP dynamics and algorithm's policy $\mathbb{A}_t$ (which is in turn a measurable function of $\mathcal{F}_{t-1}$).

We observe that the second term is exactly

$$\mathbb{P}_0\left[S_{1:H}^{(t)} = s_{1:H} \mid \mathcal{F}_{t-1}\right] = \frac{1}{|\mathcal{S}_1|} \cdot \frac{1}{2^{H-1}},$$

since the state $s_1$ appears with probability $1/|\mathcal{S}_1|$ and the transitions in $M_0$ are uniform to the next state in the combination lock, so each sequence is equally as likely.

For the first term, again the state $s_1$ appears with probability $1/|\mathcal{S}_1|$. Suppose that $\mathrm{idx}(s_1) \notin \mathcal{J}_{\mathrm{rel}}^{\pi^\star}$. Then the dynamics of $\mathbb{P}_{0,\pi^\star,\phi}$ for all $\phi \in \Phi$ are exactly the same as $M_0$, so again the probability in this case is $1/(|\mathcal{S}_1|2^{H-1})$. Now consider when $\mathrm{idx}(s_1) \in \mathcal{J}_{\mathrm{rel}}^{\pi^\star}$. At some point $\widehat{h} \in [H+1]$, the policy $\mathbb{A}_t$ will deviate from $\pi^\star$ for the first time (if $\mathbb{A}_t$ never deviates from $\pi^\star$ we set $\widehat{h} = H+1$). The layer $\widehat{h}$ is only a function of $s_1$ and $\mathbb{A}_t$ and does not depend on the MDP dynamics. The correct decoder must assign $\phi(s_{1:\widehat{h}-1}) = \mathrm{GOOD}$ and $\phi(s_{\widehat{h}:H}) = \mathrm{BAD}$, so therefore we have

$$\mathbb{P}_{0,\pi^\star}\left[S_{1:H}^{(t)} = s_{1:H} \mid \mathcal{F}_{t-1}\right] = \mathbb{P}_{0,\pi^\star}\left[\phi(s_{1:\widehat{h}-1}) = \mathrm{GOOD} \text{ and } \phi(s_{\widehat{h}:H}) = \mathrm{BAD} \mid \mathcal{F}_{t-1}\right]$$

If $s_1 \notin \mathcal{F}_{t-1}$, i.e., we are seeing $s_1$ for the first time, then the conditional distribution over the labels given by $\phi$ is the same as the unconditioned distribution:

$$\mathbb{P}_{0,\pi^\star}\left[\phi(s_{1:\widehat{h}-1}) = \mathrm{GOOD} \text{ and } \phi(s_{\widehat{h}:H}) = \mathrm{BAD} \mid \mathcal{F}_{t-1}\right] = \frac{1}{|\mathcal{S}_1|} \cdot \frac{1}{2^{H-1}}.$$

Otherwise, if $s_1 \in \mathcal{F}_{t-1}$ then we bound the conditional probability by 1.

$$\mathbb{P}_{0,\pi^\star}\left[S_{1:H}^{(t)} = s_{1:H} \mid \mathcal{F}_{t-1}\right] \leq \frac{1}{|\mathcal{S}_1|}.$$

Putting this all together we can compute

$$\mathbb{P}_{0,\pi^\star}\left[S_{1:H}^{(t)} = s_{1:H} \mid \mathcal{F}_{t-1}\right] \begin{cases} = \frac{1}{|\mathcal{S}_1|} \cdot \frac{1}{2^{H-1}} & \text{if } \mathrm{idx}(s_1) \notin \mathcal{J}_{\mathrm{rel}}^{\pi^\star}, \\ = \frac{1}{|\mathcal{S}_1|} \cdot \frac{1}{2^{H-1}} & \text{if } \mathrm{idx}(s_1) \in \mathcal{J}_{\mathrm{rel}}^{\pi^\star} \text{ and } s_1 \notin \mathcal{F}_{t-1}, \\ \leq \frac{1}{|\mathcal{S}_1|} & \text{if } \mathrm{idx}(s_1) \in \mathcal{J}_{\mathrm{rel}}^{\pi^\star} \text{ and } s_1 \in \mathcal{F}_{t-1}, \\ = 0 & \text{otherwise.} \end{cases}$$

Therefore we have the bound

$$\left| \mathbb{P}_{0,\pi^\star} \left[ S_{1:H}^{(t)} = s_{1:H} \mid \mathcal{F}_{t-1} \right] - \mathbb{P}_0 \left[ S_{1:H}^{(t)} = s_{1:H} \mid \mathcal{F}_{t-1} \right] \right| \leq \frac{1}{|\mathcal{S}_1|} \mathbb{1} \left\{ \mathrm{idx}(s_1) \in \mathcal{J}_{\mathrm{rel}}^{\pi^\star}, s_1 \in \mathcal{F}_{t-1} \right\}.$$

Summing over all possible sequences $s_{1:H}$ we have

$$\mathrm{TV}(\mathbb{P}_0[\cdot|\mathcal{F}_{t-1}], \mathbb{P}_{0,\pi^\star}[\cdot|\mathcal{F}_{t-1}]) \leq \frac{1}{2} \cdot \frac{(t-1) \cdot 2^{H-1}}{|\mathcal{S}_1|},$$

since the only sequences $s_{1:H}$ for which the difference in the two measures are nonzero are the ones for which $s_1 \in \mathcal{F}_{t-1}$, of which there are $(t-1) \cdot 2^{H-1}$ of them.

Lastly, taking expectations and summing over $t = 1$ to $T_{\max}$ and plugging in the value of $|\mathcal{S}_1| = 2^{2H}$ we have the final bound. $\qquad\square$

The next lemma is a straightforward modification of (Domingues et al., 2021, Lemma 5), with varying rewards instead of varying transitions.

**Lemma 7.** *Let $M$ and $M'$ be two MDPs that are identical in transition and differ in the reward distributions, denote $r_h(s,a)$ and $r'_h(s,a)$. Assume that for all $(s,a)$ we have $r_h(s,a) \ll r'_h(s,a)$. Then for any stopping time $\eta$ with respect to $(\mathcal{F}^t)_{t \geq 1}$ that satisfies $\mathbb{P}_M[\eta < \infty] = 1$,*

$$\mathrm{KL}\left( \mathbb{P}_M^{I_\eta} \parallel \mathbb{P}_{M'}^{I_\eta} \right) = \sum_{s \in \mathcal{S}, a \in \mathcal{A}, h \in [H]} \mathbb{E}_M[N_{s,a,h}^\eta] \cdot \mathrm{KL}(r_h(s,a) \| r'_h(s,a)),$$

*where $N_{s,a,h}^\eta := \sum_{t=1}^{\eta} \mathbb{1}\left\{ (S_h^{(t)}, A_h^{(t)}) = (s,a) \right\}$ and $I_\eta : \Omega \mapsto \bigcup_{t \geq 1} \mathcal{I}_t : \omega \mapsto I_{\eta(\omega)}(\omega)$ is the random vector representing the history up to episode $\eta$.*

**Lemma 8** (Lemma 1, Garivier et al. (2019)). *Consider a measurable space $(\Omega, \mathcal{F})$ equipped with two distributions $\mathbb{P}_1$ and $\mathbb{P}_2$. For any $\mathcal{F}$-measurable function $Z : \Omega \mapsto [0,1]$ we have*

$$\mathrm{KL}(\mathbb{P}_1 \| \mathbb{P}_2) \geq \mathrm{kl}(\mathbb{E}_1[Z] \| \mathbb{E}_2[Z]).$$

# F   Proofs for Section 6

## F.1   Algorithm Sketch

In this section we provide a high level sketch of Algorithm 1. We will describe the key algorithmic ideas, as well as the empirical estimation of policy-specific MRPs, in the two phases of POPLER.

**State Identification Phase**

The goal of the state identification phase is to discover all such petal states that are reachable with probability $\Omega(\varepsilon/D)$. The algorithm proceeds in a loop and sequentially grows the set $\mathcal{T}$, which contains tuples of the form $(s, \pi_s)$, where $s \in \bigcup_{\pi \in \Pi} \mathcal{S}_\pi$ is a sufficiently reachable petal state (for some policy) and $\pi_s$ denotes a policy that reaches $s$ with probability $\Omega(\varepsilon/D)$. We also denote $\mathcal{S}^{\mathrm{rch}} := \{s : (s, \pi_s) \in \mathcal{T}\}$ to denote the set of reachable states in $\mathcal{T}$. Initially, $\mathcal{T}$ only contains a dummy start state $s_\top$ and a null policy. We will collect data using the DataCollector subroutine that: for a given $(s, \pi_s) \in \mathcal{T}$, first run $\pi_s$ to reach state $s$, and if we succeed in reaching $s$, restart exploration by sampling a policy from $\mathrm{Uniform}(\Pi_{\exp})$. Note that DataCollector will be sample-efficient for any $(s, \pi_s)$ since $\Omega(\varepsilon/D)$ fraction of the trajectories obtained via $\pi_s$ are guaranteed to reach $s$ (by definition of $\pi_s$ and construction of set $\mathcal{T}$). Initially, we run DataCollector using $\mathrm{Uniform}(\Pi_{\exp})$ from the start, where we slightly abuse the notation and assume that all trajectories in the MDP start at the dummy state $s_\top$ at time step $h = 0$.

In every loop, the algorithm attempts to find a new petal state $\bar{s}$ for some $\pi \in \Pi$ that is guaranteed to be $\Omega(\varepsilon/D)$-reachable by $\pi$. This is accomplished by constructing a (estimated and partial) version of the policy-specific MRP using the datasets collected up until that loop (line 9). In particular given a policy $\pi$ and a set $\mathcal{S}^{\mathrm{rch}}_\pi = \mathcal{S}^+_\pi \cap \mathcal{S}^{\mathrm{rch}}$, we construct $\widehat{\mathfrak{M}}^\pi_{\mathcal{S}^{\mathrm{rch}}} = \mathrm{MRP}(\mathcal{S}^+_\pi, \widehat{P}^\pi, \widehat{R}^\pi, H, s_\top, s_\perp)$ which essentially compresses our empirical knowledge of the original MDP relevant to the policy $\pi$. In particular, for any states $s \in \mathcal{S}_\pi \cup \{s_\top\}$ and $s' \in \mathcal{S}_\pi \cup \{s_\perp\}$ residing in different layers $h < h'$ in the underlying MDP, we define:

- **Transition $\widehat{P}^\pi_{s \to s'}$** as:

$$\widehat{P}^\pi_{s \to s'} = \frac{1}{|\mathcal{D}_s|} \sum_{\tau \in \mathcal{D}_s} \frac{\mathbb{1}\{\pi \rightsquigarrow \tau_{h:h'}\}}{\frac{1}{|\Pi_{\mathrm{core}}|} \sum_{\pi' \in \Pi_{\mathrm{core}}} \mathbb{1}\{\pi' \rightsquigarrow \tau_{h:h'}\}} \mathbb{1}\left\{ \begin{array}{c} \tau_{h:h'} \text{ goes from } s \text{ to } s' \\ \text{without passing through any other } s'' \in \mathcal{S}_\pi \end{array} \right\}. \tag{14}$$

- **Transition $\widehat{R}^\pi_{s \to s'}$** as:

$$\widehat{R}^\pi_{s \to s'} = \frac{1}{|\mathcal{D}_s|} \sum_{\tau \in \mathcal{D}_s} \frac{R(\tau_{h:h'}) \cdot \mathbb{1}\{\pi \rightsquigarrow \tau_{h:h'}\}}{\frac{1}{|\Pi_{\mathrm{core}}|} \sum_{\pi' \in \Pi_{\mathrm{core}}} \mathbb{1}\{\pi' \rightsquigarrow \tau_{h:h'}\}} \mathbb{1}\left\{ \begin{array}{c} \tau_{h:h'} \text{ goes from } s \text{ to } s' \\ \text{without passing through any other } s'' \in \mathcal{S}_\pi \end{array} \right\}. \tag{15}$$

Clearly, the above definition implies that $\widehat{P}^\pi_{s \to s'} = 0$ and $\widehat{R}^\pi_{s \to s'} = 0$ for any $s \notin \mathcal{S}^{\mathrm{rch}}_\pi$ since $\mathcal{D}_s$ would be empty corresponding to these unexplored states. Furthermore, $P^\pi_{s_\perp \to s_\perp} = 1$ and $R^\pi_{s_\perp \to s_\perp} = 0$.

Note that since the set $\mathcal{S}^{\mathrm{rch}}$ is changing in each iteration of the loop as the algorithm collects more data, the policy-specific MRP $\widehat{\mathfrak{M}}^\pi_{\mathcal{S}^{\mathrm{rch}}}$ also changes in every iteration of the loop — in particular, more and more transitions/rewards are assigned nonzero values due to new states being added to $\mathcal{S}^{\mathrm{rch}}$. More details on policy-specific MRPs is given in Appendix F.2.

The key advantage of constructing the empirical versions of policy-specific MRPs is that they allow us to explore and find new leaf states in $\mathcal{S}_\pi$ which are reachable with probability at least $\Omega(\varepsilon/6D)$. In particular, using standard dynamic programming (subroutine EstReachability), we can check whether a candidate petal $\bar{s}$ is reachable with decent probability by $\pi$ (lines 11-12); If it is, then we add $(\bar{s}, \pi)$ to the set $\mathcal{T}$ and collect a fresh dataset using DataCollector (lines 13-14). Crucially, the importance sampling technique enables us to be sample efficient, since the same dataset $\mathcal{D}_s$ can be used to evaluate transitions/rewards in Eqs. (14) and (15) for multiple $\pi \in \Pi$ for which $s$ is a petal state. Furthermore, the number of such datasets we collect must be bounded—each $(s, \pi_s) \in \mathcal{T}$ contributes $\Omega(\varepsilon/D)$ to cumulative reachability, but since cumulative reachability is bounded from above by $\mathfrak{C}(\Pi)$ (Lemma 1), we know that $|\mathcal{T}| \leq \mathcal{O}(D \cdot \mathfrak{C}(\Pi)/\varepsilon)$.

**Evaluation Phase**

Next, POPLER moves to the evaluation phase. Using the collected data, it executes the Evaluate subroutine for every $\pi \in \Pi$ to get estimates $\widehat{V}^\pi$ (line 21) corresponding to $V^\pi$. For a given $\pi \in \Pi$, the Evaluate subroutine also constructs an empirical policy-specific MRP $\widehat{\mathfrak{M}}^\pi_{\mathcal{S}_{\mathrm{rch}}}$ for every $\pi \in \Pi$ and computes the value of $\pi$ via dynamic programming on $\widehat{\mathfrak{M}}^\pi_{\mathcal{S}_{\mathrm{rch}}}$. While the returned estimate $\widehat{V}^\pi$ is biased, in the complete proof, we will show that the bias is negligible since it is now only due to the states in the petal $\mathcal{S}_\pi$ which are *not* $\Omega(\varepsilon/D)$-reachable. Thus, we can guarantee that $\widehat{V}^\pi$ closely estimates $V^\pi$ for every $\pi \in \Pi$, and therefore POPLER returns a near-optimal policy.

### F.2 Algorithmic Details and Preliminaries

In this subsection, we provide the details of the subroutines that do not appear in the main body, in Algorithms 3, 4, and 5. The transitions and reward functions in line 5 in Algorithm 5 are computed using Eqs. (19) and (20), which are specified below, after introducing additional notation.

---

**Algorithm 3** DataCollector

---

**Require:** State: $s$, Reacher policy: $\pi_s$, Exploration policy set: $\Pi_{\mathrm{core}}$, Number of samples: $n$.
1: **if** $s = s_\top$ **then**                                        // Uniform sampling for start state $s_\top$
2:     **for** $t = 1, \ldots, n$ **do**
3:         Sample $\pi' \sim \mathrm{Uniform}(\Pi_{\mathrm{core}})$, and run $\pi'$ to collect $\tau = (s_1, a_1, \cdots, s_H, a_H)$.
4:         $\mathcal{D}_s \leftarrow \mathcal{D}_s \cup \{\tau\}$.
5:     **end for**
6: **else**                                        // $\pi_s$-based sampling for all other states $s \neq s_\top$
7:     Identify the layer $h$ such that $s \in \mathcal{S}_h$.
8:     **for** $t = 1, \ldots, n$ **do**
9:         Run $\pi_s$ for the first $h - 1$ time steps, and collect trajectory $(s_1, a_1, \cdots, s_{h-1}, a_{h-1}, s_h)$.
10:         **if** $s_h = s$ **then**
11:             Sample $\pi' \sim \mathrm{Uniform}(\Pi_{\mathrm{core}})$, and run $\pi'$ to collect remaining $(s_h, a_h, \cdots, s_H, a_H)$.
12:             $\mathcal{D}_s \leftarrow \mathcal{D}_s \cup \{\tau = (s_1, a_1, \cdots, s_H, a_H)\}$.
13:         **end if**
14:     **end for**
15: **end if**
16: **Return** dataset $\mathcal{D}_s$.

---

**Algorithm 4** EstReachability

---

**Require:** State space $S^{\mathrm{tab}}$, MRP $\mathfrak{M}$, State $\bar{s} \in \mathcal{S}^{\mathrm{tab}}$.
1: Let $P$ be the transition of $\mathfrak{M}$.
2: Initialize $V(s) = \mathbb{1}\{s = \bar{s}\}$ for all $s \in S^{\mathrm{tab}}$.
3: **Repeat** $H + 1$ times:
4:     For all $s \in S^{\mathrm{tab}}$, calculate $V(s) \leftarrow \sum_{s' \in \mathcal{S}^{\mathrm{tab}}} P_{s \to s'} \cdot V(s')$.      // Dynamic Programming
5: **Return** $V(s_\top)$.

---

We recall the definition of petals and sunflowers given in the main body (in Definitions 3 and 4). In the rest of this section, we assume that $\Pi$ is a $(K, D)$-sunflower with $\Pi_{\mathrm{core}}$ and $\mathcal{S}_\pi$ for any $\pi \in \Pi$.

**Definition 6** (Petals and Sunflowers (Definitions 3 and 4 in the main body))**.** *For a policy set $\bar{\Pi}$, and states $\bar{\mathcal{S}} \subseteq \mathcal{S}$, a policy $\pi$ is said to be a $\bar{\mathcal{S}}$-petal on $\bar{\Pi}$ if for all $h \leq h' \leq H$, and partial trajectories $\tau = (s_h, a_h, \cdots, s_{h'}, a_{h'})$ that are consistent with $\pi$: either $\tau$ is also consistent with some $\pi' \in \bar{\Pi}$, or there exists $i \in (h, h']$ s.t. $s_i \in \bar{\mathcal{S}}$.*

*A policy class $\Pi$ is said to be a $(K, D)$-sunflower if there exists a set $\Pi_{\mathrm{core}}$ of Markovian policies with $|\Pi_{\mathrm{core}}| \leq K$ such that for every policy $\pi \in \Pi$ there exists a set $\mathcal{S}_\pi \subseteq \mathcal{S}$, of size at most $D$, so that $\pi$ is an $\mathcal{S}_\pi$-petal on $\Pi_{\mathrm{core}}$.*

**Additional notation.**    Recall that we assumed that the state space $\mathcal{S} = \mathcal{S}_1 \times \ldots \mathcal{S}_H$ is layered. Thus, given a state $s$, we can infer the layer $h$ such that $s \in \mathcal{S}_h$. By definition $s_\top$ belongs to the layer $h = 0$ and $s_\perp$ belongs to the layer $h = H$. In the following, we define additional notation:

---

**Algorithm 5** Evaluate

---

**Require:** Policy set $\Pi_{\text{core}}$, Reachable states $\mathcal{S}^{\text{rch}}$, Datasets $\{\mathcal{D}_s\}_{s \in \mathcal{S}^{\text{rch}}}$, Policy $\pi$ to be evaluated.

1: Compute $\mathcal{S}^{\text{rch}}_\pi \leftarrow \mathcal{S}^+_\pi \cap \mathcal{S}^{\text{rch}}$ and $S^{\text{tab}} = \mathcal{S}^{\text{rch}}_\pi \cup \{s_\perp\}$.
2: **for** $s, s'$ in $S^{\text{tab}}$ **do**                                  // Compute transitions and rewards on $S^{\text{tab}}$
3:     Let $h, h'$ be such that $s \in \mathcal{S}_h$ and $s' \in \mathcal{S}_{h'}$
4:     **if** $h < h'$ **then**
5:         Calculate $\widehat{P}^\pi_{s \to s'}, \widehat{r}^\pi_{s \to s'}$ according to (19) and (20);
6:     **else**
7:         Set $\widehat{P}^\pi_{s \to s'} \leftarrow 0, \widehat{r}^\pi_{s \to s'} \leftarrow 0$.
8:     **end if**
9: **end for**
10: Set $\widehat{V}(s) = 0$ for all $s \in S^{\text{tab}}$.
11: **Repeat** for $H + 1$ times:                                  // Evaluate $\pi$ by dynamic programming
12:     For all $s \in S^{\text{tab}}$, calculate $\widehat{V}(s) \leftarrow \sum_{S^{\text{tab}}} \widehat{P}^\pi_{s \to s'} \cdot \left( \widehat{r}^\pi_{s \to s'} + \widehat{V}(s') \right)$.
13: **Return** $\widehat{V}(s_\top)$.

---

($a$) *Sets* $\mathfrak{T}(s \to s'; \neg \bar{\mathcal{S}})$: For any set $\bar{\mathcal{S}}$, and states $s, s' \in \mathcal{S}$, we define $\mathfrak{T}(s \to s'; \neg \bar{\mathcal{S}})$ as the set of all the trajectories that go from $s$ to $s'$ without passing through any state in $\bar{\mathcal{S}}$ in between.

More formally, let state $s$ be at layer $h$, and $s'$ be at layer $h'$. Then, $\mathfrak{T}(s \to s'; \neg \bar{\mathcal{S}})$ denotes the set of all the trajectories $\tau = (s_1, a_1, \cdots, s_H, a_H)$ that satisfy all of the following:

- $s_h = s$, where $s_h$ is the state at timestep $h$ in $\tau$.
- $s_{h'} = s'$, where $s_{h'}$ is the state at timestep $h'$ in $\tau$.
- For all $h < \widetilde{h} < h'$, the state $s_{\widetilde{h}}$, at time step $\widetilde{h}$ in $\tau$, does not lie in the set $\bar{\mathcal{S}}$.

Note that when $h' \leq h$, we define $\mathfrak{T}(s \to s'; \neg \bar{\mathcal{S}}) = \emptyset$. Additionally, we define $\mathfrak{T}(s_\top \to s; \neg \bar{\mathcal{S}})$ as the set of all trajectories that go to $s'$ (from a start state) without going through any state in $\bar{\mathcal{S}}$ in between. Finally, we define $\mathfrak{T}(s \to s_\perp; \neg \bar{\mathcal{S}})$ as the set of all the trajectories that go from $s$ at time step $h$ to the end of the episode without passing through any state in $\bar{\mathcal{S}}$ in between.

Furthermore, we use the shorthand $\mathfrak{T}_\pi(s \to s') := \mathfrak{T}(s \to s'; \neg \mathcal{S}_\pi)$ to denote the set of all the trajectories that go from $s$ to $s'$ without passing though any leaf state $\mathcal{S}_\pi$.

($b$) Using the above notation, for any $s \in \mathcal{S}$ and set $\bar{\mathcal{S}} \subseteq \mathcal{S}$, we define $\bar{d}^\pi(s; \neg \bar{\mathcal{S}})$ as the probability of reaching $s$ (from a start state) without passing through any state in $\bar{\mathcal{S}}$ in between, i.e.

$$\bar{d}^\pi(s; \neg \bar{\mathcal{S}}) = \mathbb{P}^\pi \left[ \tau \text{ reaches } s \text{ without passing through any state in } \bar{\mathcal{S}} \text{ before reaching } s \right]$$
$$= \mathbb{P}^\pi \left[ \tau \in \mathfrak{T}(s_\top \to s; \neg \bar{\mathcal{S}}) \right].$$

$$(16)$$

We next recall the notation of Markov Reward Process and formally define both the population versions of policy-specific MRPs.

**Markov Reward Process (MRP).** A Markov reward process $\mathfrak{M} = \text{MRP}(\mathcal{S}, P, R, H, s_\top, s_\perp)$ is defined over the state space $\mathcal{S}$ with start state $s_\top$ and end state $s_\perp$, for trajectory length $H + 2$. Without loss of generality, we assume that $\{s_\top, s_\perp\} \in \mathcal{S}$. The transition kernel is denoted by $P : \mathcal{S} \times \mathcal{S} \to [0, 1]$, such that for any $s \in \mathcal{S}$, $\sum_{s'} P_{s \to s'} = 1$; the reward kernel is denoted by $R : \mathcal{S} \times \mathcal{S} \to \Delta([0, 1])$. Throughout, we use the notation $\to$ to signify that the transitions and rewards are defined along the edges of the MRP.

A trajectory in $\mathfrak{M}$ is of the form $\tau = (s_\top, s_1, \cdots, s_H, s_\perp)$, where $s_h \in \mathcal{S}$ for all $h \in [H]$. Furthermore, from any state $s \in \mathcal{S}$, the MRP transitions[7] to another state $s' \in \mathcal{S}$ with probability $P_{s \to s'}$, and

---

[7]Our definition of Markov Reward Processes (MRP) deviates from MDPs that we considered in the paper, in the sense that we do not assume that the state space $\mathcal{S}$ is layered in an MRP. This variation is only adapted to simplify the proofs and the notation in the rest of the paper.

obtains the rewards $r_{s \to s'} \sim R_{s \to s'}$. Thus,

$$\mathbb{P}^{\mathfrak{M}}[\tau] = P_{s_\top \to s_1} \cdot \left( \prod_{h=1}^{H-1} P_{s_h \to s_{h+1}} \right) \cdot P_{s_H \to s_\perp},$$

and the rewards

$$R^{\mathfrak{M}}(\tau) = r_{s_\top \to s_1} + \sum_{h=1}^{H} r_{s_h \to s_{h+1}} + r_{s_H \to s_\perp}.$$

Furthermore, in all the MRPs that we consider in the paper, we have $P_{s_\perp \to s_\perp} = 1$ and $r_{s_\perp \to s_\perp} = 0$.

**Policy-Specific Markov Reward Processes.** A key technical tool in our analysis will be policy-specific MRPs that depend on the set $\mathcal{S}^{\mathrm{rch}}$ of the states that we have explored so far. Recall that for any policy $\pi$, $\mathcal{S}_\pi^+ = \mathcal{S}_\pi \cup \{s_\top, s_\perp\}$, $\mathcal{S}_\pi^{\mathrm{rch}} = \mathcal{S}_\pi^+ \cap \mathcal{S}^{\mathrm{rch}}$ and $\mathcal{S}_\pi^{\mathrm{rem}} = \mathcal{S}_\pi^+ \setminus (\mathcal{S}_\pi^{\mathrm{rch}} \cup \{s_\perp\})$. We define the expected and the empirical versions of policy-specific MRPs below; see Figure 2 for an illustration.

(a) **Expected Version of Policy-Specific MRP.** We define $\mathfrak{M}_{\mathcal{S}^{\mathrm{rch}}}^\pi = \mathrm{MRP}(\mathcal{S}_\pi^+, P^\pi, r^\pi, H, s_\top, s_\perp)$ where

- *Transition Kernel $P^\pi$:* For any $s \in \mathcal{S}_\pi^{\mathrm{rch}}$ and $s' \in \mathcal{S}_\pi^+$, we have

$$P_{s \to s'}^\pi = \mathbb{E}^\pi \left[ \mathbb{1}\{\tau \in \mathfrak{T}_\pi(s \to s')\} \big| s_h = s \right], \qquad (17)$$

where the expectation above is w.r.t. the trajectories drawn using $\pi$ in the underlying MDP, and $h$ denotes the time step such that $s \in \mathcal{S}_h$ (again, in the underying MDP). Thus, the transition $P_{s \to s'}^\pi$ denotes the probability of taking policy $\pi$ from $s$ and directly transiting to $s'$ without visiting any other states in $\mathcal{S}_\pi$. Furthermore, $P_{s \to s'}^\pi = \mathbb{1}\{s' = s_\perp\}$ for all $s \in \mathcal{S}_\pi^{\mathrm{rem}} \cup \{s_\perp\}$.

- *Reward Kernel $r^\pi$:* For any $s \in \mathcal{S}_\pi^{\mathrm{rch}}$ and $s' \in \mathcal{S}_\pi^+$, we have

$$r_{s \to s'}^\pi := \mathbb{E}^\pi \left[ R(\tau_{h:h'}) \mathbb{1}\{\tau \in \mathfrak{T}_\pi(s \to s')\} \big| s_h = s \right], \qquad (18)$$

where $R(\tau_{h:h'})$ denotes the reward for the partial trajectory $\tau_{h:h'}$ in the underlying MDP. The reward $r_{s \to s'}^\pi$ denotes the expectation of rewards collected by taking policy $\pi$ from $s$ and directly transiting to $s'$ without visiting any other states in $\mathcal{S}_\pi$. Furthermore, $r_{s \to s'}^\pi = 0$ for all $s \in \mathcal{S}_\pi^{\mathrm{rem}} \cup \{s_\perp\}$.

Throughout the analysis, we use $\mathbb{P}^{\mathfrak{M}}[\cdot] := \mathbb{P}^{\mathfrak{M}_{\mathcal{S}^{\mathrm{rch}}}^\pi}[\cdot]$ and $\mathbb{E}^{\mathfrak{M}}[\cdot] := \mathbb{E}^{\mathfrak{M}_{\mathcal{S}^{\mathrm{rch}}}^\pi}[\cdot]$ as a shorthand, whenever clear from the context.

(b) **Empirical Version of Policy-Specific MRPs.** Since the learner only has sampling access to the underlying MDP, it can not directly construct the MRP $\mathfrak{M}_{\mathcal{S}^{\mathrm{rch}}}^\pi$. Instead, in Algorithm 1, the learner constructs an empirical estimate for $\mathfrak{M}_{\mathcal{S}^{\mathrm{rch}}}^\pi$, defined as $\widehat{\mathfrak{M}}_{\mathcal{S}^{\mathrm{rch}}}^\pi = \mathrm{MRP}(\mathcal{S}_\pi^+, \widehat{P}^\pi, \widehat{r}^\pi, H, s_\top, s_\perp)$ where

- *Transition Kernel $\widehat{P}^\pi$:* For any $s \in \mathcal{S}_\pi^{\mathrm{rch}}$ and $s' \in \mathcal{S}_\pi^+$, we have

$$\widehat{P}_{s \to s'}^\pi = \frac{|\Pi_{\mathrm{core}}|}{|\mathcal{D}_s|} \sum_{\tau \in \mathcal{D}_s} \frac{\mathbb{1}\{\pi \rightsquigarrow \tau_{h:h'}\}}{\sum_{\pi' \in \Pi_{\mathrm{core}}} \mathbb{1}\{\pi' \rightsquigarrow \tau_{h:h'}\}} \mathbb{1}\{\tau \in \mathfrak{T}_\pi(s \to s')\}, \qquad (19)$$

where $\Pi_{\mathrm{core}}$ denotes the core of the sunflower corresponding to $\Pi$ and $\mathcal{D}_s$ denotes a dataset of trajectories collected via $\mathsf{DataCollector}(s, \pi_s, \Pi_{\mathrm{core}}, n_2)$. Furthermore, $\widehat{P}_{s \to s'}^\pi = \mathbb{1}\{s' = s_\perp\}$ for all $s \in \mathcal{S}_\pi^{\mathrm{rem}} \cup \{s_\perp\}$.

- *Reward Kernel $\widehat{r}^\pi$:* For any $s \in \mathcal{S}_\pi^{\mathrm{rch}}$ and $s' \in \mathcal{S}_\pi^+$, we have

$$\widehat{r}_{s \to s'}^\pi = \frac{|\Pi_{\mathrm{core}}|}{|\mathcal{D}_s|} \sum_{\tau \in \mathcal{D}_s} \frac{\mathbb{1}\{\pi \rightsquigarrow \tau_{h:h'}\}}{\sum_{\pi' \in \Pi_{\mathrm{core}}} \mathbb{1}\{\pi' \rightsquigarrow \tau_{h:h'}\}} \mathbb{1}\{\tau \in \mathfrak{T}_\pi(s \to s')\} R(\tau_{h:h'}), \qquad (20)$$

where $\Pi_{\mathrm{core}}$ denotes the core of the sunflower corresponding to $\Pi$, $\mathcal{D}_s$ denotes a dataset of trajectories collected via $\mathsf{DataCollector}(s, \pi_s, \Pi_{\mathrm{core}}, n_2)$, and $R(\tau_{h:h'}) = \sum_{i=h}^{h'-1} r_i$. Furthermore, $\widehat{r}_{s \to s'}^\pi = 0$ for all $s \in \mathcal{S}_\pi^{\mathrm{rem}}$.

The above approximates the MRP given by (14) and (15) in the main body.

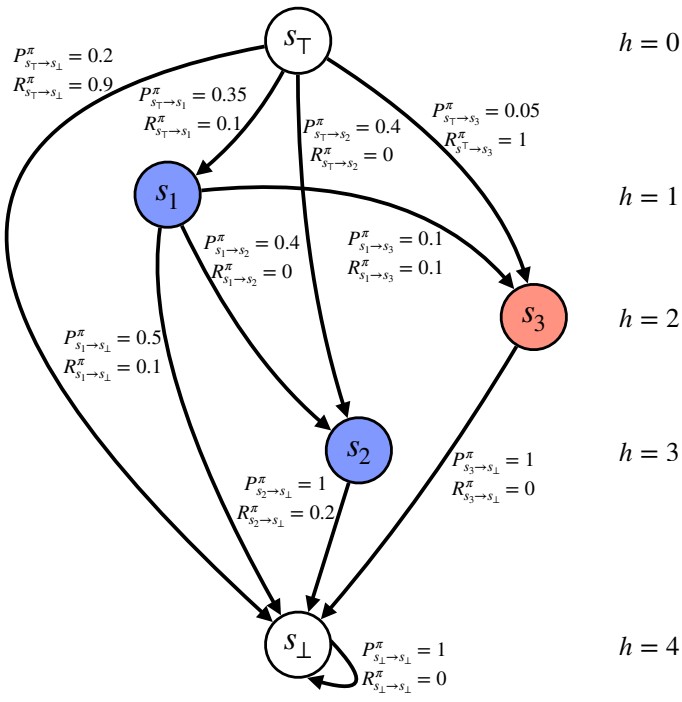

Figure 2: Illustration of an MRP $\mathfrak{M}^\pi_{\mathcal{S}^{\mathrm{rch}}}$ with $\mathcal{S}_\pi = \{s_1, s_2, s_3\}$ and $\mathcal{S}^{\mathrm{rch}} = \{s_1, s_2\}$. In the original MDP $M$, $s_1 \in \mathcal{S}_1$, $s_2 \in \mathcal{S}_3$, and $s_3 \in \mathcal{S}_2$. The edges are labeled with the values of $P^\pi_{s \to s'}$ and $R^\pi_{s \to s'}$. Notice that (1) there are no edges from $s_2 \to s_3$ or $s_3 \to s_2$ because trajectories cannot go from later layers to earlier ones; (2) since $s_3 \notin \mathcal{S}^{\mathrm{rch}}$, there is no edge from $s_3 \to s_2$, and instead we have $P^\pi_{s_3 \to s_\perp} = 1$ and $R^\pi_{s_3 \to s_\perp} = 0$; (3) $s_\perp$ is an absorbing state with no rewards.

**Properties of Trajectories in the Policy-Specific MRPs.** We state several properties of trajectories in the policy-specific MRPs, which will be used in the proofs. Let $\tau = (s_\top, s_1, \cdots, s_H, s_\perp)$ denote a trajectory from either $\mathfrak{M}^\pi_{\mathcal{S}^{\mathrm{rch}}}$ or $\widehat{\mathfrak{M}}^\pi_{\mathcal{S}^{\mathrm{rch}}}$.

(1) For some $k \leq H$ we have $s_1, \cdots, s_k \in \mathcal{S}_\pi$ and $s_{k+1} = \cdots = s_H = s_\perp$ (if $k = H$ we say the second condition is trivially met).

(2) Each state in $s_1, \cdots, s_k$ is unique.

(3) Letting $\mathsf{h}(s)$ denote the layer that a state $s \in \mathcal{S}_\pi$ is in, we have $\mathsf{h}(s_1) < \cdots < \mathsf{h}(s_k)$.

(4) Either (a) $s_1, \cdots, s_k \in \mathcal{S}^{\mathrm{rch}}_\pi$, or (b) $s_1, \cdots, s_{k-1} \in \mathcal{S}^{\mathrm{rch}}_\pi$ and $s_k \in \mathcal{S}^{\mathrm{rem}}_\pi$.

**Parameters Used in Algorithm 1.** Here, we list all the parameters that are used in Algorithm 1 and its subroutines:

$$n_1 = C_1 \frac{(D+1)^4 K^2 \log(|\Pi|(D+1)/\delta)}{\varepsilon^2},$$
$$n_2 = C_2 \frac{D^3(D+1)^2 K^2 \log(|\Pi|(D+1)^2/\delta)}{\varepsilon^3}, \tag{21}$$

where $C_1, C_2 > 0$ are absolute numerical constants, which will be specified later in the proofs.

### F.3 Supporting Technical Results

We start by stating the following variant of the classical simulation lemma (Kearns and Singh, 2002; Agarwal et al., 2019; Foster et al., 2021a).

**Lemma 9** (Simulation lemma (Foster et al., 2021a, Lemma F.3)). *Let* $\mathfrak{M} = (\mathcal{S}, P, r, H, s_\top, s_\perp)$ *be a markov reward process. Then, the empirical version* $\widehat{\mathfrak{M}} = (\mathcal{S}, \widehat{P}, \widehat{r}, H, s_\top, s_\perp)$ *corresponding to* $\mathfrak{M}$

*satisfies:*

$$|V_{\mathrm{MRP}} - \widehat{V}_{\mathrm{MRP}}| \leq \sum_{s \in \mathcal{S}} d_{\mathfrak{M}}(s) \cdot \left( \sum_{s' \in \mathcal{S}} |P_{s \to s'} - \widehat{P}_{s \to s'}| + |r_{s \to s'} - \widehat{r}_{s \to s'}| \right),$$

*where $d_{\mathfrak{M}}(s)$ is the probability of reaching $s$ under $\mathfrak{M}$, and $V_{\mathrm{MRP}}$ and $\widehat{V}_{\mathrm{MRP}}$ denotes the value of $s_\top$ under $\mathfrak{M}$ and $\widehat{\mathfrak{M}}$ respectively.*

The following technical lemma shows that for any policy $\pi$, the empirical version of policy-specific MRP closely approximates its expected version.

**Lemma 10.** *Let Algorithm 1 be run with the parameters given in Eq. (21), and consider any iteration of the while loop in line 5 with the instantaneous set $\mathcal{S}^{\mathrm{rch}}$. Further, suppose that $|\mathcal{D}_s| \geq \frac{\varepsilon n_2}{24 D}$ for all $s \in \mathcal{S}^{\mathrm{rch}}$. Then, with probability at least $1 - \delta$, the following hold:*

(a) *For all $\pi \in \Pi$, $s \in \mathcal{S}_\pi^{\mathrm{rch}}$ and $s' \in \mathcal{S}_\pi \cup \{s_\perp\}$,*

$$\max\left\{|P_{s \to s'}^\pi - \widehat{P}_{s \to s'}^\pi|, |r_{s \to s'}^\pi - \widehat{r}_{s \to s'}^\pi|\right\} \leq \frac{\varepsilon}{12 D(D + 1)}.$$

(b) *For all $\pi \in \Pi$ and $s' \in \mathcal{S}_\pi \cup \{s_\perp\}$,*

$$\max\{|P_{s_\top \to s'}^\pi - \widehat{P}_{s_\top \to s'}^\pi|, |r_{s_\top \to s'}^\pi - \widehat{r}_{s_\top \to s'}^\pi|\} \leq \frac{\varepsilon}{12(D + 1)^2}.$$

In the sequel, we define the event that the conclusion of Lemma 10 holds as $\mathcal{E}_{\mathrm{est}}$.

**Proof.** Fix any $\pi \in \Pi$. We first prove the bound for $s \in \mathcal{S}_\pi^{\mathrm{rch}}$. Let $s$ be at layer $h$. Fix any policy $\pi \in \Pi$, and consider any state $s' \in \mathcal{S}_\pi \cup \{s_\perp\}$, where $s'$ is at layer $h'$. Note that since $\Pi$ is a $(K, D)$-sunflower, with its core $\Pi_{\mathrm{core}}$ and petals $\{\mathcal{S}_\pi\}_{\pi \in \Pi}$, we must have that any trajectory $\tau \in \mathfrak{T}_\pi(s \to s')$ is also consistent with at least one $\pi_e \in \Pi_{\mathrm{core}}$. Furthermore, for any such $\pi_e$, we have

$$\mathbb{P}^{\pi_e}[\tau_{h:h'} \mid s_h = s] = \prod_{i=h}^{h'-1} P[s_{i+1} \mid s_i, \pi_e(s_i), s_h = s]$$

$$= \prod_{i=h}^{h'-1} P[s_{i+1} \mid s_i, \pi(s_i), s_h = s] = \mathbb{P}^\pi[\tau_{h:h'} \mid s_h = s], \tag{22}$$

where the second line holds because both $\pi \rightsquigarrow \tau_{h:h'}$ and $\pi_e \rightsquigarrow \tau_{h:h'}$. Next, recall from Eq. (17), that

$$P_{s \to s'}^\pi = \mathbb{E}^\pi[\mathbb{1}\{\tau \in \mathfrak{T}_\pi(s \to s')\} \mid s_h = s]. \tag{23}$$

Furthermore, from Eq. (19), recall that the empirical estimate $\widehat{P}_{s \to s'}^\pi$ of $P_{s \to s'}^\pi$ is given by :

$$\widehat{P}_{s \to s'}^\pi = \frac{1}{|\mathcal{D}_s|} \sum_{\tau \in \mathcal{D}_s} \frac{\mathbb{1}\{\tau \in \mathfrak{T}_\pi(s \to s')\}}{\frac{1}{|\Pi_{\mathrm{core}}|} \sum_{\pi_e \in \Pi_{\mathrm{core}}} \mathbb{1}\{\pi_e \rightsquigarrow \tau_{h:h'}\}}, \tag{24}$$

where the dataset $\mathcal{D}_s$ consists of i.i.d. samples, and is collected in lines 10-12 in Algorithm 3 (DataCollector), by first running the policy $\pi_s$ for $h$ timesteps and if the trajectory reaches $s$, then executing $\pi_e \sim \mathrm{Uniform}(\Pi_{\mathrm{core}})$ for the remaining time steps (otherwise this trajectory is rejected). Let the law of this process be $q$. We thus note that,

$$\mathbb{E}_{\tau \sim q}\left[\widehat{P}_{s \to s'}^\pi\right]$$

$$= \mathbb{E}_{\tau \sim q}\left[\frac{\mathbb{1}\{\tau \in \mathfrak{T}_\pi(s \to s')\}}{\frac{1}{|\Pi_{\mathrm{core}}|} \sum_{\pi_e \in \Pi_{\mathrm{core}}} \mathbb{1}\{\pi_e \rightsquigarrow \tau_{h:h'}\}} \mid s_h = s\right]$$

$$= \sum_{\tau \in \mathfrak{T}_\pi(s \to s')} \mathbb{P}_q[\tau_{h:h'} \mid s_h = s] \cdot \frac{1}{\frac{1}{|\Pi_{\mathrm{core}}|} \sum_{\pi_e \in \Pi_{\mathrm{core}}} \mathbb{1}\{\pi_e \rightsquigarrow \tau_{h:h'}\}}$$

$$\overset{(i)}{=} \sum_{\tau \in \mathfrak{T}_\pi(s \to s')} \frac{1}{|\Pi_{\text{core}}|} \sum_{\pi'_e \in \Pi_{\text{core}}} \mathbb{1}\{\tau \in \mathfrak{T}_{\pi'_e}(s \to s')\} \, \mathbb{P}^{\pi'_e}[\tau_{h:h'} \mid s_h = s] \cdot \frac{1}{\frac{1}{|\Pi_{\text{core}}|} \sum_{\pi_e \in \Pi_{\text{core}}} \mathbb{1}\{\pi_e \rightsquigarrow \tau_{h:h'}\}}$$

$$\overset{(ii)}{=} \sum_{\tau \in \mathfrak{T}_\pi(s \to s')} \frac{1}{|\Pi_{\text{core}}|} \sum_{\pi'_e \in \Pi_{\text{core}}} \mathbb{P}^\pi[\tau_{h:h'} \mid s_h = s] \cdot \frac{\mathbb{1}\{\pi'_e \rightsquigarrow \tau_{h:h'}\}}{\frac{1}{|\Pi_{\text{core}}|} \sum_{\pi_e \in \Pi_{\text{core}}} \mathbb{1}\{\pi_e \rightsquigarrow \tau_{h:h'}\}}$$

$$= \sum_{\tau \in \mathfrak{T}_\pi(s \to s')} \mathbb{P}^\pi[\tau_{h:h'} \mid s_h = s]$$

$$\overset{(iii)}{=} \mathbb{E}^\pi[\mathbb{1}\{\tau \in \mathfrak{T}_\pi(s \to s')\} \mid s_h = s] = P^\pi_{s \to s'},$$

where $(i)$ follows from the sampling strategy in Algorithm 3 after observing $s_h = s$, and $(ii)$ simply uses the relation (22) since both $\pi'_e \rightsquigarrow \tau_{h:h'}$ and $\pi \rightsquigarrow \tau_{h:h'}$ hold. Finally, in $(iii)$, we use the relation (23).

We have shown that $\widehat{P}^\pi_{s \to s'}$ is an unbiased estimate of $P^\pi_{s \to s'}$ for any $\pi$ and $s, s' \in \mathcal{S}^+_\pi$. Thus, using Hoeffding's inequality (Lemma 17), followed by a union bound, we get that with probability at least $1 - \delta/4$, for all $\pi \in \Pi$, $s \in \mathcal{S}^{\text{rch}}_\pi$, and $s' \in \mathcal{S}_\pi \cup \{s_\perp\}$,

$$|\widehat{P}^\pi_{s \to s'} - P^\pi_{s \to s'}| \le K \sqrt{\frac{2 \log(4|\Pi|D(D+1)/\delta)}{|\mathcal{D}_s|}},$$

where the additional factor of $K$ appears because for any $\tau \in \mathfrak{T}_\pi(s \to s')$, there must exist some $\pi_e \in \Pi_{\text{core}}$ that is also consistent with $\tau$ (as we showed above), which implies that each of the terms in Eq. (24) satisfies the bound a.s.:

$$\left| \frac{\mathbb{1}\{\tau \in \mathfrak{T}_\pi(s \to s')\}}{\frac{1}{|\Pi_{\text{core}}|} \sum_{\pi_e \in \Pi_{\text{core}}} \mathbb{1}\{\pi_e \rightsquigarrow \tau_{h:h'}\}} \right| \le |\Pi_{\text{core}}| = K.$$

Since $|\mathcal{D}_s| \ge \frac{\varepsilon n_2}{24D}$ by assumption, we have

$$|\widehat{P}^\pi_{s \to s'} - P^\pi_{s \to s'}| \le K \sqrt{\frac{48D \log(4|\Pi|D(D+1)/\delta)}{\varepsilon n_2}}.$$

Repeating a similar argument for the empirical reward estimation in Eq. (19), we get that with probability at least $1 - \delta/4$, for all $\pi \in \Pi$, and $s \in \mathcal{S}^{\text{rch}}_\pi$ and $s' \in \mathcal{S}_\pi \cup \{s_\perp\}$, we have that

$$|\widehat{r}^\pi_{s \to s'} - r^\pi_{s \to s'}| \le K \sqrt{\frac{48D \log(4|\Pi|D(D+1)/\delta)}{\varepsilon n_2}}.$$

Similarly, we can also get for any $\pi \in \Pi$ and $s' \in \mathcal{S}_\pi \cup \{s_\perp\}$, with probability at least $1 - \delta/2$,

$$\max\left\{ |\widehat{r}^\pi_{s_\top \to s'} - r^\pi_{s_\top \to s'}|, |\widehat{P}^\pi_{s_\top \to s'} - P^\pi_{s_\top \to s'}| \right\} \le K \sqrt{\frac{2 \log(4|\Pi|(D+1)/\delta)}{|\mathcal{D}_{s_\top}|}}$$

$$= K \sqrt{\frac{2 \log(4|\Pi|(D+1)/\delta)}{n_1}},$$

where the last line simply uses the fact that $|\mathcal{D}_{s_\top}| = n_1$. The final statement is due to a union bound on the above results. This concludes the proof of Lemma 10. $\square$

**Lemma 11.** *Fix a policy $\pi \in \Pi$ and a set of reachable states $\mathcal{S}^{\text{rch}}$, and consider the policy-specific MRP $\mathfrak{M}^\pi_{\mathcal{S}^{\text{rch}}}$ (as defined by Eqs. (17) and (18)). Then for any $s \in \mathcal{S}^{\text{rem}}_\pi$, the quantity $\bar{d}^\pi(s; \neg \mathcal{S}^{\text{rem}}_\pi) = d^{\mathfrak{M}}(s)$, where $d^{\mathfrak{M}}(s)$ is the occupancy of state $s$ in $\mathfrak{M}^\pi_{\mathcal{S}^{\text{rch}}}$.*

**Proof.** We use $\bar{\tau}$ to denote a trajectory in $\mathfrak{M}^\pi_{\mathcal{S}^{\text{rch}}}$ and $\tau$ to denote a "corresponding" (in a sense which will be described shortly) trajectory in the original MDP $M$. For any $s \in \mathcal{S}^{\text{rem}}_\pi$, we have

$$d^{\mathfrak{M}}(s) = \sum_{\bar{\tau} \text{ s.t. } s \in \bar{\tau}} \mathbb{P}^{\mathfrak{M}}[\bar{\tau}] = \sum_{k=0}^{H-1} \sum_{\bar{s}_1, \bar{s}_2, \cdots, \bar{s}_k \in \mathcal{S}^{\text{rch}}_\pi} \mathbb{P}^{\mathfrak{M}}[\bar{\tau} = (s_\top, \bar{s}_1, \cdots, \bar{s}_k, s, s_\perp, \cdots)].$$

The first equality is simply due to the definition of $d^{\mathfrak{M}}$. For the second equality, we sum up over all possible sequences which start at $s_{\top}$, pass through some (variable length) sequence of states $\bar{s}_1, \cdots, \bar{s}_k \in \mathcal{S}_\pi^{\mathrm{rch}}$, then reach $s$ and the terminal state $s_\perp$. By definition of the policy-specific MRP, we know that once the MRP transits to a state $s \in \mathcal{S}_\pi^{\mathrm{rem}}$, it must then transit to $s_\perp$ and repeat $s_\perp$ until the end of the episode.

Now fix a sequence $\bar{s}_1, \cdots, \bar{s}_k \in \mathcal{S}_\pi^{\mathrm{rch}}$. We relate the term in the summand to the probability of corresponding trajectories in the original MDP $M$. To avoid confusion, we let $s_{h_1}, \ldots, s_{h_k} \in \mathcal{S}_\pi^{\mathrm{rch}}$ denote the corresponding sequence of states in the original MDP, which are unique and satisfy $h_1 < h_2 < \cdots < h_k$. We also denote $s_{h_{k+1}} = s$.

Using the definition of $\mathfrak{M}_{\mathcal{S}^{\mathrm{rch}}}^{\pi}$, we write

$$
\mathbb{P}^{\mathfrak{M}}[\bar{\tau} = (s_{\top}, \bar{s}_1, \cdots, \bar{s}_k, s, s_\perp, \cdots)]
$$
$$
= \prod_{i=1}^{k} \mathbb{P}^{M,\pi}\left[\tau_{h_i:h_{i+1}} \in \mathfrak{T}_\pi(s_{h_i} \to s_{h_{i+1}}) \mid \tau[h_i] = s_{h_i}\right]
$$
$$
= \mathbb{P}^{M,\pi}[\forall i \in [k+1],\ \tau[h_i] = s_{h_i},\ \text{and}\ \forall h \in [h_{k+1}]\backslash\{h_1, \cdots, h_{k+1}\},\ \tau[h] \notin \mathcal{S}_\pi].
$$

Now we sum over all sequences $\bar{s}_1, \cdots, \bar{s}_k \in \mathcal{S}_\pi^{\mathrm{rch}}$ to get

$$
d^{\mathfrak{M}}(s)
$$
$$
= \sum_{k=0}^{H-1} \sum_{s_{h_1}, \cdots, s_{h_k} \in \mathcal{S}_\pi^{\mathrm{rch}}} \mathbb{P}^{M,\pi}[\forall i \in [k+1],\ \tau[h_i] = s_{h_i},\ \text{and}\ \forall h \in [h_{k+1}]\backslash\{h_1, \cdots, h_{k+1}\},\ \tau[h] \notin \mathcal{S}_\pi]
$$
$$
= \mathbb{P}^{M,\pi}[s \in \tau\ \text{and}\ \forall h \in [h_{k+1} - 1],\ \tau[h] \notin \mathcal{S}_\pi^{\mathrm{rem}}]
$$
$$
= \mathbb{P}^{\pi}[\tau \in \mathfrak{T}(s_{\top} \to s; \neg\mathcal{S}_\pi^{\mathrm{rem}})] = \bar{d}^{\pi}(s; \neg\mathcal{S}_\pi^{\mathrm{rem}}).
$$

The second equality follows from the definition of $\mathcal{S}_\pi^{\mathrm{rem}}$, and the last line is the definition of the $\bar{d}$ notation. This concludes the proof of Lemma 11. $\qquad\square$

**Lemma 12.** *With probability at least $1 - 2\delta$, any $(\bar{s}, \pi)$ that is added into $\mathcal{T}$ (in line 13 of Algorithm 1) satisfies $d^{\pi}(\bar{s}) \geq \varepsilon/12D$.*

**Proof.** For any $(\bar{s}, \pi) \in \mathcal{T}$, when we collect $\mathcal{D}_{\bar{s}}$ in Algorithm 3, the probability that a trajectory will be accepted (i.e. the trajectory satisfies the "if" statement in line 10) is exactly $d^{\pi}(\bar{s})$. Thus, using Hoeffding's inequality (Lemma 17), with probability at least $1 - \delta/D|\Pi|$,

$$
\left|\frac{|\mathcal{D}_{\bar{s}}|}{n_2} - d^{\pi}(\bar{s})\right| \leq \sqrt{\frac{2\log(D|\Pi|/\delta)}{n_2}}.
$$

Since $|\mathcal{T}| \leq D|\Pi|$, by union bound, the above holds for every $(\bar{s}, \pi) \in \mathcal{T}$ with probability at least $1 - \delta$. Let us denote this event as $\mathcal{E}_{\mathrm{data}}$. Under $\mathcal{E}_{\mathrm{data}}$, for any $(\bar{s}, \pi)$ that satisfies $d^{\pi}(\bar{s}) \geq \frac{\varepsilon}{12D}$,

$$
|\mathcal{D}_{\bar{s}}| \geq n_2 d^{\pi}(\bar{s}) - \sqrt{2n_2\log(D|\Pi|/\delta)} \geq \frac{\varepsilon n_2}{12D} - \frac{\varepsilon n_2}{24D} = \frac{\varepsilon n_2}{24D}, \tag{25}
$$

where the second inequality follows by the bound on $d^{\pi}(\bar{s})$ and our choice of parameter $n_2$ in Eq. (21).

In the following, we prove by induction that every $(\bar{s}, \pi)$ that is added into $\mathcal{T}$ in the while loop from lines 7-17 in Algorithm 1 satisfies $d^{\pi}(\bar{s}) \geq \frac{\varepsilon}{12D}$. This is trivially true at initialization when $\mathcal{T} = \{(s_{\top}, \mathrm{Null})\}$, since every trajectory starts at the dummy state $s_{\top}$, for which we have $d^{\mathrm{Null}}(s_{\top}) = 1$.

We now proceed to the induction hypothesis. Suppose that in some iteration of the while loop, every tuple $(\bar{s}, \pi) \in \mathcal{T}$ satisfies $d^{\pi}(\bar{s}) \geq \varepsilon/12D$, and that $(\bar{s}', \pi')$ is a new tuple that will be added to $\mathcal{T}$. We will show that $(\bar{s}', \pi')$ will also satisfy $d^{\pi'}(\bar{s}') \geq \varepsilon/12D$.

Recall that $\mathcal{S}_{\pi'}^{+} = \mathcal{S}_{\pi'} \cup \{s_{\top}, s_\perp\}$, $\mathcal{S}_{\pi'}^{\mathrm{rch}} = \mathcal{S}_{\pi'}^{+} \cap \mathcal{S}^{\mathrm{rch}}$, and $\mathcal{S}_{\pi'}^{\mathrm{rem}} = \mathcal{S}_{\pi'}^{+} \backslash \mathcal{S}_{\pi'}^{\mathrm{rch}}$. Let $\mathfrak{M}_{\mathcal{S}^{\mathrm{rch}}}^{\pi'} = \mathrm{MRP}(\mathcal{S}_{\pi'}^{+}, P^{\pi'}, r^{\pi'}, H, s_{\top}, s_\perp)$ be the policy-specific MRP, where $P^{\pi'}$ and $r^{\pi'}$ are defined in Eqs. (17) and (18) respectively for the policy $\pi'$. Similarly let $\widehat{\mathfrak{M}}_{\mathcal{S}^{\mathrm{rch}}}^{\pi'} =$

$\mathrm{MRP}(\mathcal{S}_{\pi'}^+, \widehat{P}^{\pi'}, \widehat{r}^{\pi'}, H, s_\top, s_\bot)$ denote the estimated policy-specific MRP, where $\widehat{P}^{\pi'}$ and $r^{\pi'}$ are defined using (19) and (20) respectively. Note that for any state $s \in \mathcal{S}_{\pi'}^{\mathrm{rch}}$, the bound in (25) holds.

For the rest of the proof, we assume that the event $\mathcal{E}_{\mathrm{est}}$, defined in Lemma 10, holds (this happens with probability at least $1 - \delta$). By definition of $\mathcal{E}_{\mathrm{est}}$, we have

$$|P_{s \to s'}^{\pi'} - \widehat{P}_{s \to s'}^{\pi'}| \le \frac{\varepsilon}{12D(D+1)}, \qquad \text{for all} \qquad s' \in \mathcal{S}_{\pi'} \cup \{s_\bot\}. \tag{26}$$

Furthermore, note that $\widehat{d}^{\pi'}(\bar{s}') \leftarrow \mathsf{EstReachability}(\mathcal{S}_{\pi'}^+, \widehat{\mathfrak{M}}_{\mathcal{S}^{\mathrm{rch}}}^{\pi'}, \bar{s}')$ since in Algorithm 4 we start with $V(s) = \mathbb{1}\{s = \bar{s}'\}$. Furthermore, using Lemma 9, we have

$$
\begin{aligned}
|\widehat{d}^{\pi'}(\bar{s}') - d^{\mathfrak{M}}(\bar{s}')| &\le (D+1) \sup_{s \in \mathcal{S}_{\pi'}^{\mathrm{rch}}, s' \in \mathcal{S}_{\pi'} \cup \{s_\bot\}} |\widehat{P}_{s \to s'}^{\pi'} - P_{s \to s'}^{\pi'}| \\
&\le \frac{\varepsilon}{12D(D+1)} \cdot (D+1) = \frac{\varepsilon}{12D}.
\end{aligned}
\tag{27}
$$

where the second inequality follows from (26). Additionally, Lemma 11 states that $d^{\mathfrak{M}}(\bar{s}') = \bar{d}^{\pi'}(\bar{s}'; \neg \mathcal{S}_{\pi'}^{\mathrm{rem}})$. Therefore we obtain

$$|\bar{d}^{\pi'}(\bar{s}'; \neg \mathcal{S}_{\pi'}^{\mathrm{rem}}) - \widehat{d}^{\pi'}(\bar{s}')| \le \frac{\varepsilon}{12D}.$$

Thus, if the new state-policy pair $(\bar{s}', \pi')$ is added into $\mathcal{T}$, we will have

$$\bar{d}^{\pi'}(\bar{s}'; \neg \mathcal{S}_{\pi'}^{\mathrm{rem}}) \ge \frac{\varepsilon}{6D} - \frac{\varepsilon}{12D} = \frac{\varepsilon}{12D}.$$

Furthermore, by definition of $\bar{d}$ we have

$$\bar{d}^{\pi'}(\bar{s}'; \neg \mathcal{S}_{\pi'}^{\mathrm{rem}}) = \mathbb{P}^{\pi'}[\tau \in \mathfrak{T}(s_\top \to \bar{s}'; \neg \mathcal{S}_{\pi'}^{\mathrm{rem}})] \le \mathbb{P}^{\pi'}[\bar{s}' \in \tau] = d^{\pi'}(\bar{s}'),$$

so we have proved the induction hypothesis $d^{\pi'}(\bar{s}') \ge \varepsilon/12D$ for the next round. This concludes the proof of Lemma 12.

$\square$

The next lemma establishes that Algorithm 1 will terminate after finitely many rounds, and that after termination will have explored all sufficiently reachable states.

**Lemma 13.** *With probability at least $1 - 2\delta$,*

(a) *The while loop in line 5 in Algorithm 1 will terminate after at most $\frac{12HD\mathfrak{C}(\Pi)}{\varepsilon}$ rounds.*

(b) *After the termination of the while loop, for any $\pi \in \Pi$, the remaining states $s \in \mathcal{S}_\pi^{\mathrm{rem}}$ that are not added to $\mathcal{S}^{\mathrm{rch}}$ satisfy $\bar{d}^\pi(s; \neg \mathcal{S}_\pi^{\mathrm{rem}}) \le \varepsilon/4D$.*

Notice that according to our algorithm, the same state cannot be added multiple times into $\mathcal{S}^{\mathrm{rch}}$. Therefore, $|\mathcal{S}^{\mathrm{rch}}| \le D|\Pi|$, and the maximum number of rounds of the while loop is $D|\Pi|$ (i.e., the while loop eventually terminates).

**Proof.** We prove each part separately.

(a) First, note that from the definition of coverability and Lemma 1, we have

$$\sum_{s \in \mathcal{S}} \sup_{\pi \in \Pi} d^\pi(s) \le HC^{\mathsf{cov}}(\Pi; M) \le H\mathfrak{C}(\Pi).$$

Furthermore, Lemma 12 states that every $(s, \pi_s) \in \mathcal{T}$ satisfies $d^{\pi_s}(s) \ge \varepsilon/12D$. Thus, at any point in Algorithm 1, we have

$$\sum_{s \in \mathcal{S}^{\mathrm{rch}}} \sup_{\pi \in \Pi} d^\pi(s) \ge \sum_{s \in \mathcal{S}^{\mathrm{rch}}} d^{\pi_s}(s) \ge |\mathcal{T}| \cdot \frac{\varepsilon}{12D}.$$

Since, $\mathcal{S}^{\mathrm{rch}} \subseteq \mathcal{S}$, the two bounds indicate that

$$|\mathcal{T}| \leq \frac{12HD\mathfrak{C}(\Pi)}{\varepsilon}.$$

Since every iteration of the while loop adds one new $(s, \pi_s)$ to $\mathcal{T}$, the while loop terminates after at most $12HD\mathfrak{C}(\Pi)/\varepsilon$ many rounds.

(b) We know that once the while loop has terminated, for every $\pi \in \Pi$ and $\bar{s} \in \mathcal{S}^{\mathrm{rem}}_\pi$, we must have $\widehat{d}^\pi(\bar{s}) \leq \varepsilon/6D$, or else the condition in line 12 in Algorithm 1 is violated.

Fix any such $(\bar{s}, \pi)$ pair. Inspecting the proof of Lemma 12, we see that

$$|\bar{d}^\pi(\bar{s}; \neg \mathcal{S}^{\mathrm{rem}}_\pi) - \widehat{d}^\pi(\bar{s})| \leq \frac{\varepsilon}{12D}.$$

To conclude, we get

$$\bar{d}^\pi(s; \neg \mathcal{S}^{\mathrm{rem}}_\pi) \leq \frac{\varepsilon}{6D} + \frac{\varepsilon}{12D} = \frac{\varepsilon}{4D}.$$

$\square$

**Lemma 14.** *Suppose that the conclusions of Lemmas 10 and 13 hold. Then for every $\pi \in \Pi$, the estimated value $\widehat{V}^\pi$ computed in Algorithm 1 satisfies*

$$|\widehat{V}^\pi - V^\pi| \leq \varepsilon.$$

**Proof.** We will break up the proof into two steps. First, we show that for any $\pi$, the value estimate $\widehat{V}^\pi$ obtained using the empirical policy-specific MRP $\widehat{\mathfrak{M}}^\pi_{\mathcal{S}^{\mathrm{rch}}}$ is close to its value in the policy-specific MRP $\mathfrak{M}^\pi_{\mathcal{S}^{\mathrm{rch}}}$, as defined via (17) and (18). We denote this quantity as $V^\pi_{\mathrm{MRP}}$. Then, we will show that $V^\pi_{\mathrm{MRP}}$ is close to $V^\pi$, the value of the policy $\pi$ in the original MDP $M$.

**Part 1: $\widehat{V}^\pi$ is close to $V^\pi_{\mathrm{MRP}}$.** Note that the output $\widehat{V}^\pi$ of Algorithm 5 is exact the value function of MRP $\widehat{\mathfrak{M}}^\pi_{\mathcal{S}^{\mathrm{rch}}}$ defined by Eqs. (19) and (20). When $D = 0$, by part (b) of Lemma 10, we obtain

$$|\widehat{V}^\pi - V^\pi_{\mathrm{MRP}}| = |\widehat{r}^\pi_{s_\top \to s_\perp} - r^\pi_{s_\top \to s_\perp}| \leq \frac{\varepsilon}{12(D+1)^2} \leq \frac{\varepsilon}{2}.$$

When $D \geq 1$, using Lemma 10, we have

$$|r^\pi_{s_\top \to s'} - \widehat{r}^\pi_{s_\top \to s'}| \leq \frac{\varepsilon}{12(D+1)^2}, \quad |P^\pi_{s_\top \to s'} - \widehat{P}^\pi_{s_\top \to s'}| \leq \frac{\varepsilon}{12(D+1)^2}, \quad \forall s' \in \mathcal{S}_\pi \cup \{s_\perp\},$$

$$|r^\pi_{s \to s'} - \widehat{r}^\pi_{s \to s'}| \leq \frac{\varepsilon}{12D(D+1)}, \quad |P^\pi_{s \to s'} - \widehat{P}^\pi_{s \to s'}| \leq \frac{\varepsilon}{12D(D+1)}, \quad \forall s \in \mathcal{S}^{\mathrm{rch}}_\pi, s' \in \mathcal{S}^+_\pi \cup \{s_\perp\}.$$

By the simulation lemma (Lemma 9), we get

$$|\widehat{V}^\pi - V^\pi_{\mathrm{MRP}}| \leq 2(D+2) \max_{s, s' \in \mathcal{S}^+_\pi} \left( \left| P^\pi_{s \to s'} - \widehat{P}^\pi_{s \to s'} \right| + |r^\pi_{s \to s'} - \widehat{r}^\pi_{s \to s'}| \right)$$

$$\leq 2(D+2) \left( \frac{\varepsilon}{12D(D+1)} + \frac{\varepsilon}{12D(D+1)} \right) \leq \frac{\varepsilon}{2}.$$

**Part 2 : $V^\pi_{\mathrm{MRP}}$ is close to $V^\pi$.** As in the proof of Lemma 11, let us consider different trajectories $\bar{\tau}$ that are possible in $\mathfrak{M}^\pi_{\mathcal{S}^{\mathrm{rch}}}$. We can represent $\bar{\tau} = (s_\top, \bar{s}_1, \cdots, \bar{s}_k, s_\perp, \cdots)$ where the states $\bar{s}_1, \cdots, \bar{s}_k$ are distinct and all except possibly $\bar{s}_k$ belong to $\mathcal{S}^{\mathrm{rch}}_\pi$, and the states after $s_\perp$ are just repeats of $s_\perp$ until the end of the episode. Let $s_{h_1}, s_{h_2}, \ldots, s_{h_k}$ be the same sequence (in the original MDP $M$) Again, we have

$$\mathbb{P}^{\mathfrak{M}}[\bar{\tau} = (s_\top, \bar{s}_1, \cdots, \bar{s}_k, s_\perp, \cdots)]$$
$$= \mathbb{P}^\pi[\forall i \in [k], \ \tau[h_i] = s_{h_i}, \text{ and } \forall h \in [H] \backslash \{h_1, \cdots, h_k\}, \ \tau[h] \notin \mathcal{S}_\pi],$$

where recall that $\mathbb{P}^{\mathfrak{M}}$ denotes probability under the $\mathfrak{M}^{\pi}_{\mathcal{S}^{\mathrm{rch}}}$, and $\mathbb{P}^{\pi}$ denotes the probability under trajectories drawn according to $\pi$ in the underlying MDP; $\mathbb{E}^{\mathfrak{M}}$ and $\mathbb{E}^{\pi}$ are defined similarly.

Furthermore, the expectation of rewards we collected in $\mathfrak{M}^{\pi}_{\mathcal{S}^{\mathrm{rch}}}$ with trajectories $\bar{\tau}$ is

$$\mathbb{E}^{\mathfrak{M}}[R[\bar{\tau}]\mathbb{1}\{(s_{\top}, \bar{s}_1, \cdots, \bar{s}_k, s_{\perp}, \cdots)\}]$$
$$=\mathbb{E}^{\pi}[R[\tau]\mathbb{1}\{\forall i \in [k],\ \tau[h_i] = s_{h_i},\ \text{and}\ \forall h \in [H]\backslash\{h_1, \cdots, h_k\},\ \tau[h] \notin \mathcal{S}_{\pi}\}].$$

Next, we sum over all possible trajectories. However, note that the only trajectories that are possible in $M$ whose corresponding trajectories are *not accounted for* in $\mathfrak{M}^{\pi}_{\mathcal{S}^{\mathrm{rch}}}$ are precisely those that visit states in $\mathcal{S}_{\pi}$, after visiting some $s_{h_k}$ in the remaining states $\mathcal{S}^{\mathrm{rem}}_{\pi}$ (since, by construction, the MRP transitions directly to $s_{\perp}$ after encountering a state in $\mathcal{S}^{\mathrm{rem}}_{\pi}$). Thus,

$$V^{\pi}_{\mathrm{MRP}} = \mathbb{E}^{\pi}[R[\tau](\mathbb{1}\{\tau \cap \mathcal{S}^{\mathrm{rem}}_{\pi} = \emptyset\} + \mathbb{1}\{\exists k \in [H] : s_{h_k} \in \mathcal{S}^{\mathrm{rem}}_{\pi}\ \text{and}\ \forall h > h_k : s_h \notin \mathcal{S}_{\pi}\})],$$

where the first term corresponds to trajectories that do not pass through $\mathcal{S}^{\mathrm{rem}}_{\pi}$, and the second term corresponds to trajectories that passes through some state in $\mathcal{S}^{\mathrm{rem}}_{\pi}$ but then does not go through any other state in $\mathcal{S}_{\pi}$. On the other hand,

$$V^{\pi} = \mathbb{E}^{\pi}[R[\tau]].$$

Clearly, $V^{\pi}_{\mathrm{MRP}} \leq V^{\pi}$. Furthermore, we also have

$$V^{\pi} - V^{\pi}_{\mathrm{MRP}} = \mathbb{E}^{\pi}[R[\tau]\mathbb{1}\{\tau \cap \mathcal{S}^{\mathrm{rem}}_{\pi} \neq \emptyset\} - \mathbb{1}\{\exists k \in [H] : s_{h_k} \in \mathcal{S}^{\mathrm{rem}}_{\pi}\ \text{and}\ \forall h > h_k : s_h \notin \mathcal{S}_{\pi}\}]$$
$$\leq \mathbb{E}^{\pi}[R[\tau]\mathbb{1}\{\tau \cap \mathcal{S}^{\mathrm{rem}}_{\pi} \neq \emptyset\}]$$
$$\leq D \cdot \frac{\varepsilon}{4D} = \frac{\varepsilon}{4},$$

where the first inequality follows by just ignoring the second indicator term, and the second inequality follows by taking a union bound over all possible values of $\mathcal{S}^{\mathrm{rem}}_{\pi}$ as well as the conclusion of Lemma 13.

Putting it all together, we get that

$$|\widehat{V}^{\pi} - V^{\pi}| \leq |V^{\pi} - V^{\pi}_{\mathrm{MRP}}| + |\widehat{V}^{\pi} - V^{\pi}_{\mathrm{MRP}}| \leq \frac{\varepsilon}{4} + \frac{\varepsilon}{2} < \varepsilon.$$

This concludes the proof of Lemma 14. $\qquad \square$

### F.4 Proof of Theorem 4

We assume the events defined in Lemmas 10, 12 and 13 hold (which happens with probability at least $1 - 2\delta$). With our choices of $n_1, n_2$ in Eq. (21), the total number of samples used in our algorithm is at most

$$n_1 + n_2 \cdot \frac{12HD\mathfrak{C}(\Pi)}{\varepsilon} = \widetilde{\mathcal{O}}\left(\left(\frac{1}{\varepsilon^2} + \frac{HD^6\mathfrak{C}(\Pi)}{\varepsilon^4}\right) \cdot K^2 \log \frac{|\Pi|}{\delta}\right).$$

After the termination of the while loop, we know that for any policy $\pi \in \Pi$ and $s \in \mathcal{S}^{\mathrm{rem}}_{\pi}$ we have

$$\bar{d}^{\pi}(s; \neg \mathcal{S}^{\mathrm{rem}}_{\pi}) \leq \frac{\varepsilon}{4D}.$$

Therefore, by Lemma 14, we know for every $\pi \in \Pi$, $|\widehat{V}^{\pi} - V^{\pi}| \leq \varepsilon$. Hence the output policy $\widehat{\pi} \in \arg\max_{\pi} \widehat{V}^{\pi}$ satisfies

$$\max_{\pi \in \Pi} V^{\pi} - V^{\widehat{\pi}} \leq 2\varepsilon + \widehat{V}^{\pi} - \widehat{V}^{\widehat{\pi}} \leq 2\varepsilon.$$

Rescaling $\varepsilon$ by $2\varepsilon$ and $\delta$ by $2\delta$ concludes the proof of Theorem 4. $\qquad \square$

### F.5 Sunflower Property is Insufficient By Itself

We give an example of a policy class $\Pi$ for which the sunflower property holds for $K, D = \mathrm{poly}(H)$ but $\mathfrak{C}(\Pi) = 2^H$. Therefore, in light of Theorem 2, the sunflower property by itself cannot ensure statistically efficient agnostic PAC RL in the online access model.

The example is as follows: Consider a binary tree MDP with $2^H - 1$ states and action space $\mathcal{A} = \{0, 1\}$. The policy class $\Pi$ will be able to get to every $(s, a)$ pair in layer $H$. To define the policies, we consider each possible trajectory $\tau = (s_1, a_1, \cdots, s_H, a_H)$ and let:

$$\Pi := \left\{ \pi_\tau : \pi_\tau(s) = \begin{cases} a_i & \text{if } s_i \in \tau, \\ 0 & \text{otherwise,} \end{cases} \right\}.$$

Thus it is clear that $\mathfrak{C}(\Pi) = 2^H$, but the sunflower property holds with $K = 1$, $D = H$ by taking $\Pi_{\text{core}} = \{\pi_0\}$ (the policy which always picks $a = 0$).

# G  Infinite Policy Classes

In this section we discuss the extensions of our results to infinite policy classes.

## G.1  Definitions and Preliminary Lemmas

We will state our results in terms of the Natarajan dimension, which is a generalization of the VC dimension used to study multiclass learning. We note that the results in this section could be stated in terms of other complexity measures from multiclass learning such as the graph dimension and DS dimension (see, e.g., Natarajan, 1989; Shalev-Shwartz and Ben-David, 2014; Daniely and Shalev-Shwartz, 2014; Brukhim et al., 2022a); for simplicity we analyze guarantees in terms of the Natarajan dimension.

**Definition 7** (Natarajan Dimension (Natarajan, 1989))**.** *Let $\mathcal{X}$ be an instance space and $\mathcal{Y}$ be a finite label space. Given a class $\mathcal{H} \subseteq \mathcal{Y}^{\mathcal{X}}$, we define its* Natarajan dimension*, denoted $\mathrm{Ndim}(\mathcal{H})$, to be the maximum cardinality of a set $C \subseteq \mathcal{X}$ that satisfies the following: there exists $h_0, h_1 : C \to \mathcal{Y}$ such that (1) for all $x \in C$, $h_0(x) \neq h_1(x)$, and (2) for all $B \subseteq C$, there exists $h \in \mathcal{H}$ such that for all $x \in B$, $h(x) = h_0(x)$ and for all $x \in C \backslash B$, $h(x) = h_1(x)$.*

A notation we will use throughout is the projection operator. For a hypothesis class $\mathcal{H} \subseteq \mathcal{Y}^{\mathcal{X}}$ and a finite set $X = (x_1, \cdots, x_n) \in \mathcal{X}^n$, we define the projection of $\mathcal{H}$ on to $X$ as

$$\mathcal{H}\big|_X := \{(h(x_1), \cdots, h(x_n)) : h \in \mathcal{H}\}.$$

**Lemma 15** (Sauer's Lemma for Natarajan Classes (Haussler and Long, 1995))**.** *Given a hypothesis class $\mathcal{H} \subseteq \mathcal{Y}^{\mathcal{X}}$ with $|\mathcal{Y}| = K$ and $\mathrm{Ndim}(\mathcal{H}) \leq d$, we have for every $X = (x_1, \cdots, x_n) \in \mathcal{X}^n$,*

$$\left|\mathcal{H}\big|_X\right| \leq \left(\frac{ne(K+1)^2}{2d}\right)^d.$$

**Theorem 6** (Multiclass Fundamental Theorem (Shalev-Shwartz and Ben-David, 2014))**.** *For any class $\mathcal{H} \subseteq \mathcal{Y}^{\mathcal{X}}$ with $\mathrm{Ndim}(\mathcal{H}) = d$ and $|\mathcal{Y}| = K$, the minimax sample complexity of $(\varepsilon, \delta)$ agnostic PAC learning $\mathcal{H}$ can be bounded as*

$$\Omega\left(\frac{d + \log(1/\delta)}{\varepsilon^2}\right) \leq n(\Pi; \varepsilon, \delta) \leq \mathcal{O}\left(\frac{d \log K + \log(1/\delta)}{\varepsilon^2}\right).$$

**Definition 8** (Pseudodimension)**.** *Let $\mathcal{X}$ be an instance space. Given a hypothesis class $\mathcal{H} \subseteq \mathbb{R}^{\mathcal{X}}$, its pseudodimension, denoted $\mathrm{Pdim}(\mathcal{H})$, is defined as $\mathrm{Pdim}(\mathcal{H}) := \mathrm{VC}(\mathcal{H}^+)$, where $\mathcal{H}^+ := \{(x, \theta) \mapsto \mathbb{1}\{h(x) \leq \theta\} : h \in \mathcal{H}\}$.*

**Definition 9** (Covering Numbers)**.** *Given a hypothesis class $\mathcal{H} \subseteq \mathbb{R}^{\mathcal{X}}$, $\alpha > 0$, and $X = (x_1, \cdots, x_n) \in \mathcal{X}^n$, the covering number $\mathcal{N}_1(\mathcal{H}, \alpha, X)$ is the minimum cardinality of a set $C \subset \mathbb{R}^n$ such that for any $h \in \mathcal{H}$ there exists a $c \in C$ such that $\frac{1}{n} \sum_{i=1}^n |h(x_i) - c_i| \leq \alpha$.*

**Lemma 16** (Jiang et al. (2017), see also Pollard (2012); Luc et al. (1996))**.** *Let $\mathcal{H} \subset [0, 1]^{\mathcal{X}}$ be a real-valued hypothesis class, and let $X = (x_1, \cdots, x_n)$ be i.i.d. samples drawn from some distribution $\mathcal{D}$ on $\mathcal{X}$. Then for any $\alpha > 0$*

$$\mathbb{P}\left[\sup_{h \in \mathcal{H}}\left|\frac{1}{n}\sum_{i=1}^n h(x_i) - \mathbb{E}[h(x)]\right| > \alpha\right] \leq 8\mathbb{E}[\mathcal{N}_1(\mathcal{H}, \alpha/8, X)] \cdot \exp\left(-\frac{n\alpha^2}{128}\right).$$

*Furthermore if $\mathrm{Pdim}(\mathcal{H}) \leq d$ then we have the bound*

$$\mathbb{P}\left[\sup_{h \in \mathcal{H}}\left|\frac{1}{n}\sum_{i=1}^n h(x_i) - \mathbb{E}[h(x)]\right| > \alpha\right] \leq 8e(d+1)\left(\frac{16e}{\alpha}\right)^d \cdot \exp\left(-\frac{n\alpha^2}{128}\right),$$

*which is at most $\delta$ as long as $n \geq \frac{128}{\alpha^2}\left(d \log \frac{16e}{\alpha} + \log(8e(d+1)) + \log \frac{1}{\delta}\right)$.*

## G.2  Generative Model Lower Bound

First we address the lower bound. Observe that it is possible to achieve a lower bound that depends on $\mathrm{Ndim}(\Pi)$ with the following construction. First, identify the layer $h \in [H]$ such that the witnessing

set $C$ contains the maximal number of states in $\mathcal{S}_h$; by pigeonhole principle there must be at least $\mathrm{Ndim}(\Pi)/H$ such states in layer $h$. Then, we construct an MDP which "embeds" a hard multiclass learning problem at layer $h$ over these states. A lower bound of $\Omega\left(\frac{\mathrm{Ndim}(\Pi)}{H\varepsilon^2} \cdot \log\frac{1}{\delta}\right)$ follows from Theorem 6.

By combining Theorem 2 with the above we get the following corollary.

**Corollary 3** (Lower Bound for Generative Model with Infinite Policy Classes). *For any policy class $\Pi$, the minimax sample complexity $(\varepsilon, \delta)$-PAC learning $\Pi$ is at least*

$$n_{\mathsf{gen}}(\Pi; \varepsilon, \delta) \geq \Omega\left(\frac{\mathfrak{C}(\Pi) + \mathrm{Ndim}(\Pi)/H}{\varepsilon^2} \cdot \log\frac{1}{\delta}\right).$$

Again, since the generative model setting is easier than online RL, this lower bound also extends to the online RL setting.

Our bound is additive in $\mathfrak{C}(\Pi)$ and $\mathrm{Ndim}(\Pi)$; we do not know if it is possible to strengthen this to be a product of the two factors, as we will achieve in the upper bound in the next section.

### G.3  Generative Model Upper Bound

For the upper bounds, we can replace the dependence on $\log|\Pi|$ with $\mathrm{Ndim}(\Pi)$ (and additional log factors). In particular, we can modify the analysis of the TrajectoryTree to account for infinite policy classes. Recall that our analysis of TrajectoryTree required us to prove a uniform convergence guarantee for the estimate $\widehat{V}^\pi$: with probability at least $1 - \delta$, for all $\pi \in \Pi$, we have $|\widehat{V}^\pi - V^\pi| \lesssim \varepsilon$. We previously used a union bound over $|\Pi|$, which gave us the $\log|\Pi|$ dependence. Now we sketch an argument to replace it with $\mathrm{Ndim}(\Pi)$.

Let $\mathcal{T}$ be the set of all possible trajectory trees. We introduce the notation $v^\pi : \mathcal{T} \to \mathbb{R}$ to denote the function that takes as input a trajectory tree $\widehat{T}$ (for example, as sampled by TrajectoryTree) and returns the value of running $\pi$ on it. Then we can rewrite the desired uniform convergence guarantee:

$$\text{w.p. at least } 1 - \delta, \quad \sup_{\pi \in \Pi} \left| \frac{1}{n} \sum_{i=1}^{n} v^\pi(\widehat{T}_i) - \mathbb{E}\left[v^\pi(\widehat{T})\right] \right| \leq \varepsilon. \tag{28}$$

In light of Lemma 16, we will compute the pseudodimension for the function class $\mathcal{V}^\Pi = \{v^\pi : \pi \in \Pi\}$. Define the subgraph class

$$\mathcal{V}^{\Pi,+} := \left\{ (\widehat{T}, \theta) \mapsto \mathbb{1}\left\{v^\pi(\widehat{T}) \leq \theta\right\} : \pi \in \Pi \right\} \subseteq \{0,1\}^{\mathcal{T} \times \mathbb{R}}$$

By definition, $\mathrm{Pdim}(\mathcal{V}^\Pi) = \mathrm{VC}(\mathcal{V}^{\Pi,+})$. Fix any $X = \left\{(\widehat{T}_1, \theta_1), \cdots, (\widehat{T}_d, \theta_d)\right\} \in (\mathcal{T} \times \mathbb{R})^d$. In order to show that $\mathrm{VC}(\mathcal{V}^{\Pi,+}) \leq d$ for some value of $d$ it suffices to prove that $\left|\mathcal{V}^{\Pi,+}\big|_X\right| < 2^d$.

For any index $t \in [d]$, we also denote $\pi(\vec{s}_i) \in \mathcal{A}^{\leq H\mathfrak{C}(\Pi)}$ to be the vector of actions selected by $\pi$ on all $\Pi$-reachable states in $\widehat{T}_i$ (of which there are at most $H \cdot \mathfrak{C}(\Pi)$). We claim that

$$\left|\mathcal{V}^{\Pi,+}\big|_X\right| \leq |\{(\pi(\vec{s}_1), \cdots, \pi(\vec{s}_d)) : \pi \in \Pi\}| =: \left|\Pi\big|_X\right|. \tag{29}$$

This is true because once the $d$ trajectory trees are fixed, for every $\pi \in \Pi$, the value of the vector $\mathcal{V}^{\{\pi\},+}\big|_X \in \{0,1\}^d$ is determined by the trajectory that $\pi$ takes in every trajectory tree. This in turn is determined by the assignment of actions to every reachable state in all the $d$ trajectory trees, of which there are at most $\mathfrak{C}(\Pi) \cdot H \cdot d$ of. Therefore, we can upper bound the size of $\mathcal{V}^{\Pi,+}\big|_X$ by the number of ways any $\pi \in \Pi$ assign actions to every state in $\widehat{T}_1, \cdots, \widehat{T}_d$.

Applying Lemma 15 to Eq. (29), we get that

$$\left|\mathcal{V}^{\Pi,+}\big|_X\right| \leq \left(\frac{H\mathfrak{C}(\Pi)d \cdot e \cdot (A+1)^2}{2\mathrm{Ndim}(\Pi)}\right)^{\mathrm{Ndim}(\Pi)}.$$

For the choice of $d = \widetilde{\mathcal{O}}(\mathrm{Ndim}(\Pi))$, the previous display is at most $2^d$, thus proving the bound on $\mathrm{Pdim}(\mathcal{V}^\Pi)$. Lastly, the bound can be plugged back into Lemma 16 to get a bound on the error of TrajectoryTree: the statement in Eq. (28) holds using

$$n = \widetilde{\mathcal{O}}\left( H\mathfrak{C}(\Pi) \cdot \frac{\mathrm{Ndim}(\Pi) + \log\frac{1}{\delta}}{\varepsilon^2} \right) \quad \text{samples.}$$

This in turn yields a guarantee on $\widehat{\pi}$ returned by TrajectoryTree.

### G.4 Online RL Upper Bound

The modified analysis for the online RL upper bound (Theorem 4) proceeds similarly; we sketch the ideas below.

There are two places in the proof of Theorem 4 which require a union bound over $|\Pi|$: the event $\mathcal{E}_{\mathrm{est}}$ (defined by Lemma 10) that the estimated transitions and rewards of the MRPs are close to their population versions, and the event $\mathcal{E}_{\mathrm{data}}$ (defined by Lemma 12) that the datasets collected are large enough. The latter is easy to address, since we can simply modify the algorithm's while loop to break after $\mathcal{O}\left(\frac{HD\mathfrak{C}(\Pi)}{\varepsilon}\right)$ iterations and union bound over the size of the set $|\mathcal{T}|$ instead of the worst-case bound on the size $D|\Pi|$. For $\mathcal{E}_{\mathrm{data}}$, we follow a similar strategy as the analysis for the generative model upper bound.

Fix a state $s$. Recall that the estimate for the probability transition kernel in the MDP in Eq. (19) takes the form

$$\widehat{P}^\pi_{s \to s'} = \frac{1}{|\mathcal{D}_s|} \sum_{\tau \in \mathcal{D}_s} \frac{\mathbb{1}\{\pi \rightsquigarrow \tau_{h:h'}\}}{\frac{1}{|\Pi_{\mathrm{core}}|} \sum_{\pi' \in \Pi_{\mathrm{core}}} \mathbb{1}\{\pi_e \rightsquigarrow \tau_{h:h'}\}} \mathbb{1}\{\tau \in \mathfrak{T}_\pi(s \to s')\}.$$

(The analysis for the rewards is similar, so we omit it from this proof sketch.)

We set up some notation. Define the function $p^\pi_{s \to s'} : (\mathcal{S} \times \mathcal{A} \times \mathbb{R})^H \to [0, |\Pi_{\mathrm{core}}|]$ as

$$p^\pi_{s \to s'}(\tau) := \frac{\mathbb{1}\{\pi \rightsquigarrow \tau_{h:h'}\}}{\frac{1}{|\Pi_{\mathrm{core}}|} \sum_{\pi' \in \Pi_{\mathrm{core}}} \mathbb{1}\{\pi_e \rightsquigarrow \tau_{h:h'}\}} \mathbb{1}\{\tau \in \mathfrak{T}_\pi(s \to s')\}, \tag{30}$$

with the implicit restriction of the domain to trajectories $\tau$ for which the denominator is nonzero. We have $\mathbb{E}[p^\pi_{s \to s'}(\tau)] = P^\pi_{s \to s'}$. Also let $\Pi_s = \{\pi \in \Pi : s \in \mathcal{S}_\pi\}$.

Restated in this notation, our objective is to show the uniform convergence guarantee

$$\text{w.p. at least } 1 - \delta, \quad \sup_{\pi \in \Pi_s, s' \in \mathcal{S}_\pi} \left| \frac{1}{|\mathcal{D}_s|} \sum_{\tau \in \mathcal{D}_s} p^\pi_{s \to s'}(\tau) - \mathbb{E}[p^\pi_{s \to s'}(\tau)] \right| \leq \varepsilon. \tag{31}$$

Again, in light of Lemma 16, we need to compute the pseudodimension for the function class $\mathcal{P}^{\Pi_s} = \{p^\pi_{s \to s'} : \pi \in \Pi_s, s' \in \mathcal{S}_\pi\}$, since these are all possible transitions that we might use the dataset $\mathcal{D}_s$ to evaluate. Define the subgraph class

$$\mathcal{P}^{\Pi_s,+} := \{(\tau, \theta) \mapsto \mathbb{1}\{p^\pi_{s \to s'}(\tau) \leq \theta\} : \pi \in \Pi_s, s' \in \mathcal{S}_\pi\}.$$

Fix the set $X = \{(\tau_1, \theta_1), \cdots, (\tau_d, \theta_d)\} \in ((\mathcal{S} \times \mathcal{A} \times \mathbb{R})^H \times \mathbb{R})^d$, where the trajectories $\tau_1, \cdots, \tau_d$ pass through $s$. We also denote $\mathcal{S}_X$ to be the union of all states which appear in $\tau_1, \cdots, \tau_d$. In order to show a bound that $\mathrm{Pdim}(\mathcal{P}^{\Pi_s}) \leq d$ it suffices to prove that $\left|\mathcal{P}^{\Pi_s,+}\big|_X\right| < 2^d$.

We first observe that

$$\left|\mathcal{P}^{\Pi_s,+}\big|_X\right| \leq 1 + \sum_{s' \in \mathcal{S}_X} |\{(\mathbb{1}\{p^\pi_{s \to s'}(\tau_1) \leq \theta_1\}, \cdots, \mathbb{1}\{p^\pi_{s \to s'}(\tau_d) \leq \theta_d\}) : \pi \in \Pi_s\}|.$$

The inequality follows because for any choice $s' \notin \mathcal{S}_X$, we have

$$(\mathbb{1}\{p^\pi_{s \to s'}(\tau_1) \leq \theta_1\}, \cdots, \mathbb{1}\{p^\pi_{s \to s'}(\tau_d) \leq \theta_d\}) = \vec{0},$$

no matter what $\pi$ is, contributing at most 1 to the count. Furthermore, once we have fixed $s'$ and the $\{\tau_1, \cdots, \tau_d\}$ the quantities $\frac{1}{|\Pi_{\mathrm{core}}|} \sum_{\pi' \in \Pi_{\mathrm{core}}} \mathbb{1}\{\pi_e \rightsquigarrow \tau_{i,h:h'}\}$ for every $i \in [d]$ are constant (do not depend on $\pi$), so we can reparameterize $\theta'_i := \theta_i \cdot \frac{1}{|\Pi_{\mathrm{core}}|} \sum_{\pi' \in \Pi_{\mathrm{core}}} \mathbb{1}\{\pi_e \rightsquigarrow \tau_{i,h:h'}\}$ to get:

$$\left|\mathcal{P}^{\Pi_s,+}\big|_X\right| \leq 1 + \sum_{s' \in \mathcal{S}_X} |\{(b_1(\pi), \cdots, b_d(\pi)) : \pi \in \Pi\}|, \tag{32}$$

$$\text{where} \quad b_i(\pi) := \mathbb{1}\{\mathbb{1}\{\pi \leadsto \tau_{i,h:h'}\}\mathbb{1}\{\tau_i \in \mathfrak{T}_\pi(s \to s')\} \le \theta_i'\}.$$

Now we count how many values the vector $(b_1(\pi), \cdots, b_d(\pi))$ can take for different $\pi \in \Pi_s$. Without loss of generality, we can (1) assume that the $\theta_i' = 0$ (since a product of indicators can only take values in $\{0, 1\}$, and if $\theta' \ge 1$ then we must have $b_i(\pi) = 1$ for every $\pi$), and (2) $s' \in \tau_i$ for each $i \in [d]$ (otherwise $b_i(\pi) = 0$ for every $\pi \in \Pi$). So we can rewrite $b_i(\pi) = \mathbb{1}\{\pi \leadsto \tau_{i,h:h'}\}\mathbb{1}\{\tau_i \in \mathfrak{T}_\pi(s \to s')\}$. For every fixed choice of $s'$ we upper bound the size of the set as:

$$\left|\{(b_1(\pi), \cdots, b_d(\pi)) : \pi \in \Pi\}\right|$$

$$\overset{(i)}{\le} \left|\{(\mathbb{1}\{\pi \leadsto \tau_{1,h:h'}\}, \cdots, \mathbb{1}\{\pi \leadsto \tau_{d,h:h'}\}) : \pi \in \Pi\}\right|$$
$$\times \left|\{(\mathbb{1}\{\tau_1 \in \mathfrak{T}_\pi(s \to s')\}, \cdots, \mathbb{1}\{\tau_d \in \mathfrak{T}_\pi(s \to s')\}) : \pi \in \Pi\}\right|$$

$$\overset{(ii)}{\le} \left|\left\{\left(\pi(s^{(1)}), \pi(s^{(2)}), \cdots, \pi(s^{(dH)})\right) : \pi \in \Pi\right\}\right| \times \left|\left\{\left(\mathbb{1}\left\{s^{(1)} \in \mathcal{S}_\pi\right\}, \cdots, \mathbb{1}\left\{s^{(dH)} \in \mathcal{S}_\pi\right\}\right) : \pi \in \Pi\right\}\right|$$

$$\overset{(iii)}{\le} \left(\frac{dH \cdot e(A+1)^2}{2\mathrm{Ndim}(\Pi)}\right)^{\mathrm{Ndim}(\Pi)} \times (dH)^D. \tag{33}$$

The inequality $(i)$ follows by upper bounding by the Cartesian product. The inequality $(ii)$ follows because (1) for the first term, the vector $(\mathbb{1}\{\pi \leadsto \tau_{1,h:h'}\}, \cdots, \mathbb{1}\{\pi \leadsto \tau_{d,h:h'}\})$ is determined by the number of possible behaviors $\pi$ has over all $dH$ states in the trajectories, and (2) for the second term, the vector $(\mathbb{1}\{\tau_1 \in \mathfrak{T}_\pi(s \to s')\}, \cdots, \mathbb{1}\{\tau_d \in \mathfrak{T}_\pi(s \to s')\})$ is determined by which of the $dH$ states lie in the petal set for $\mathcal{S}_\pi$. The inequality $(iii)$ follows by applying Lemma 15 to the first term and Sauer's Lemma to the second term, further noting that every petal $\mathcal{S}_\pi$ set has cardinality at most $D$.

Combining Eqs. (32) and (33) we get the final bound that

$$\left|\mathcal{P}^{\Pi_s,+}\big|_X\right| \le 1 + (dH)^{D+1} \cdot \left(\frac{dH \cdot e(A+1)^2}{2\mathrm{Ndim}(\Pi)}\right)^{\mathrm{Ndim}(\Pi)}.$$

To conclude the calculation, we observe that this bound is $< 2^d$ whenever $d = \widetilde{\mathcal{O}}(D + \mathrm{Ndim}(\Pi))$, which we can again use in conjunction with Lemma 16 to prove the desired uniform convergence statement found in Eq. (31). Ultimately this allows us to replace the $\log|\Pi|$ with $\widetilde{\mathcal{O}}(D + \mathrm{Ndim}(\Pi))$ in the upper bound of Theorem 4; the precise details are omitted.

# H  Connections to Other Complexity Measures

We show relationships between the spanning capacity and several other combinatorial measures of complexity.

For every $h \in [H]$ we denote the state space at layer $h$ as $\mathcal{S}_h := \{s_{(j,h)} : j \in [K]\}$ for some $K \in \mathbb{N}$. We will restrict ourselves to binary action spaces $\mathcal{A} = \{0, 1\}$, but the definitions and results can be extended to larger (but finite) action spaces. In addition, we will henceforth assume that all policy classes $\Pi$ under consideration satisfy the following stationarity assumption.

**Assumption 1.** *The policy class $\Pi$ satisfies* stationarity*: for every $\pi \in \Pi$ we have*

$$\pi(s_{(j,1)}) = \pi(s_{(j,2)}) = \cdots = \pi(s_{(j,H)}) \quad \text{for every } j \in [K].$$

*For any $\pi \in \Pi$ and $j \in [K]$, we use $\pi(j)$ as a shorthand to denote the value of $\pi(s_{(j,h)})$ for every $h$.*

The stationarity assumption is not required but is useful for simplifying the definitions and results.

## H.1  Definitions and Relationships

First, we state several complexity measures in learning theory. For further discussion on these quantities, see (Foster et al., 2021c; Li et al., 2022).

**Definition 10** (Combinatorial Eluder Dimension). *Fix any stationary base policy $\bar{\pi}$. The combinatorial eluder dimension of $\Pi$ w.r.t. $\bar{\pi}$, denoted $\dim_{\mathsf{E}}(\Pi; \bar{\pi})$, is the length of the longest sequence $(j_1, \pi_1), \ldots, (j_N, \pi_N)$ such that for every $\ell \in [N]$:*

$$\pi_\ell(j_\ell) \ne \bar{\pi}(j_\ell), \quad \text{and} \quad \forall k < \ell, \ \pi_\ell(j_k) = \bar{\pi}(j_k).$$

*We define the combinatorial eluder dimension of $\Pi$ as $\dim_{\mathsf{E}}(\Pi) := \sup_{\pi \in \Pi} \dim_{\mathsf{E}}(\Pi; \bar{\pi})$.*[8]

**Definition 11** (Star Number (Hanneke and Yang, 2015))**.** *Fix any stationary base policy $\bar{\pi}$. The* star number *of $\Pi$ w.r.t. $\bar{\pi}$, denoted $\dim_{\mathsf{S}}(\Pi; \bar{\pi})$, is the length of the longest sequence $(j_1, \pi_1), \ldots, (j_N, \pi_N)$ such that for every $\ell \in [N]$:*

$$\pi_\ell(j_\ell) \neq \bar{\pi}(j_\ell), \quad \text{and} \quad \forall k \neq \ell, \ \pi_\ell(j_k) = \bar{\pi}(j_k).$$

*We define the star number of $\Pi$ as $\dim_{\mathsf{S}}(\Pi) := \sup_{\pi \in \Pi} \dim_{\mathsf{S}}(\Pi; \bar{\pi})$.*

**Definition 12** (Threshold Dimension (Alon et al., 2019; Li et al., 2022))**.** *Fix any stationary base policy $\bar{\pi}$. The* threshold dimension *of $\Pi$ w.r.t. $\bar{\pi}$, denoted $\dim_{\mathsf{T}}(\Pi; \bar{\pi})$, is the length of the longest sequence $(j_1, \pi_1), \ldots, (j_N, \pi_N)$ such that for every $\ell \in [N]$:*

$$\forall m \geq \ell, \ \pi_\ell(j_m) \neq \bar{\pi}(j_m), \quad \text{and} \quad \forall k < \ell, \ \pi_\ell(j_k) = \bar{\pi}(j_k).$$

*We define the threshold dimension of $\Pi$ as $\dim_{\mathsf{T}}(\Pi) := \sup_{\pi \in \Pi} \dim_{\mathsf{T}}(\Pi; \bar{\pi})$.*

**Relationships Between Complexity Measures.**   From (Li et al., 2022, Theorem 8) we have the relationship for every $\Pi$:

$$\max\{\dim_{\mathsf{S}}(\Pi), \dim_{\mathsf{T}}(\Pi)\} \leq \dim_{\mathsf{E}}(\Pi) \leq 4^{\max\{\dim_{\mathsf{S}}(\Pi), \dim_{\mathsf{T}}(\Pi)\}}. \tag{34}$$

The lower bound is obvious from the definitions and in general cannot be improved; the upper bound also cannot be improved beyond constant factors in the exponent (Li et al., 2022).

We also remark that it is clear from the definitions that VC dimension is a lower bound on all three (eluder, star, threshold); however, $\mathrm{VC}(\Pi)$ can be arbitrarily smaller.

## H.2   Bounds on Spanning Capacity

Now we investigate bounds on the spanning capacity in terms of the aforementioned quantities.

**Theorem 7.** *For any policy class $\Pi$ satisfying Assumption 1 we have*

$$\max\left\{ \ \min\{\dim_{\mathsf{S}}(\Pi), H+1\}, \ \min\left\{2^{\lfloor \log_2 \dim_{\mathsf{T}}(\Pi) \rfloor}, 2^H\right\} \ \right\} \leq \mathfrak{C}(\Pi) \leq 2^{\dim_{\mathsf{E}}(\Pi)}.$$

We give several remarks on Theorem 7. The proof is deferred to the following subsection.

It is interesting to understand to what degree we can improve the bounds in Theorem 7. On the lower bound side, we note that the each of the terms individually cannot be sharpened:

- For the singleton class $\Pi_{\mathrm{sing}}$ we have $\mathfrak{C}(\Pi_{\mathrm{sing}}) = \min\{K, H+1\}$ and $\dim_{\mathsf{S}}(\Pi_{\mathrm{sing}}) = K$.
- For the threshold class $\Pi_{\mathrm{thres}} := \{\pi_i(j) \mapsto \mathbb{1}\{j \geq i\} : i \in [K]\}$, when $K$ is a power of two, it can be shown that $\mathfrak{C}(\Pi_{\mathrm{thres}}) = \min\{K, 2^H\}$ and $\dim_{\mathsf{T}}(\Pi_{\mathrm{thres}}) = K$.

While we also provide an upper bound in terms of $\dim_{\mathsf{E}}(\Pi)$, we note that there can be a huge gap between the lower bound and the upper bound. In fact, our upper bound is likely very loose since we are not aware of any policy class for which the upper bound is non-vacuous, i.e. $2^{\dim_{\mathsf{E}}(\Pi)} \ll \min\{2^H, |\Pi|, 2KH\}$ (implying that our bound improves on Proposition 3). It would be interesting to understand how to improve the upper bound (possibly, to scale polynomially with $\dim_{\mathsf{E}}(\Pi)$, or more directly in terms of some function of $\dim_{\mathsf{S}}(\Pi)$ and $\dim_{\mathsf{T}}(\Pi)$); we leave this as a direction for future research.

Lastly, we remark that the lower bound of $\mathfrak{C}(\Pi) \geq \min\{\Omega(\dim_{\mathsf{T}}(\Pi)), 2^H\}$ is a generalization of previous bounds which show that linear policies cannot be learned with $\mathrm{poly}(H)$ sample complexity (e.g., Du et al., 2019b), since linear policies (even in 2 dimensions) have infinite threshold dimension.

---

[8]Our definition of the combinatorial eluder dimension comes from Li et al. (2022) and is also called the "policy eluder dimension" in the paper Foster et al. (2021c). In particular, it is defined with respect to a base function $\bar{\pi}$. This differs in spirit from the original definition (Russo and Van Roy, 2013) as well as the combinatorial variant (Mou et al., 2020), which for every $\ell$ asks for witnessing *pairs* of policies $\pi_\ell, \pi'_\ell$. Our definition is never larger than the original version since we require that $\pi'_\ell = \bar{\pi}$ to be fixed for every $\ell \in [N]$.

### H.2.1 Proof of Theorem 7

We will prove each bound separately.

**Star Number Lower Bound.** Let $\bar{\pi} \in \Pi$ and the sequence $(j_1, \pi_1), \ldots, (j_N, \pi_N)$ witness $\dim_{\mathsf{S}}(\Pi) = N$. We construct a deterministic MDP $M$ for which the cumulative reachability at layer $h_{\max} := \min\{N, H\}$ (Definition 1) is at least $\min\{N, H + 1\}$. The transition dynamics of $M$ are as follows; we will only specify the transitions until $h_{\max} - 1$ (afterwards, the transitions can be arbitrary).

- The starting state of $M$ at layer $h = 1$ is $s_{(j_1, 1)}$.
- (On-Chain Transitions): For every $h < h_{\max}$,

$$P(s' \mid s_{(j_h, h)}, a) = \begin{cases} \mathbb{1}\{s' = s_{(j_{h+1}, h+1)}\} & \text{if } a = \bar{\pi}(s_{(j_h, h)}), \\ \mathbb{1}\{s' = s_{(j_h, h+1)}\} & \text{if } a \neq \bar{\pi}(s_{(j_h, h)}). \end{cases}$$

- (Off-Chain Transitions): For every $h < h_{\max}$, state index $\tilde{j} \neq j_h$, and action $a \in \mathcal{A}$,

$$P(s' \mid s_{(\tilde{j}, h)}, a) = \mathbb{1}\{s' = s_{(\tilde{j}, h+1)}\}.$$

We now compute the cumulative reachability at layer $h_{\max}$. If $N \leq H$, the the number of $(s, a)$ pairs that $\Pi$ can reach in $M$ is $N$ (namely the pairs $(s_{(j_1, N)}, 1), \cdots, (s_{(j_N, N)}, 1)$). On the other hand, if $N > H$, then the number of $(s, a)$ pairs that $\Pi$ can reach in $M$ is $H + 1$ (namely the pairs $(s_{(j_1, H)}, 1), \cdots, (s_{(j_H, H)}, 1), (s_{(j_H, H)}, 0)$). Thus we have shown that $\mathfrak{C}(\Pi) \geq \min\{N, H + 1\}$.

**Threshold Dimension Lower Bound.** Let $\bar{\pi} \in \Pi$ and the sequence $(j_1, \pi_1), \ldots, (j_N, \pi_N)$ witness $\dim_{\mathsf{T}}(\Pi) = N$. We define a deterministic MDP $M$ as follows. Set $h_{\max} = \min\{\lfloor \log_2 N \rfloor, H\}$. Up until layer $h_{\max}$, the MDP will be a full binary tree of depth $h_{\max}$; afterward, the transitions will be arbitrary. It remains to assign state labels to the nodes of the binary tree (of which there are $2^{h_{\max}} - 1 \leq N$). We claim that it is possible to do so in a way so that every policy $\pi_\ell$ for $\ell \in [2^{h_{\max}}]$ reaches a different state-action pair at layer $h_{\max}$. Therefore the cumulative reachability of $\Pi$ on $M$ is at least $2^{h_{\max}} = \min\{2^{\lfloor \log_2 N \rfloor}, 2^H\}$ as claimed.

It remains to prove the claim. The states of $M$ are labeled $j_2, \cdots, j_{2^{h_{\max}}}$ according to the order they are traversed using inorder traversal of a full binary tree of depth $h_{\max}$ (Cormen et al., 2022). One can view the MDP $M$ as a binary search tree where the action 0 corresponds to going left and the action 1 corresponds to going right. Furthermore, if we imagine that the leaves of the binary search tree at depth $h_{\max}$ are labeled from left to right with the values $1.5, 2.5, \cdots, 2^{h_{\max}} + 0.5$, then it is clear that for any $\ell \in [2^{h_{\max}}]$, the trajectory generated by running $\pi_\ell$ on $M$ is exactly the path obtained by searching for the value $\ell + 0.5$ in the binary search tree. Thus we have shown that the cumulative reachability of $\Pi$ on $M$ is the number of leaves at depth $h_{\max}$, thus proving the claim.

**Eluder Dimension Upper Bound.** Let $\dim_{\mathsf{E}}(\Pi) = N$. We only need to prove this statement when $N \leq H$, as otherwise the statement already follows from Proposition 3. Let $(M^\star, h^\star)$ be the MDP and layer which witness $\mathfrak{C}(\Pi)$. Also denote $s_1$ to be the starting state of $M^\star$. For any state $s$, we denote $\mathrm{child}_0(s)$ and $\mathrm{child}_1(s)$ to be the states in the next layer which are reachable by taking $a = 0$ and $a = 1$ respectively.

For any reachable state $s$ at layer $h$ in the MDP $M^\star$ we define the function $f(s)$ as follows. For any state $s$ at layer $h^\star$, we set $f(s) := 1$ if the state-action pairs $(s, 0)$ and $(s, 1)$ are both reachable by $\Pi$; otherwise we set $f(s) := 0$. For states in layers $h < h^\star$ we set

$$f(s) := \begin{cases} \max\{f(\mathrm{child}_0(s)), f(\mathrm{child}_1(s))\} + 1 & \text{if both } (s, 0) \text{ and } (s, 1) \text{ are reachable by } \Pi, \\ f(\mathrm{child}_0(s)) & \text{if only } (s, 0) \text{ is reachable by } \Pi, \\ f(\mathrm{child}_1(s)) & \text{if only } (s, 1) \text{ is reachable by } \Pi. \end{cases}$$

We claim that for any state $s$, the contribution to $\mathfrak{C}(\Pi)$ by policies that pass through $s$ is at most $2^{f(s)}$. We prove this by induction. Clearly, the base case of $f(s) = 0$ or $f(s) = 1$ holds. If only one child of $s$ is reachable by $\Pi$ then the contribution to $\mathfrak{C}(\Pi)$ by policies that pass through $s$ equal to the contribution to $\mathfrak{C}(\Pi)$ by policies that pass through the child of $s$. If both children of $s$ are reachable by $\Pi$ then the contribution towards $\mathfrak{C}(\Pi)$ by policies that pass through $s$ is upper bounded by the

sum of the contribution towards $\mathfrak{C}(\Pi)$ by policies that pass through the two children, i.e. it is at most $2^{f(\mathrm{child}_0(s))} + 2^{f(\mathrm{child}_1(s))} \le 2^{f(s)}$. This concludes the inductive argument.

Now we bound $f(s_1)$. Observe that the quantity $f(s_1)$ counts the maximum number of layers $h_1, h_2, \cdots, h_L$ that satisfy the following property: there exists a trajectory $\tau = (s_1, a_1, \cdots, s_H, a_H)$ for which we can find $L$ policies $\pi_1, \pi_2, \cdots, \pi_L$ so that each policy $\pi_\ell$ when run on $M^\star$ (a) reaches $s_{h_\ell}$, and (b) takes action $\pi_\ell(s_{h_\ell}) \ne a_{h_\ell}$. Thus, by definition of the eluder dimension we have $f(s_1) \le N$. Therefore, we have shown that the cumulative reachability of $\Pi$ in $M^\star$ is at most $2^N$. $\qquad\square$

# I  Extension: Can Expert Feedback Help in Agnostic PAC RL?

For several policy classes, the spanning capacity may be quite large, and our lower bounds (Theorem 2 and 3) demonstrate an unavoidable dependence on $\mathfrak{C}(\Pi)$. In this section, we investigate whether it is possible to achieve bounds which are independent of $\mathfrak{C}(\Pi)$ and instead only depend on $\mathrm{poly}(A, H, \log|\Pi|)$ under a stronger feedback model.

Our motivation comes from practice. It is usually uncommon to learn from scratch: often we would like to utilize domain expertise or prior knowledge to learn with fewer samples. For example, during training one might have access to a simulator which can roll out trajectories to estimate the optimal value function $Q^\star$, or one might have access to expert advice / demonstrations. However, this access does not come for free; estimating value functions with a simulator requires some computation, or the "expert" might be a human who is providing labels or feedback on the performance of the algorithm. Motivated by this, we consider additional feedback in the form of an expert oracle.

**Definition 13** (Expert oracle). *An expert oracle* $\mathsf{O}_{\mathrm{exp}} : \mathcal{S} \times \mathcal{A} \to \mathbb{R}$ *is a function which given an* $(s, a)$ *pair as input returns the Q value of some expert policy* $\pi_\circ$, *denoted* $Q^{\pi_\circ}(s, a)$.

Definition 13 is a natural formulation for understanding how expert feedback can be used for agnostic RL in large state spaces. We do not require $\pi_\circ$ to be the optimal policy (either over the given policy class $\Pi$ or over all $\mathcal{A}^{\mathcal{S}}$). The objective is to compete with $\pi_\circ$, i.e., with probability at least $1 - \delta$, return a policy $\widehat{\pi}$ such that $V^{\widehat{\pi}} \ge V^{\pi_\circ} - \varepsilon$ using few online interactions with the MDP and calls to $\mathsf{O}_{\mathrm{exp}}$.

A sample efficient algorithm (one which uses at most $\mathrm{poly}(A, H, \log|\Pi|, \varepsilon^{-1}, \delta^{-1})$ online trajectories and calls to the oracle) must use *both* forms of access. Our lower bounds (Theorem 2 and 3) show that an algorithm which only uses online access to the MDP must use $\Omega(\mathfrak{C}(\Pi))$ samples. Likewise, an algorithm which only queries the expert oracle must use $\Omega(SA)$ queries because it does not know the dynamics of the MDP, so the best it can do is just learn the optimal action on every state.

**Relationship to Prior Works.**   The oracle $\mathsf{O}_{\mathrm{exp}}$ is closely related to several previously considered settings (Golowich and Moitra, 2022; Gupta et al., 2022; Amortila et al., 2022). Prior work (Golowich and Moitra, 2022; Gupta et al., 2022) has studied tabular RL with *inexact* predictions for either the optimal $Q^\star$ or $V^\star$. They assume access to the entire table of values; since we study the agnostic RL setting with a large state space, we formalize access to predictions via the expert oracle. Amortila et al. (2022) study a related expert action oracle under the assumption of linear value functions. They show that in a generative model, with $\mathrm{poly}(d)$ resets and queries to an expert action oracle, one can learn an $\varepsilon$-optimal policy, thus circumventing known hardness results for the linear value function setting. Up to a factor of $A$, one can simulate queries to the expert action oracle by querying $\mathsf{O}_{\mathrm{exp}}(s, a)$ for each $a \in \mathcal{A}$.

## I.1  Upper Bound under Realizability

Under realizability (namely, $\pi_\circ \in \Pi$), it is known that the dependence on $\mathfrak{C}(\Pi)$ can be entirely removed with few queries to the expert oracle.

**Theorem 8.** *For any* $\Pi$ *such that* $\pi_\circ \in \Pi$, *with probability at least* $1 - \delta$, *the AggreVaTe algorithm (Ross and Bagnell, 2014) computes an* $\varepsilon$-*optimal policy using*

$$n_1 = O\left(\frac{A^2 H^2}{\varepsilon^2} \cdot \log \frac{|\Pi|}{\delta}\right) \text{ online trajectories} \quad\text{and}\quad n_2 = O\left(\frac{A^2 H^2}{\varepsilon^2} \cdot \log \frac{|\Pi|}{\delta}\right) \text{ calls to } \mathsf{O}_{\mathrm{exp}}.$$

The proof is omitted; it can be found in (Ross and Bagnell, 2014; Agarwal et al., 2019). We also note that actually we require a slightly weaker oracle than $O_{\mathrm{exp}}$: the AggreVaTe algorithm only queries the value of $Q^{\pi_\circ}$ on $(s, a)$ pairs which are encountered in online trajectories.

## I.2 Lower Bound in Agnostic Setting

Realizability of the expert policy used for $O_{\mathrm{exp}}$ is a rather strong assumption in practice. For example, one might choose to parameterize $\Pi$ as a class of neural networks, but one would like to use human annotators to give expert feedback on the actions taken by the learner; here, it is unreasonable to assume that realizability of the expert policy holds.

We sketch a lower bound in Theorem 9 that shows that without realizability ($\pi_\circ \notin \Pi$), we can do no better than $\Omega(\mathfrak{C}(\Pi))$ queries to a generative model or queries to $O_{\mathrm{exp}}$.

**Theorem 9** (informal). *For any $H \in \mathbb{N}$, $C \in [2^H]$, there exists a policy class $\Pi$ with $\mathfrak{C}(\Pi) = |\Pi| = C$, expert policy $\pi_\circ \notin \Pi$, and family of MDPs $\mathcal{M}$ with state space $\mathcal{S}$ of size $O(2^H)$, binary action space, and horizon $H$ such that any algorithm that returns a $1/4$-optimal policy must either use $\Omega(C)$ queries to a generative model or $\Omega(C)$ queries to the $O_{\mathrm{exp}}$.*

Before sketching the proof, several remarks are in order.

- By comparing with Theorem 8, Theorem 9 demonstrates that realizability of the expert policy is crucial for circumventing the dependence on spanning capacity via the expert oracle.

- In the lower bound construction of Theorem 9, $\pi_\circ$ is the optimal policy. Furthermore, while $\pi_\circ \notin \Pi$, the lower bound still has the property that $V^{\pi_\circ} = V^\star = \max_{\pi \in \Pi} V^\pi$; that is, the best-in-class policy $\widetilde{\pi} := \arg\max_{\pi \in \Pi} V^\pi$ attains the same value as the optimal policy. This is possible because there exist multiple states for which $\widetilde{\pi}(s) \neq \pi_\circ(s)$, however these states have $d^{\widetilde{\pi}}(s) = 0$. Thus, we also rule out guarantees of the form $V^{\widehat{\pi}} \geq \max_{\pi \in \Pi} V^\pi - \varepsilon$.

- Since the oracle $O_{\mathrm{exp}}$ is stronger than the expert action oracle (Amortila et al., 2022) (up to a factor of $A$), the lower bound extends to this weaker feedback model. Investigating further assumptions that enable statistically tractable agnostic learning with expert feedback is an interesting direction for future work.

**Proof Sketch of Theorem 9.** We present the construction as well as intuition for the lower bound, leaving out a formal information-theoretic proof.

**Construction of MDP Family.** We describe the family of MDPs $\mathcal{M}$. In every layer, the state space is $\mathcal{S}_h = \{s_{(j,h)} : j \in [2^h]\}$, except at $\mathcal{S}_H$ where we have an additional terminating state, $\mathcal{S}_H = \{s_{(j,h)} : j \in [2^H]\} \cup \{s_\perp\}$. The action space is $\mathcal{A} = \{0, 1\}$.

The MDP family $\mathcal{M} = \{M_{b,f^\star}\}_{b \in \mathcal{A}^{H-1}, f^\star \in \mathcal{A}^{\mathcal{S}_H}}$ is parameterized by a bit sequence $b \in \mathcal{A}^{H-1}$ as well as a labeling function $f^\star \in \mathcal{A}^{\mathcal{S}_H}$. The size of $\mathcal{M}$ is $2^{H-1} \cdot 2^{2^H}$. We now describe the transitions and rewards for any $M_{b,f^\star}$. In the following, let $s_b \in \mathcal{S}_{H-1}$ be the state that is reached by playing the sequence of actions $(b[1], b[2], \cdots, b[H-2])$ for the first $H-2$ layers.

- **Transitions.** For the first $H-2$ layers, the transitions are the same for $M_{b,f^\star} \in \mathcal{M}$. At layer $H-1$, the transition depends on $b$.

  - For any $h \in \{1, 2, \ldots, H-2\}$, the transitions are deterministic and given by a tree process: namely

    $$P(s' \mid s_{(j,h)}, a) = \begin{cases} \mathbb{1}\{s' = s_{(2j-1,h+1)}\} & \text{if } a = 0, \\ \mathbb{1}\{s' = s_{(2j,h+1)}\} & \text{if } a = 1. \end{cases}$$

  - At layer $H-1$, for the state $s_b$, the transition is $P(s' \mid s_b, a) = \mathbb{1}\{s' = s_\perp\}$ for any $a \in \mathcal{A}$. For all other states, the transitions are uniform to $\mathcal{S}_H$, i.e., for any $s \in \mathcal{S}_{H-1} \backslash \{s_b\}$, $a \in \mathcal{A}$, the transition is $P(\cdot \mid s, a) = \mathrm{Uniform}(\mathcal{S}_H \backslash \{s_\perp\})$.

- **Rewards.** The rewards depend on the $b \in \mathcal{A}^{H-1}$ and $f^\star \in \mathcal{A}^{\mathcal{S}_H}$.

- The reward at layer $H - 1$ is $R(s, a) = \mathbb{1}\{s = s_b, a = b[H - 1]\}$.
- The reward at layer $H$ is

$$
\begin{aligned}
R(s_\perp, a) &= 0 && \text{for any } a \in \mathcal{A}, \\
R(s, a) &= \mathbb{1}\{a = f^\star(s)\} && \text{for any } s \neq s_\perp, a \in \mathcal{A}.
\end{aligned}
$$

From the description of the transitions and rewards, we can compute the value of $Q^\star(\cdot, \cdot)$.

- *Layers $1, \cdots, H - 2$:* For any $s \in \mathcal{S}_1 \cup \mathcal{S}_2 \cup \cdots \cup \mathcal{S}_{H-2}$ and $a \in \mathcal{A}$, the $Q$-value is $Q^\star(s, a) = 1$.

- *Layer $H - 1$:* At $s_b$, the $Q$-value is $Q^\star(s_b, a) = \mathbb{1}\{a = b[H - 1]\}$. For other states $s \in \mathcal{S}_{H-1} \backslash \{s_b\}$, the $Q$-value is $Q^\star(s, a) = 1$ for any $a \in \mathcal{A}$.

- *Layer $H$:* At $s_\perp$, the $Q$-value is $Q^\star(s_\perp, a) = 0$ for any $a \in \mathcal{A}$. For other states $s \in \mathcal{S}_H \backslash \{s_\perp\}$, the $Q$-value is $Q^\star(s, a) = \mathbb{1}\{a = f^\star(s)\}$.

Lastly, the optimal value is $V^\star = 1$.

**Expert Oracle.** The oracle $\mathsf{O}_{\exp}$ returns the value of $Q^\star(s, a)$.

**Policy Class.** The policy class $\Pi$ is parameterized by bit sequences of length $H - 1$. Denote the function $\mathrm{bin} : \{0, 1, \ldots, 2^{H-1}\} \mapsto \mathcal{A}^{H-1}$ that returns the binary representation of the input. Specifically,

$$
\Pi := \{\pi_b : b \in \{\mathrm{bin}(i) : i \in \{0, 1, \ldots, C - 1\}\}\},
$$

where each $\pi_b$ is defined such that $\pi_b(s) := b[h]$ if $s \in \mathcal{S}_h$, and $\pi_b(s) := 0$ otherwise. By construction it is clear that $\mathfrak{C}(\Pi) = |\Pi| = C$.

**Lower Bound Argument.** Consider any $M_{b, f^\star}$ where $b \in \{\mathrm{bin}(i) : i \in \{0, 1, \ldots, C - 1\}\}$ and $f^\star \in \mathcal{A}^{\mathcal{S}_H}$. There are two ways for the learner to identify a $1/4$-optimal policy in $M_{b, f^\star}$:

- Find the value of $b$, and return the policy $\pi_b$, which has $V^{\pi_b} = 1$.

- Estimate $\widehat{f} \approx f^\star$, and return the policy $\pi_{\widehat{f}}$ which picks arbitrary actions for any $s \in \mathcal{S}_1 \cup \mathcal{S}_2 \cup \cdots \cup \mathcal{S}_{H-1}$ and picks $\pi_{\widehat{f}}(s) = \widehat{f}(s)$ on $s \in \mathcal{S}_H$.

We claim that in any case, the learner must either use many samples from a generative model or many calls to $\mathsf{O}_{\exp}$. First, observe that since the transitions and rewards at layers $1, \cdots, H - 2$ are known and identical for all $M_{b, f^\star} \in \mathcal{M}$, querying the generative model on these states does not provide the learner with any information. Furthermore, in layers $1, \cdots, H - 2$, every $(s, a)$ pair has $Q^\star(s, a) = 1$, so querying $\mathsf{O}_{\exp}$ on these $(s, a)$ pairs also does not provide any information to the learner. Thus, we consider learners which query the generative model or the expert oracle at states in layers $H - 1$ and $H$.

In order to identify $b$, the learner must identify which $(s, a)$ pair at layer $H - 1$ achieves reward of 1. They can do this either by (1) querying the generative model at a particular $(s, a)$ pair and observing if $r(s, a) = 1$ (or if the transition goes to $s_\perp$); or (2) querying $\mathsf{O}_{\exp}$ at a particular $(s, a)$ pair and observing if $Q^\star(s, a) = 0$ (which informs the learner that $s_b = s$ and $b[H - 1] = 1 - a$). In either case, the learner must expend $\Omega(C)$ queries in total in order to identify $b$.

To learn $f^\star$, the learner must solve a supervised learning problem over $\mathcal{S}_H \backslash s_\perp$. They can learn the identity of $f^\star(s)$ by querying either the generative model or the expert oracle on $\mathcal{S}_H$. Due to classical supervised learning lower bounds, learning $f^\star$ requires $\Omega(\mathrm{VC}(\mathcal{A}^{\mathcal{S}_H})) = \Omega(2^H)$ queries. $\qquad\square$

## J   Technical Tools

**Lemma 17** (Hoeffding's Inequality). *Let $Z_1, \cdots, Z_n$ be independent bounded random variables with $Z_i \in [a, b]$ for all $i \in [n]$. Then*

$$\mathbb{P}\left[|\frac{1}{n}\sum_{i=1}^{n} Z_i - \mathbb{E}[Z_i]| \geq t\right] \leq 2\exp\left(-\frac{2nt^2}{(b-a)^2}\right).$$

**Lemma 18** (Multiplicative Chernoff Bound). *Let $Z_1, \cdots, Z_n$ be i.i.d. random variables taking values in $\{0, 1\}$ with expectation $\mu$. Then for any $\delta > 0$,*

$$\mathbb{P}\left[\frac{1}{n}\sum_{i=1}^{n} Z_i \geq (1+\delta)\cdot\mu\right] \leq \exp\left(-\frac{\delta^2\mu n}{2+\delta}\right).$$

*Furthermore for any $\delta \in (0, 1)$,*

$$\mathbb{P}\left[\frac{1}{n}\sum_{i=1}^{n} Z_i \leq (1-\delta)\cdot\mu\right] \leq \exp\left(-\frac{\delta^2\mu n}{2}\right).$$

