# OpenReview forum: "When is Agnostic Reinforcement Learning Statistically Tractable?"
_NeurIPS.cc/2023/Conference — NeurIPS 2023 poster_

### Official Review · Reviewer_HA2Z · 2023-06-22

**Soundness:** 3 good
**Presentation:** 3 good
**Contribution:** 2 fair
**Rating:** 7
**Confidence:** 3

**Summary:**

The authors study the problem of sample complexity in finite horizon MDPs from an agnostic perspective. More specifically, the term agnostic refers to the following setting: given a (finite) policy class $\Pi$, they aim at understanding the number of samples that are required to output a policy that is $\epsilon$-optimal w.r.t. the best policy within $\Pi$ is.

To study the problem, they introduce the concept of *spanning capacity*, a novel complexity metric that depends solely on $\Pi$ and is independent of the MDP dynamics.

Then, the authors study two interaction settings (i.e., generative model and online setting) and derive the following results:
- Generative model setting. The authors show that the spanning capacity describes the learning complexity up to an $H \log(|\Pi|)$ factor. This is proven by deriving lower and upper bounds that are explicitly dependent on the spanning capacity.
- Online setting. In this case, the results can be summarized as follows:
    - The spanning capacity fails at describing the complexity of online RL. Indeed, the authors present a lower bound that shows that a superpolynomial-in-the-horizon number of samples are still needed for agnostic learning (i.e., it cannot be lower-bounded by any polynomial function of the spanning capacity).
    - By restricting the set $\Pi$ to "sunflower" policies, provably efficient learning is possible. In this sense, the authors propose POPLER, an algorithm whose complexity scales with the spanning capacity and characteristic that depends on the structure of $\Pi$.



**Strengths:**

Provably efficient learning is a significant and open problem for RL agents, with a longstanding history of different methods and approaches. The authors thoroughly review existing works (Appendix A) and properly contextualize their results within the field.
In this sense, novel ideas, such as the ones proposed by the authors, are of interest to the NeurIPS community that is interested in RL theory.

More specifically, the notion of spanning capacity is, to the best of my knowledge, original and of potential interest.
Indeed:
1) As the authors show in Proposition 3, this metric recovers, in worst-case scenarios, existing upper and lower bounds available in the literature. Furthermore, in more favorable settings, it is significantly smaller.
2) Furthermore, as the author shows in Theorem 1 and 2, it describes the min-max sample complexity (up to H*log(\Pi) factors) in the generative model setting. Although, due to this mismatch, the problem is still open, the progress remains significant.

In the online interaction setting, this metric turns out to be not descriptive enough (i.e., Theorem 5, a superpolynomial-in-the-horizon number of samples are still needed for agnostic learning). Although, in some sense, this is a drawback (see weaknesses below), this negative result can potentially provide insights to guide future research in metrics that properly describe provably efficient agnostic RL in online settings.

Finally, (minor) by carefully restricting the set of policies, the authors are able to show that the spanning capacity can be used to upper-bound the complexity in the online setting as well. This adds an additional (but minor, see below why) point in favor of the spanning capacity.

On the clarity. Overall, the paper is a nice read, and it is relatively easy to follow (except a minor suggestion, see below).


**Weaknesses:**

**Online RL**
As commented above, the authors show that the spanning capacity turns out to be not descriptive enough for providing polynomial bounds on the sample complexity of agnostic RL in an online interaction setting. Nevertheless, the metric is in itself something that the authors propose. For this reason, one might object that this is a potential drawback of the metric that is *designed* by the authors.
A main question, in this sense, remains open: Is there any metric (e.g., a generalization of the spanning capacity) that characterizes both scenarios?


**On sunflower policies: online RL under restrictive assumptions on $\Pi$**
Currently, this point seems the major weakness of the presented work. Indeed, I find their definition quite involved, and it is unclear if it is the result of an artifact of the analysis rather than something necessary to reach polynomial sample complexity in the online setting. The fact that a lower bound is missing reinforces this doubt. I invite the authors to comment on this point.


**On the clarity**. I would suggest the authors to improve the clarity on line 206-213, maybe by introducing some figures. Indeed, I still miss this natural interpretation of the spanning capacity.


**Minors**:
- The term "surprising" to refer to the mismatch between online RL and the generative model setting might be mitigated. Indeed, the generative model setting is intuitively easier.
- It is remarked multiple times that other works make stronger assumptions that are rarely satisfied in practice (e.g., realizability). Nevertheless, stronger theoretical results (\epsilon-optimality) are usually achieved under these assumptions. This should be stated more clearly in my opinion.
- line 176 typo: "loogmk"
- The definition of the set of MDPs seems improper. To the best of my understanding, state, and action spaces are fixed over this class. In this sense, e.g., M^sto is not the set of all MDPs with horizon H, but the set of all MDPs with horizon H and state-action spaces given by (S,A)



**Questions:**

1) I invite the authors to discuss the relationship between their work and problem-dependent learning complexity analysis (e.g., "Optimistic PAC RL: the instance-dependent view", Tirinzoni et al., ALT 2023).

2) See the point on sunflower policies in the weakness section.

**Limitations:**

The authors discuss the limitations of their work as open questions in the conclusion section.

Potential negative societal impact: the paper deals with foundational research on the sample complexity of RL. I don't see a direct path to negative applications.

---

> ### Author Rebuttal · Authors · 2023-08-09
>
> We thank the reviewer for their comments and time. Below we respond to your points and questions.
>
> **A Metric that Characterizes Both Scenarios.** To the best of our knowledge, there is no metric that captures the complexity of agnostic PAC RL in both generative and online RL settings. We introduced and investigated spanning complexity because:
>
> - It is the right complexity measure in the generative model setting.
> - Since online RL is at least as hard as RL with a generative model, spanning complexity also lower bounds the sample complexity in online RL.
> - It is closely related to the notion of coverability, which was introduced in prior work [1] for realizable value-function based RL.
>
> We view this as an important question for future research.
>
> **Sunflower Property.** We address this concern, along with other questions/points on the sunflower property, in a joint response to all the reviewers.
>
> **A Natural Interpretation of the Spanning Capacity.** In the final version, we will include a clearer exposition of the spanning capacity, as well as figures that illustrate it.
>
>
> **Separation between Online RL and Generative Model.** In light of our results in Section 4, it was natural to conjecture that spanning capacity also characterized online RL. Indeed, this was true for many nontrivial policy classes we considered.
>
> We agree with the reviewer that the generative model is intuitively easier, since one can simulate online RL with a generative model. However, separation results between the generative model and online RL are exceedingly rare, and in this sense we view this separation as ``surprising''. In fact, the only policy classes for which we can show a separation are the nonexplicit constructions from Theorem 3 based on the probabilistic method  (specifically, see our Lemma 4).
>
> **Definition of Set of MDPS.** You are correct, we will clarify this in our revision.
>
> **Comparison to Problem-Dependent Learning.** We thank the reviewer for bringing [2] to our attention. Please find below a summary of the key differences:
>
> - [2] studies instance-dependent guarantees for tabular RL. Their bounds depend on the suboptimality gaps as well as reachability probabilities. Furthermore, their bound in Theorem 2 has a linear dependence on $|\mathcal{S}|$, the state space size. In contrast, our work focuses on the large state space setting where we do not want any extraneous dependence on the state space size. We also do not investigate instance-dependent bounds. It would be interesting to see if POPLER can be adapted to achieve refined instance-dependent bounds.
> - Algorithmically, the work [2] uses an optimism-based approach. In contrast, POPLER is based on reachable state identification. In this sense, POPLER is actually more similar to MOCA [3], which studies instance-dependent PAC RL for tabular (again, note that we work in the more general large state space setting, which necessitates technical innovations for efficient exploration).
>
> We will add a citation to [2, 3], and a more detailed comparison, in the final version of the paper.
>
> [1] Tengyang Xie, Dylan J. Foster, Yu Bai, Nan Jiang, Sham M. Kakade. "The Role of Coverage in Online Reinforcement Learning".
>
> [2] Andrea Tirinzoni, Aymen Al-Marjani, Emilie Kaufmann. "Optimistic PAC Reinforcement Learning: the Instance-Dependent View."
>
> [3] Andrew Wagenmaker, Max Simchowitz, Kevin Jamieson. "Beyond No Regret: Instance-Dependent PAC Reinforcement Learning."

---

> > ### Comment · Reviewer_HA2Z · 2023-08-18
> > **Ack**
> >
> > I thank the authors for their in-depth rebuttal. I have no further questions for the authors.
> >
> > I am raising my score to 7 as many concerns that I raised have been addressed. I think this submission is of interest to the community. Nevertheless, I am still a bit skeptical about the arguments on the sunflower properties.

---

### Official Review · Reviewer_c49B · 2023-06-25

**Soundness:** 3 good
**Presentation:** 4 excellent
**Contribution:** 3 good
**Rating:** 6
**Confidence:** 4

**Summary:**

This paper studies the minimax sample complexity of learning the best policy within a given policy class $\Pi$, i.e., the best sample complexity of learning $\argmax_{\pi\in \Pi}V^\pi$ in the worst-case MDPs. As a motivation, Proposition 1 shows that without future assumptions on the structure of $\Pi$, the best policy is not PAC-learnable in the worst case MDPs. Hence, this paper proposes a new complexity measure, spanning capacity, that counts maximum number of reachable states by policies in $\Pi$ for all deterministic MDP. With generative models, this paper shows that the spanning capacity characterizes the minimax sample complexity of learning $\Pi$. For online RL, this paper requires an additional structural assumption called sunflower property and proves corresponding upper bounds.

**Strengths:**

This paper studies a novel aspect of PAC learning in MDPs (called agnostic reinforcement learning) --- learning the best policy within a policy class with no modeling assumptions for the dynamics or value function class. The agnostic reinforcement learning problem could be a new approach toward RL with function approximations since the realizability assumptions on the complicated dynamics/value function class are no longer needed, and it is indeed unclear whether those assumptions hold for practical environments.

This paper proves novel upper bounds agnostic reinforcement learning setting by introducing a new complexity measure called the spanning capacity. With generative models, this paper proves that the spanning capacity is necessary and sufficient for agonistic reinforcement learning. This is morally a strong result albeit the simplicity of its proof. This paper also proves results for the classic online RL setting with additional structural assumptions on the underlying MDP.

This paper is well-written --- exposition of the results is clear with the help of concrete examples. The results and assumptions in this paper are well motivated by corresponding lower bounds (Proposition 1 and Theorem 2).

This paper mainly focuses on the minimax complexity of agonistic RL. While this perspective eliminates the dependence on the realizability assumptions on the underlying MDP, it is possible that the resulting complexity is over-pessimistic. It would be interesting to have an instance-dependent complexity measure that depends on some property of the underlying MDP.


**Weaknesses:**

The sunflower structure in Section 6 lacks necessary motivations and seems a bit out of nowhere. I understand that this assumption helps extend the IS technique to the online RL setting. However, it’s unclear to me whether this assumption is necessary/nature for agonistic RL. Intuitively, why/when should we expect the sunflower structure to hold?

[Minor] The computation complexity of the algorithm is linear in the size of the policy class. Is it possible to have some efficient implementation with the help of some standard computation oracles?


**Questions:**

-	Alg. 1 takes as input the set $S_\pi$ and $\Pi_{core}$ defined by Def. 3. They depend on the underlying MDP (because the partial trajectory is generated by the MDP, if I understand correctly). Is the algorithm assumed to have the knowledge of $S_\pi$ and $\Pi_{core}$? If not, how should the algorithm compute $S_\pi$ and $\Pi_{core}$?

-	In Theorem 3, the spanning capacity is $H^{O(\ell)}$, while the sample complexity lower bound is $\epsilon^{-\ell}$ with $\epsilon=H^{O(1)}$, which means that the sample complexity lower bound is polynomial in the spanning capacity. Then why is the spanning capacity not sufficient to characterize the minimax sample complexity?


**Limitations:**

The authors adequately addressed the limitations and potential negative societal impact.

---

> ### Author Rebuttal · Authors · 2023-08-09
>
> We thank the reviewer for their comments and time. We respond to your questions and points below.
>
> **Sunflower Structure Lacks Motivation.** We address this concern, along with other questions/points on the sunflower property, in a joint response to all the reviewers.
>
> **Computational Complexity.** Yes, the reviewer is correct that our algorithm POPLER will have computational complexity which depends polynomially on $|\Pi|$, which is prohibitively large for practical RL scenarios. Getting computationally or oracle-efficient algorithms for agnostic RL is a fascinating direction for future research.
>
> **The sets $\mathcal{S}\_\pi$ and $\Pi\_{\mathrm{core}}$.** We apologize for any confusion. We would like to correct the reviewer here: the sets $\\{\mathcal{S}\_\pi \\}\_{\pi \in \Pi}$ and $\Pi\_{\mathrm{core}}$ do not depend on the underlying MDP dynamics, but only on the policy class $\Pi$ (as well as $\mathcal{S}, \mathcal{A}, H$). The partial trajectories in Definition 3 can be arbitrary as long as they are consistent with the policy.
>
> We can assume that POPLER takes as input  $\\{\mathcal{S}\_\pi \\}\_{\pi \in \Pi}$ and $\Pi\_{\mathrm{core}}$ because these sets can be computed before by enumerating over all possible choices, and picking the ones which optimize the bound in Theorem 4. This point will be clarified in the final version.
>
> **The Sample Complexity Lower Bound.** We thank the reviewer for pointing out this issue. This can be fixed by expanding the range of acceptable $\epsilon$ and $\ell$. Here is an updated version of Theorem 3.
>
> **Theorem 3.** *Fix any sufficiently large $H$. Let $\epsilon \in (1/2^{O(H)}, O(1/H))$ and $\ell \in \\{2, 3, \dots, H\\}$ such that $1/\epsilon^\ell \le 2^H$. There exists a policy class $\Pi$ of size $O(1/\epsilon^\ell)$ with $\mathfrak{C}(\Pi) \le O(H^{4\ell + 2})$ and a family of MDPs $\mathcal{M}$ with state space $\mathcal{S}$ of size $2^{O(H)}$, binary action space, and horizon $H$ such that: for any $(\epsilon, 1/8)$-PAC algorithm, there exists an $M \in \mathcal{M}$ for which the algorithm must collect at least $\Omega(\min \\{ 1/\epsilon^\ell, 2^{H/3}\\})$ online trajectories in expectation.*
>
> To interpret this result, let us pick $\epsilon = 1/2^{\sqrt{H}}$ and $\ell = \sqrt{H}$. Then we get the following corollary, which demonstrates that the sample complexity for learning this class cannot be a polynomial function of $\mathfrak{C}(\Pi)$, $\epsilon$, and $\log |\Pi|$.
>
> **Corollary.** *For any sufficiently large $H$, there exists a policy class $\Pi$ with $\mathfrak{C}(\Pi) = 2^{O(\sqrt{H} \log H)}$ such that for any $(1/2^{\sqrt{H}}, 1/8)$ PAC algorithm, there exists an MDP for which the algorithm must collect at least $2^{\Omega(H)}$ online trajectories in expectation.*
>
> We will include this correction in the final version of the paper.

---

> > ### Comment · Reviewer_c49B · 2023-08-19
> > **Thank you for the response**
> >
> > Thank you for the response. While the evidence about the necessity of the sunflower property partly addressed my concerns, I still think this paper could benefit a lot from establishing somewhat rigorous/provable claims about it. Given that my score is already weak accept, I will keep my score as it is.

---

### Official Review · Reviewer_Gfxz · 2023-07-16

**Soundness:** 3 good
**Presentation:** 3 good
**Contribution:** 3 good
**Rating:** 6
**Confidence:** 3

**Summary:**

This paper studies conditions on which agnostic reinforcement learning is statistically tractable. The paper introduces a new concept of complexity measure called spanning capacity, which sorely depends on the policy class. The authors studies in what cases the sample complexity of agnostic RL can be polynomial to spanning capacity.
The contributions include 1) for generative model, the authors show spanning capacity is a necessary and sufficient complexity measure for agnostic RL, 2) for online model, the authors show spanning capacity is insufficient: they prove a lower bound superpolynomial to the spanning capacity of policy class. 3) the authors propose a strong assumption of the policy class called sunflower structure, and propose an algorithm that is statistically efficient to spanning capacity under the sunflower structure assumption.

**Strengths:**

- Clarify: the paper is very well-written. I appreciate the intuitions and examples given in the main text. In addition, the proofs in the appendix are well organized and easy to follow.
- Novelty and Significance: While prior work has studied the sample complexity for RL problems, what assumptions are sufficient or necessary for statistically efficient is still not clear. The concept spanning capability is new and reasonable to me.
- Technical correctness: I didn’t find any technical errors in the proofs included in the appendix, but I admit I was not able to go through all the details in the appendix.

**Weaknesses:**

- The (K,D)-sunflower structure is an overly strong assumption and not reasonable enough. It seems to only work when there is a small subset of policies (so K is small) that covers most states (so D is small).
- Although Algorithm 1 is sample efficient and only needs to collect polynomial trajectotires, the algorithm itself is not efficient: the algorithm needs to traverse all policies in $\Pi$ to find the policy that can reach states which cannot be reached by $\Pi_{\text{core}}$. The complexity of Algorithm 1 is at least $O(|\Pi|)$.
- For Theorem 4, it seems the sample complexity is about $O({1}/{\epsilon^4})$, which is not optimal on the order of $\epsilon$. This also implies the sample complexity will be larger than $\widetilde{\mathcal{O}} (\min\{A^H. |\Pi|, HSA\}/\epsilon^2)$ when $\epsilon$ is small enough.


**Questions:**

- in Definition 2, how is the second equation established? It is not clear to me how to make the inf on $\mu$ to the sum on $(s,a)$ pairs.
- It seems coverability coefficient is always no larger than the span capability. Is coverability a sufficient complexity measure for generative model? Moreover, I note that the authors claim coverability is insufficient for online RL. However, considering that spanning capability is also insufficient for online RL, where is the key difference.
- in appendix Lemma 2(1), what does $N$ means. The authors have not introduced this notation in Section E.


One minor error: in appendix Line 865 and 867, it should be $j\in [2^{2H}]$ instead of $j\in [2^{H}]$.



**Limitations:**

Theoretical work. No limitations.

---

> ### Author Rebuttal · Authors · 2023-08-09
>
> We thank the reviewer for their comments and time. Below we respond to your questions and points.
>
> **The Sunflower Property is Overly Strong.**
> We address this concern, along with other questions/points on the sunflower property, in a joint response to all the reviewers.
>
> **Regarding Definition 2.**
> The equivalence in the second equation follows by setting $\mu\_h(s,a) \propto \max\_{\pi \in \Pi} d^\pi\_h(s,a)$. This result was established in the prior work [1] in their Lemma 3. We refer the reviewer to look at [1] for a detailed proof and more intuition.
>
> **Is Coverability Sufficient for Generative Model?**
> We do not know, but we conjecture that one can show that coverability itself is insufficient when $\mathfrak{C}(\Pi)$ is large, i.e., one cannot adapt to ``easy'' instances. It may be possible to provide such a lower bound using low-rank MDPs (for example, see the lower bound construction in [2]).
>
> A related point: for online RL, we can actually replace $\mathfrak{C}(\Pi)$ by the smaller coverability coefficient in the theorem. However, in this setting, we need to make the additional sunflower property assumption.
>
> Understanding when we can get guarantees in terms of coverability (or similar instance-dependent measures) is an exciting direction for future work.
>
> **Lemma 2 (1)** This is a typo. Here, $N$ denotes the size of $\Pi^{(\ell)}$.
>
> **Minor Error.** Yes, you are correct. This mistake is made in a couple of places, and we will correct it.
>
> [1] Tengyang Xie, Dylan J. Foster, Yu Bai, Nan Jiang, Sham M. Kakade. "The Role of Coverage in Online Reinforcement Learning".
>
> [2] Christoph Dann, Yishay Mansour, Mehryar Mohri, Ayush Sekhari, Karthik Sridharan. "Agnostic Reinforcement Learning with Low-Rank MDPs and Rich Observations".

---

> > ### Comment · Reviewer_Gfxz · 2023-08-18
> >
> > Thanks for the detailed response to my questions. My major concerns have been addressed. Given the score indicates accept, I meantain this score.

---

### Official Review · Reviewer_yJrU · 2023-07-24

**Soundness:** 4 excellent
**Presentation:** 4 excellent
**Contribution:** 3 good
**Rating:** 7
**Confidence:** 4

**Summary:**

This work studies agnostic learning in RL, and seeks to characterize when, given some policy class $\Pi$, it is possible to learn an $\epsilon$-optimal policy in $\Pi$ regardless of the MDP. They propose a novel complexity measure—the spanning capacity—which depends only on the policy class $\Pi$ (i.e. is independent of the underlying MDP) which they show is both a necessary and sufficient measure of learnability when the learner has generative access to the MDP. In the fully online setting, however, it is shown that the spanning capacity is no longer a sufficient measure of capacity. To characterize efficient learning in the online setting, they introduce the notion of a ``sunflower’’ policy class, and show that if a policy class has bounded spanning capacity and is a sunflower class, then efficient learning is possible in the online setting as well.

**Strengths:**

1. I believe this work makes an interesting and novel contribution to the RL literature. There has been much interest over the last several years in developing general complexity measures that characterize when efficient learning in RL is possible, but existing work has typically made stronger assumptions—for example, access to a model or value function class where realizability holds (i.e. the true model/value function is in the class)—than this work, which does not assume realizability, and simply considers the question of finding the best policy in some set of policies.
2. The results in this work show that there is a formal separation between the generative model setting and online access setting in policy learning framework considered here. To my knowledge, such a result has not been previously known, and is an interesting observation.
3. In the generative setting, the characterization given by the spanning capacity is very clean: for any policy class $\Pi$, efficient learning is possible if and only if the spanning capacity is small. While this is minimax over MDPs (see below), it is not minimax over policy classes—it applies to any policy class.
4. The spanning capacity is a very intuitive measure of complexity and admits a clean interpretation.
5. The paper is very well-written and reads very nicely.

**Weaknesses:**

1. While the spanning capacity does provide a clean characterization of the learnability of a given policy class, it is a very worst-case measure over problem instances. It could be the case that the spanning capacity of a given policy class is very large, implying that efficient learning is not possible in general with the given policy class, but where on many MDPs efficient learning with the given policy class is still possible. In particular, for the spanning capacity to be large, there only needs to exist a single deterministic MDP for which it is difficult to find the optimal policy in class on—there could be many MDPs where it is very easy to find the optimal policy in class on, but this is not captured by the spanning capacity. In contrast, in practice we are typically interested not just in the policy class but the interaction between the policy class and the underlying dynamics of the MDP, as this characterizes how feasible it is to actually learn a good policy for a given problem. Due to this worst-case nature, for settings with large state or action spaces, the examples given of policy classes with bounded spanning capacity are very simple and not representative of the sorts of policy classes that would actually be used (in theory or practice).
2. While it is shown that the sunflower condition is a sufficient condition for learning in the online setting, it is not clear it is a necessary condition (and furthermore, the scaling on $(K,D)$ given in Theorem 4 is likely not tight). Some discussion of this would be helpful.
3. In principle, ignoring computation cost, I believe the optimal values of $K$ and $D$ for Theorem 4 could be computed a priori (before interacting with the MDP). This would be helpful to clarify around Theorem 4 (in particular on lines 296-298). Since POPLER takes as input $K$ and $D$, the bounded stated in Theorem 4 cannot be optimized for $K$ and $D$ after the fact, but can be optimized if the optimal $K$ and $D$ are computed before running POPLER (I found this somewhat unclear from the discussion on 296-298).
4. I understand that the main contributions of this paper are statistical, but it would still be helpful to comment on the computational efficiency of the proposed algorithms.
5. Several additional papers that should be cited: [1] also considers agnostic RL (albeit with stronger assumptions on the realizability of the model class). [2,3] are additional works on RL with function approximation that are generally relevant to this work.
6. Minor typo: line 103 should read “access” not “saccess”.

[1] Wagenmaker, Andrew, and Kevin G. Jamieson. "Instance-dependent near-optimal policy identification in linear mdps via online experiment design." Advances in Neural Information Processing Systems 35 (2022): 5968-5981.

[2] Foster, Dylan J., Noah Golowich, and Yanjun Han. "Tight guarantees for interactive decision making with the decision-estimation coefficient." arXiv preprint arXiv:2301.08215 (2023).

[3] Zhong, Han, et al. "GEC: A Unified Framework for Interactive Decision Making in MDP, POMDP, and Beyond." CoRR (2022).


**Questions:**

1. The DEC [4] is a recently proposed measure of complexity that applies in the realizable setting. While the setting considered here does not assume realizability, and so is not directly comparable to the DEC, some comparison with the DEC would be helpful. In particular, one could apply the DEC to the model class $\mathcal{M}$ the set of all MDPs with no more than $S$ states, $A$ actions, and horizon $H$ (and with the decision space set to the given policy class $\Pi$). One would get realizability for free in this setting, so the characterization given by the DEC could be compared to the spanning capacity. How would the DEC scale with this choice of $\mathcal{M}$ for the policy classes presented on lines 200-204?

2. Are there examples where $S$ and $H$ are both large, the policy set is reasonably expressive (e.g. it contains an $\epsilon$-optimal policy for a reasonable class of MDPs), and the spanning capacity is bounded? For example, say we have some featurization $\phi(s,a) \in \mathbb{R}^d$, and our class $\Pi$ is defined as $\Pi = ( \pi^w \ : \ w \in \mathcal{W} )$ for $\pi^w(s) = argmax_{a} \phi(s,a)^\top w$ and $\mathcal{W}$, for example, a cover of $\mathbb{R}^d$. Can the spanning capacity be shown to scale polynomially with $d$ in this case, or is the best that can be shown still the bounds given in Proposition 3? It is known that smoothed versions of such policy classes are sufficient for settings such as linear MDPs when $\mathcal{W}$ is an $\epsilon$-cover of $\mathbb{R}^d$ (that is, such a policy class can be shown to contain an $\epsilon$-optimal policy class for any linear MDP), so if the spanning capacity is bounded by $poly(d)$ in this case (even though $|\Pi|$ scales exponentially in $d$) would provide a compelling large state-space example where spanning capacity scales reasonably.

[4] Foster, Dylan J., et al. "The statistical complexity of interactive decision making." arXiv preprint arXiv:2112.13487 (2021).

**Limitations:**

Yes

---

> ### Author Rebuttal · Authors · 2023-08-09
>
> We thank the reviewer for their comments and time. Below we respond to your questions and points.
>
> **Spanning Capacity is Worst-Case.**
> We agree with the reviewer that spanning capacity is a worst-case notion, much like classic learning theoretic quantities like VC dimension and Littlestone dimension.
>
> In fact, understanding complexity measures that depend on the model class (and thus implicitly the policy class corresponding to optimal policies for those models) is already an active area of research in the RL theory community (see our Appendix A for several representative papers). On the other hand, our work considers the other extreme when the learner only has access to a policy class. This approach, to the best of our knowledge, is not well-studied in the RL theory community, and our work is the first to consider (worst-case) *structural* assumptions on the policy class.
>
> Understanding the complexity of restricting to both specific model classes and policy classes is an important direction in RL theory, and we see our work as taking a step towards that direction.
>
> **Unclear if Sunflower Property is Necessary.** We address this concern, along with other questions/points on the sunflower property, in a joint response to all the reviewers. Furthermore, we note that we did not focus on optimizing the polynomial dependence on $D, K$, and $1/\epsilon$ in our Theorem 4.
>
> **On $(K,D)$ in Theorem 4.** We apologize for any confusion. The reviewer is correct that leaving computation aside, the optimal choices of $K$ and $D$ can be computed a priori (before running POPLER) by explicitly enumerating over all possible choices for $\Pi\_\mathrm{core}$ and $\\{ \mathcal{S}\_\pi  \\}_{\pi \in \Pi}$ and finding the best values of $K$ and $D$ that optimize the bound in Theorem 4. We will clarify this in the final version.
>
> **Computational Complexity of Our Algorithms.** Given $K$, $D$, $\Pi\_{\mathrm{core}}$, the sets $\\{\mathcal{S}\_{\pi}\\}\_{\pi \in \Pi}$, our algorithm POPLER has running time that scales polynomially with $|\Pi|$ as well as $S, A, H$, which we agree is prohibitive for large scale RL problems appearing in practice. However, as the reviewers also noted, our focus in this paper is to understand the statistical complexity of agnostic RL, which is the first step towards getting practical algorithms in agnostic RL and to the best of our knowledge has not been explored before. Getting computationally or oracle-efficient algorithms for agnostic RL is a fascinating direction for future research.
>
> **Additional References.** We thank the reviewer for the additional references, which will be incorporated into the final version.
>
> **Relationship to DEC.** Let $\mathrm{Dec}(\mathcal{M}; \Pi)$ denote the DEC of the model class $\mathcal{M}$ consisting of all stochastic MDPs (with state space S and action space A) and decision class $\Pi$. Let $\mathfrak{C}(\Pi)$ denote the spanning capacity of $\Pi$. Further, for the sake of comparison, assume that $\epsilon = O(1)$ and ignore constant and $H$ dependent factors. On the upper bound side, we have that $\mathrm{Dec}(\mathcal{M}; \Pi) \leq \mathfrak{C}(\Pi)$. This is because:
>
> - As suggested by Xie et al. 2022, $\mathrm{Dec}(\mathcal{M}; \Pi)$ is upper-bounded by the worst-case coverage of any MDP in $\mathcal{M}$ with decisions limited to $\Pi$. (see Section 6.1 in their paper for details).
> - As proved in our Lemma 1, $\mathfrak{C}(\Pi)$ is equal to the worst-case coverage of any $\mathcal{M}$ with decisions limited to $\Pi$.
> - The two results together imply that the DEC is upper bounded by the spanning capacity.
>
> On the lower bound side, it is not clear if $\mathfrak{C}(\Pi) \leq \mathrm{Dec}(\mathcal{M}; \Pi)$. A direct comparison of our results with those for Foster et al. 2021 with the model class $\mathcal{M}$ and decision space $\Pi$ is not fruitful. On the one hand, our Theorem 2 suggests that the lower bound for Agnostic RL scales as $\mathfrak{C}(\Pi)$. On the other hand, Foster et al. 2021 obtain an upper bound of $\mathrm{Dec}(\mathcal{M}; \Pi) \log(|\mathcal{M}|)$ samples. Comparing the two establishes $\mathfrak{C}(\Pi) \leq \mathrm{Dec}(\mathcal{M}; \Pi) \log(|\mathcal{M}|)$ which is a vacuous comparison as  $\log(|\mathcal{M}|) \propto |\mathcal{S}|$ could be prohibitively large for the set of all stochastic MDPs.
>
>
> **Examples of Expressive Policy Classes with Bounded Spanning Capacity.** Since spanning capacity is a worst-case measure, many natural examples do not admit policy classes with bounded spanning capacity. For the class of ``linear policies'' that the reviewer asked about, from the prior works, we know that:
>
>   - Linear MDP assumption + linear policy class: it is possible to find an $\epsilon$-optimal policy in $\mathrm{poly}(d)$ samples (see Theorem 3.1 of [1]).
>   - Linear policy class: if we don't assume some form of realizability (either of the dynamics class or value functions), there is an exponential in $H$ lower bound (see Theorem 4.3 of [2]).
>
> In fact, we can show, more generally, that any policy class with large threshold dimension (a learning theoretic dimension which is qualitatively equivalent to Littlestone dimension) must also have a large spanning capacity. This recovers the exponential in $H$ lower bound in [2] for linear policies. We will add more details discussing this in the final version.
>
> [1] Chi Jin, Zhuoran Yang, Zhaoran Wang, Michael I. Jordan. "Provably Efficient Reinforcement Learning with Linear Function Approximation."
>
> [2] Simon S. Du, Sham M. Kakade, Ruosong Wang, Lin F. Yang. "Is a Good Representation Sufficient for Sample Efficient Reinforcement Learning?"

---

> > ### Comment · Reviewer_yJrU · 2023-08-17
> > **Reply to rebuttal**
> >
> > I would like to thank the authors for their detailed response to my questions. I believe the majority of my concerns have been addressed, and would encourage the authors to include the answers they have given here in the final version. While I think there are still certain shortcomings to this work (lack of necessity of sunflower property, very worst-case nature of spanning capacity), I believe it does make an interesting and novel contribution to the RL literature, and will raise my score to a 7.

---

### Author Rebuttal · Authors · 2023-08-09

We thank all the reviewers for their time and valuable feedback. Since concerns about the sunflower property were raised by all the reviewers, please find a shared response below. If you have any further concerns, please let us know, and we would love to discuss further.

**Necessity of the Sunflower Property.** While we show that the sunflower property is sufficient (in addition to bounded spanning capacity), we do not know if the sunflower property is also necessary for agnostic PAC RL in the online interaction model. However, we believe that it is necessary and have *strong evidence* to support this belief:

  - All the explicit examples of policy classes that we considered in Section 3, and which are agnostic PAC learnable in online RL, satisfy it. To the best of our knowledge, we do not know of any policy class that is agnostically PAC learnable in online RL but does not satisfy the sunflower property.
  - The *only* policy class we know of that is not agnostically PAC learnable in online RL despite having bounded spanning capacity (the policy class from Theorem 3) violates the sunflower property. Furthermore, even the construction in Theorem 3 is *nonexplicit* (based on a probabilistic argument) so we think finding an explicit construction would be nontrivial.


Exploring the necessity of the sunflower property, and/or establishing the necessary conditions on $\Pi$ for agnostic learnability in online RL is an exciting direction for future work.

**Further Intuition on the Sunflower Property.**  At a high level, the sunflower property captures the intuition that there exists a small set of policies $\Pi\_{\mathrm{core}}$ that
 when executed can cover most of the trajectories that are be explored by other policies in $\Pi$. In particular, each policy $\pi \in \Pi$ can only deviate from $\Pi\_{\mathrm{core}}$ on a small set of states $\mathcal{S}\_\pi$. Intuitively, the sunflower property allows the learner to extrapolate data collected by executing a small set of policies $\Pi\_{\mathrm{core}}$ to estimate other policies in  $\pi \in \Pi$. Informally speaking, a shared structure similar to what is captured by the sunflower property seems to be crucial in order to avoid a linear dependence on $\Pi$. In the prior works that make further assumptions on the model dynamics, e.g. in Bellman-Eluder classes, etc. such shared structure was enforced via *modeling assumptions on the MDP*; our work explicitly aims to avoid making any MDP modeling assumptions.

 Finally, we also remark that, even with the sunflower structure, the problem is challenging as the number of leaf states $\\{\mathcal{S}\_\pi\\}\_{\pi \in \Pi}$ could be very large (in fact, it can scale linearly with $\Pi$ for some of our examples). Our algorithm performs non-trivial and a novel exploration approach using policy-dependent Markov Reward Processes in order to figure out the relevant leaf states and achieve sample complexity that only scales with $D$ and $K$.

---

### Decision · Program_Chairs · 2023-09-21

**Decision:**

Accept (poster)

**Comment:**

This work studies the problem of agnostic PAC reinforcement learning. The authors propose a novel complexity measure—the spanning capacity—which depends only on the policy class and which they show is both a necessary and sufficient measure of learnability when the learner has generative access to the MDP. This is no longer true in the online setting. In this setting they propose a sunflower property instead and show it is sufficient to guarantee efficient learning. The reviewers agreed this work would be a solid addition to the Neurips 2023 program.